# Rare Text Semantics Were Always There in Your Diffusion Transformer

**Seil Kang*   Woojung Han*   Dayun Ju   Seong Jae Hwang**

**Yonsei University**
{seil, dnwjddl, juda0707, seongjae}@yonsei.ac.kr

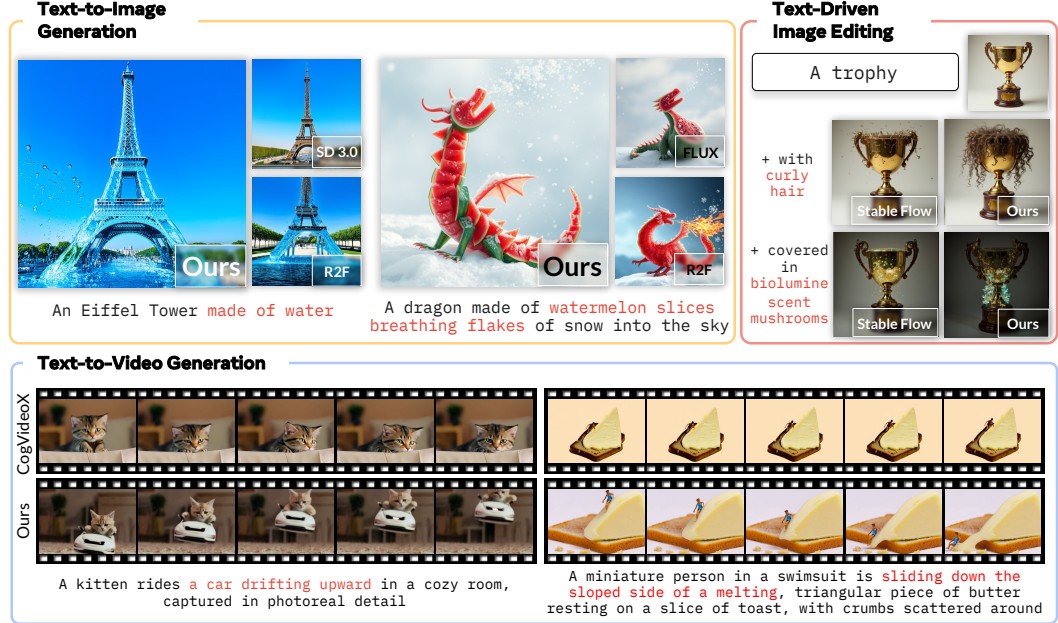

Figure 1: Our method, TORA, achieves superior semantic alignment in text-to-vision outputs for rare prompts while requiring neither finetuning, optimization, nor additional modules; Misfired phrases in the baseline and existing method outputs are highlighted in red.

## Abstract

Starting from flow- and diffusion-based transformers, Multi-modal Diffusion Transformers (MM-DiTs) have reshaped text-to-vision generation, gaining acclaim for exceptional visual fidelity. As these models advance, users continually push the boundary with imaginative or rare prompts, which advanced models still falter in generating, since their concepts are often too scarce to leave a strong imprint during pre-training. In this paper, we propose a simple yet effective intervention that surfaces rare semantics inside MM-DiTs without additional training steps, data, denoising-time optimization, or reliance on external modules (*e.g.,* large language models). In particular, the joint-attention mechanism intrinsic to MM-DiT sequentially updates text embeddings alongside image embeddings throughout transformer blocks. We find that by mathematically expanding representational basins around text token embeddings via *variance scale-up* before the joint-attention blocks, rare semantics clearly emerge in MM-DiT's outputs. Furthermore, our results generalize effectively across text-to-vision tasks, including text-to-image, text-to-video, and text-driven image editing. Our work invites generative models to reveal the semantics that users intend, once hidden yet ready to surface.

---

*equal contribution

39th Conference on Neural Information Processing Systems (NeurIPS 2025).

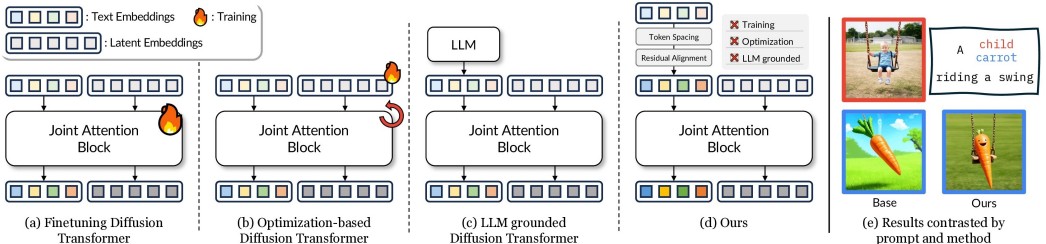

Figure 2: Comparison of existing diffusion-transformer methods: (a) Finetuning, (b) Optimization-based, (c) LLM-grounded guidance, and (d) Ours. (e): Contrastive results between rare and common prompts, comparing baseline and ours. The rotating arrow in (b) shows the latent-vector update loop at each timestep.

# 1  Introduction

Recent progress in text-conditioned generative models has spurred interest in diffusion-based models for generative vision tasks [1–4]. Today, numerous studies have introduced methods based on Multi-modal Diffusion Transformers (MM-DiT), providing sophisticated mechanisms for producing images or videos from textual prompts [3, 4]. As these models advance, users increasingly push the boundary by exploring rare or highly imaginative prompts. However, Park et al. [5] note that such rare concepts (*e.g.,* "an Eiffel tower made of water'' in Fig. 1) appear scarcely in pretraining datasets, often causing generative outputs to misalign with their text semantics. This inherent limitation can typically be overcome by finetuning [6–10], applying optimization techniques during inference [5, 10, 11], or leveraging the capabilities of large language models (LLMs) [5, 8, 12] (refer to Fig. 2(a)-(c)).

However, as illustrated in Fig. 2(e), we observe that a base MM-DiT accurately captures the semantics of a common prompt such as "A child riding a swing", whereas it struggles to represent the meaning of a rare prompt like "A carrot riding a swing". This observation suggests that the model indeed recognizes specific semantics (*e.g.*, "riding a swing") but faces challenges when visually reflecting rare textual semantics. Consequently, we suspect that the semantics of rare prompts exist within the text embeddings but remain inaccessible to the model, leading us to pose the following question: *Can rare textual semantics naturally emerge within MM-DiT embeddings?*

The studies in natural language processing provide valuable insights into this analogous problem. Prior studies [13–16] have shown that text embeddings from transformer-based models naturally exhibit anisotropy, meaning they are dominantly distributed along specific directions in their semantic space, resulting in high cosine similarities. In other words, anisotropic text embeddings geometrically form a hypercone semantic space. However, Cai et al. [15] demonstrated that many models [17–19] naturally exhibit global anisotropy yet display local isotropy (evenly distributed), enabling them to achieve strong contextual understanding by distinguishing between textual embeddings. Inspired by these findings, we examine how *anisotropy* and *isotropy* behave across MM-DiT's joint-attention blocks and how they affect rare textual semantic emergence.

We start with the intuition that clarifying text embeddings in the semantic space could preserve distinct semantics and enhance visual generation. Interestingly, our analysis reveals that *variance scale-up* of text token embeddings amplifies the local isotropy in each joint-attention block, facilitating the clearer emergence of rare semantics. For example, in Fig. 1, when processing the prompt "an Eiffel tower made of water," increasing the representational distance between each text token (*e.g.,* "tower" and "water") helps preserve distinct semantics throughout generation process. Conversely, from a global perspective, the variance scale-up increases global anisotropy, yet does not necessarily hinder semantic emergence, in line with prior findings [20–23]. Furthermore, we demonstrate that this intervention benefits contextualization during text-wise self-attention within joint-attention. Nevertheless, while promising, we clarify that variance scale-up may inadvertently amplify embedding directions unrelated to meaningful semantics. Considering all these aspects, we seek a practical approach that capitalizes on the advantages of variance scale-up while accounting for its potential pitfalls, aiming for semantic emergence robustly in text-conditioned generation.

We introduce **ToRA**, short for *Token Spacing* and *Residual Alignment*. Instead of retraining the model or using external modules (*e.g.,* LLMs) as shown in Fig. 2(a)-(c), ToRA has small interventions

(Fig 2(d)). We consider the text embedding space into two distinct subspaces: a principal space and a residual space can be partitioned via Principal Component Analysis (PCA). *Token Spacing* scales up variance within the principal space, thereby enhancing semantic distinguishability. Concurrently, *Residual Alignment* refines the embeddings by rotating their residual space toward the semantic direction [24–26], mitigating unintended side effects from the token spacing. Together, these complementary methods ensure that rare semantic tokens are not overshadowed by other tokens.

To sum up, we bear out that *rare textual semantics can naturally emerge within MM-DiT embeddings with our intervention*. TORA unleash dormant semantics seamlessly integrating into standard MM-DiT inference. Its exceptional generalizability is validated across text-to-vision tasks by the impressive outcomes in video generation and image editing shown in Fig. 1. With the help of its low-overhead and high-compatible features, TORA can be easily incorporated with existing methods [5, 27, 28]. Moreover, TORA also consistently improves semantic coherence and generative quality across a broad spectrum of textual prompts, including common prompts. Thus, our work shifts the paradigm from attempting to externally impose semantic alignment onto generative models to naturally revealing semantics that were already there, hidden yet waiting to emerge.

## 2 Preliminaries

### 2.1 Joint attention mechanism in MM-DiT.

Multi-modal Diffusion Transformer (MM-DiT) extends Diffusion Transformer (DiT) [29] to multi-modal text-to-vision by jointly processing textual and visual representations, conditioning on CLIP (L/14, G/14) [30] and T5-XXL [31] in SD 3.0 [4] and T5-XXL alone in FLUX.1 [3]. Given $V$ text tokens, $N$ latent-noise tokens, and hidden dimensions $d$, the initial text embedding $e^{\text{init}} \in \mathbb{R}^{V \times d}$ and initial latent-noise embedding $x^{\text{init}} \in \mathbb{R}^{N \times d}$ are concatenated and passed through $B$ joint-attention blocks, at each of which the joint-attention mechanism updates both the text condition and latent noise. Detailed formal explanations are in the §B.

### 2.2 Probing semantic geometry in transformer embeddings.

We primarily focus on concepts of anisotropy and isotropy in text semantic spaces [13–16]. Anisotropy describes embeddings dominated by a few directions, resulting in high cosine similarity regardless of contextual relations, while isotropy indicates embeddings evenly distributed across various directions. Previous studies quantifying these properties within transformer layers reveal that the semantic space of text embeddings is characterized by high *global anisotropy* [13, 14]. Conversely, when these embeddings are clustered by some clustering algorithms [32–34], each cluster exhibits low cosine similarity, indicating local isotropy [15]. With these insights, we examine the beneficial effects of variance scaling on the anisotropic and isotropic properties of text semantic spaces within MM-DiT blocks.

## 3 Analyses

Here, we explore how methodically scaling text embedding variance within joint-attention blocks gives rise to semantic emergence, enabling the model to bring concepts from text prompts with greater visual fidelity, particularly for rare concepts. To confirm the reliability of the observed improvements, we perform an in-depth examination of the text semantic space's *isotropic properties*, considering both local and global perspectives [13–16]. Furthermore, we discuss and analyze potential limitations, noting that variance scale-up may not always be beneficial. Additional setup details are in the §C.1.

### 3.1 Rare Text Semantics Emerges via Variance Scale-Up

We begin with a straightforward intuition [35, 36]: text embeddings should remain distinct in semantic space to effectively represent their underlying concepts, rather than becoming opaque within the model. In other words, we propose that specific semantics of text tokens inherently exist within a set of prompt embeddings, but generative models fail to visually represent these semantics as they cannot effectively retrieve the intended meanings. To address this, we suggest scaling the variance of text embeddings before joint-attention computations, ensuring each embedding has a clear representational

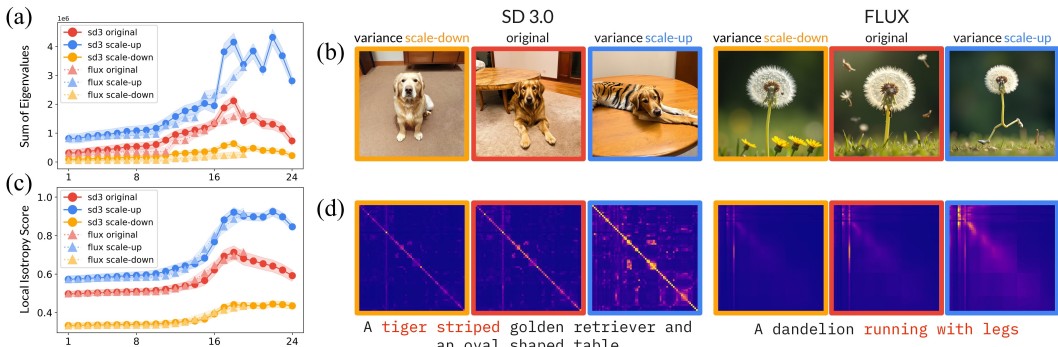

Figure 3: Effects of variance scaling. (a) Sum of eigenvalues of text embeddings across joint-attention blocks, (b) Generated images illustrating visual outcomes, (c) Local isotropy scores across joint-attention blocks, and (d) Visualization of self-attention maps for text embeddings. Results are shown for variance scale-down, original, or scale-up.

margin within the semantic space. Let $e^b$ denote the embeddings at block $b$, with mean $\bar{e}^b$ and variance $\text{Var}(e^b)$. Then, the scaled embeddings $\hat{e}^b$ are given by: $\hat{e}^b = \sigma\left(\frac{e^b - \bar{e}^b}{\sqrt{\text{Var}(e^b)}}\right) + \bar{e}^b$.

To empirically validate the effect of variance scaling, Fig. 3(a) compares the eigenvalue summation of embeddings under variance scale-down ($\sigma < 1$), original, and variance scale-up ($\sigma > 1$) conditions. Fig.3(b) illustrates that variance scale-up significantly enhances the visual representation of rare semantics (*e.g.,* `tiger striped golden retriever`). To further understand how variance scale-up contributes to these visual improvements, we examine its effectiveness in capturing contextual relationships among text tokens by comparing the averaged self-attention maps of original and variance-scaled embeddings across timesteps (Fig. 3(d)). The results indicate that variance scale-up intensifies self-attention activations among text tokens, thereby enhancing inter-token relationships. Subsequently, we conduct a detailed analysis of the semantic space's *isotropic properties*, considering both global and local perspectives [13–16], as elaborated in the following sections.

## 3.2 Variance Scale-Up Makes The Text Semantic Space More Isotropic

Prior research [15] has shown that transformer models inherently exhibit global anisotropy, whereas local isotropy tends to better capture meaningful semantic distinctions. Herein, motivated by this insight, we initially focus on local isotropy. To do so, we first quantify local isotropy at each joint-attention block of MM-DiTs using metrics introduced by Cai et al. [15]:

$$\xi_{local}(e) \triangleq 1 - \left|\mathbb{E}_c[\mathbb{E}_{i\neq j}[\cos(\dot{e}_i^c, \dot{e}_j^c)]]\right|, \tag{1}$$

where indices $i, j$ represent token embeddings, and $c$ indexes local semantic clusters identified using a Gaussian Mixture Model [34]. The centered embedding $\dot{e}_i$ is computed as $\dot{e}_i = e_i - \frac{1}{|c|}\sum_{j\in c} e_j$. Higher scores indicate increased isotropy, suggesting evenly distributed semantic directions. Fig. 3(c) compares the local isotropy scores for variance scale-down, original, and variance scale-up embeddings, empirically demonstrating that variance scale-up improves local isotropy across joint-attention blocks. Moreover, Fig. 3(a) and (c) exhibit closely corresponding trends between the block-wise local isotropy of text embeddings and their eigenvalue summation. As a result, we can confirm that variance scale-up enhances isotropy in the text semantic space, improving the model's text-conditioned visual generation process and resulting in better semantic alignment. Further analysis, other metrics [23], and derivations for local isotropy are in the §C.2.

## 3.3 High Global Anisotropy Is Harmless to Semantic Emergence

Similar to our previous experiments, we also evaluated the effect of variance scaling on global anisotropy [13]. As illustrated in Fig. 4, variance scale-up increases global anisotropy, yet it substantially improves the visual representation of specific text tokens (*e.g.,* `horned`; annotated by blue). This aligns with the machine learning community's finding [20–23] that anisotropy of global perspective allows networks to generalize better to unseen examples, and it can be naturally extended

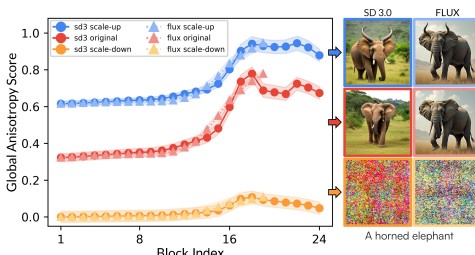

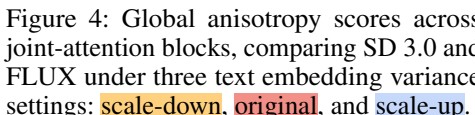

Figure 4: Global anisotropy scores across joint-attention blocks, comparing SD 3.0 and FLUX under three text embedding variance settings: scale-down, original, and scale-up.

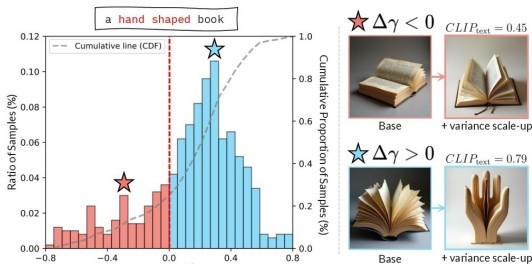

Figure 5: Comparison of $\Delta\gamma$ across 500 seeds under identical prompt conditions. Each star represents a pair of generated images (original vs. scale-up). Red ($\star$) = 0.45, Blue ($\star$) = 0.79 (CLIP$_\text{text}$).

to MM-DiTs [3, 4]. On the other hand, a prior study [14] indicated that high global anisotropy in text semantic spaces could induce semantic opacity in language models. This study proposed reducing global anisotropy by removing principal components identified via PCA [37] to enhance contextual clarity (see §C.3 for more details). However, when we applied this method to MM-DiTs, global anisotropy significantly decreased (yellow lines in Fig. 4), but the visual outputs severely deteriorated (yellow-boxed images), resulting in entirely noisy results. Thus, we conclude: (1) variance scale-up effectively enhances rare semantic emergence by amplifying local isotropy in §3.2, and (2) increased global anisotropy through variance scale-up does not negatively affect rare semantics in MM-DiTs.

### 3.4 Pitfalls of Variance Scale-Up via Semantic Vector Analysis

While variance scale-up for text token embeddings can sharpen inter-token distinctions (§3.1), it raises concerns about potentially deviating from the core semantic direction of a prompt. To explore this possibility, we define the semantic vector $s$ as the difference between a conditional text embedding $e_\text{cond}$, which captures prompt-specific conditions, and its unconditional (null) counterpart $e_\varnothing$ in classifier free guidance (CFG) [24–26]: $s \triangleq e_\text{cond} - e_\varnothing$. The vector $s$ indicates the key shift from unconditional to conditional meaning in the MM-DiT's text embedding space. Through semantic vector $s$, we measure its impact through the cosine–alignment change $\Delta\gamma = \cos(s, \hat{e}) - \cos(s, e)$. We evaluated $\Delta\gamma$ and CLIP scores across 500 random seeds using identical prompts, as illustrated in Fig. 5. The results indicate that variance scale-up successfully led to $\Delta\gamma > 0$ in approximately 75% of random seeds; however, there remain cases where it resulted in $\Delta\gamma < 0$, often failing to render rare semantic elements (*e.g.,* "hand shaped") in the generated images, resulting in lower CLIP scores compared to instances where $\Delta\gamma > 0$. Moreover, we show the mathematical proof for this phenomenon in §C.4. As a result, the variance scale-up alone cannot guarantee the preservation of semantic alignment, motivating the strategy of Residual Alignment introduced in §4.2. We also analyze with different examples that are provided in the §C.4 similar to Fig. 5.

## 4 ToRA: Token Spacing and Residual Alignment

Our previous analyses delivered valuable insights regarding the manipulation of text embeddings within MM-DiT, showing the necessity of scaling up the variance of their token embeddings to support semantic distinguishability. Motivated by these findings, we introduce TORA, as shown in Fig. 6. Specifically, TORA leverages Principal Component Analysis (PCA) to decompose the text semantic space into two distinct segments: (1) the principal space, which captures the dominant semantic dimensions that characterize text tokens, where we apply *Token Spacing* to amplify variance and enable a clear distinction between token embeddings; (2) a residual space comprising less informative or noisy components, where we apply *Residual Alignment* to counterbalance undesirable side effects of token spacing as considered in §3.4. Note that, for a single prompt, we perform PCA independently for each block at each timestep, with no sharing across blocks or timesteps.

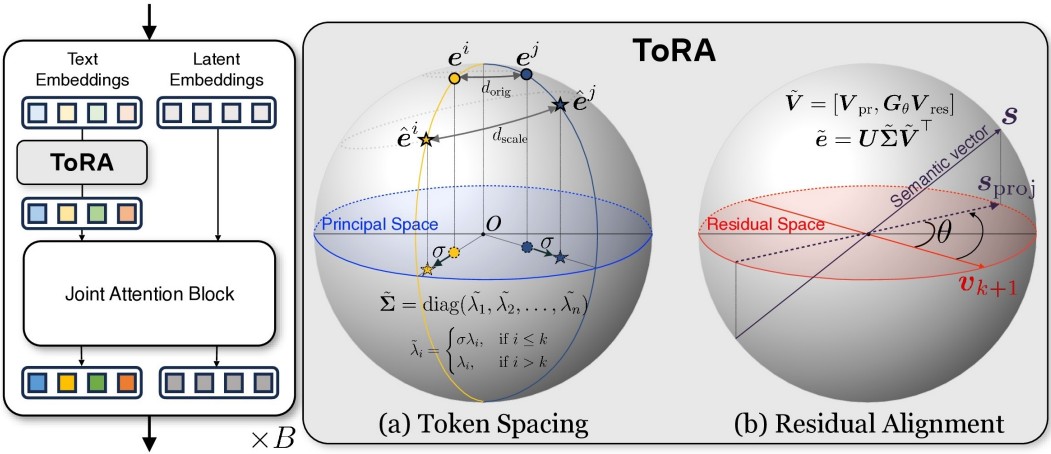

Figure 6: Overview of ToRA. (Left) ToRA is applied to text embeddings before joint-attention blocks. (Right) ToRA operates in two complementary steps: (a) *Token Spacing*, which expands distances among token embeddings by scaling singular values in the principal semantic space; and (b) *Residual Alignment*, which rotates the residual space to align with a target semantic vector.

## 4.1 Token Spacing

We propose Token Spacing to enhance the distinguishability among token embeddings, as illustrated in Fig. 6(a). At each timestep and each joint-attention block $b$, we perform PCA before applying joint-attention on the $e^b$, which is a set of text token vectors. This procedure continues sequentially in every block and timestep. This intervention is uniformly applied across all blocks and timesteps, denoting $e^b$ (for a number of blocks $b$) simply as $e$ for brevity. Initially, we decompose the space of original text embeddings $e$ within each joint attention block using Principal Component Analysis (PCA): $e = U\Sigma V^\top$, where $U$ and $V$ are orthogonal matrices, and $\Sigma$ is a diagonal matrix comprising singular values arranged in descending order ($\lambda_1 \geq \lambda_2 \geq \cdots \geq \lambda_n$). Subsequently, we automatically determine the principal subspace by identifying an optimal cutoff point for top-$k$ dimensions, known as the elbow point. This elbow point, denoted by $k$, is obtained using the Maximum Distance to Chord (MDC) method [38] (see §D.1 for the detailed methodology). Within this identified subspace, Token Spacing is realized by selectively scaling the top $k$ singular values as follows:

$$\tilde{\lambda}_i = \begin{cases} \sigma\lambda_i, & \text{if } i \leq k \\ \lambda_i, & \text{if } i > k, \end{cases} \qquad (2)$$

where $\sigma > 1$ denotes the scaling factor. Finally, the enhanced text embeddings $\hat{e} = U\tilde{\Sigma}V^\top$, where $\tilde{\Sigma} = \text{diag}(\tilde{\lambda}_1, \tilde{\lambda}_2, \ldots, \tilde{\lambda}_n)$. As visually demonstrated by the transformation from $e^i$ to $\hat{e}^i$ in Fig. 6(a), this scaling explicitly increases the embedding distances ($d_{\text{orig}} \to d_{\text{scale}}$), directly enhancing semantic distinguishability among tokens. We further show that our method, Token Spacing, produces a similar effect to amplifying the isotropy characteristics of the text semantic space discussed in §3, as detailed in the §D.2 with detailed explanations and additional analyses thereof.

## 4.2 Residual Alignment

To mitigate potential side effects from variance scale-up (§3.4), such as semantic distortion or unintended amplification of spurious representations in the embedding space, we introduce Residual Alignment targeting the residual space beyond the principal dimensions, as illustrated in Fig. 6(b). Specifically, we rotate the residual subspace spanned by singular vectors $V_{\text{res}} = [v_{k+1}, \ldots, v_n]$ using Givens rotation [39]. Given a target semantic vector $s$, defined as the difference between the conditioned text embedding vector $e_{\text{cond}}$ generated from a specific textual context and the unconditioned text embedding vector $e_\varnothing$ generated without context, we project it onto the residual subspace by removing its principal space component, as follows: $s_{\text{proj}} = s - V_{\text{pr}}V_{\text{pr}}^\top s$, where $V_{\text{pr}} = [v_1, \ldots, v_k]$. As shown in Fig. 6(b), we then perform a Givens rotation $G_\theta$ within the residual subspace to align the first residual singular vector $v_{k+1}$, the most significant direction in the residual

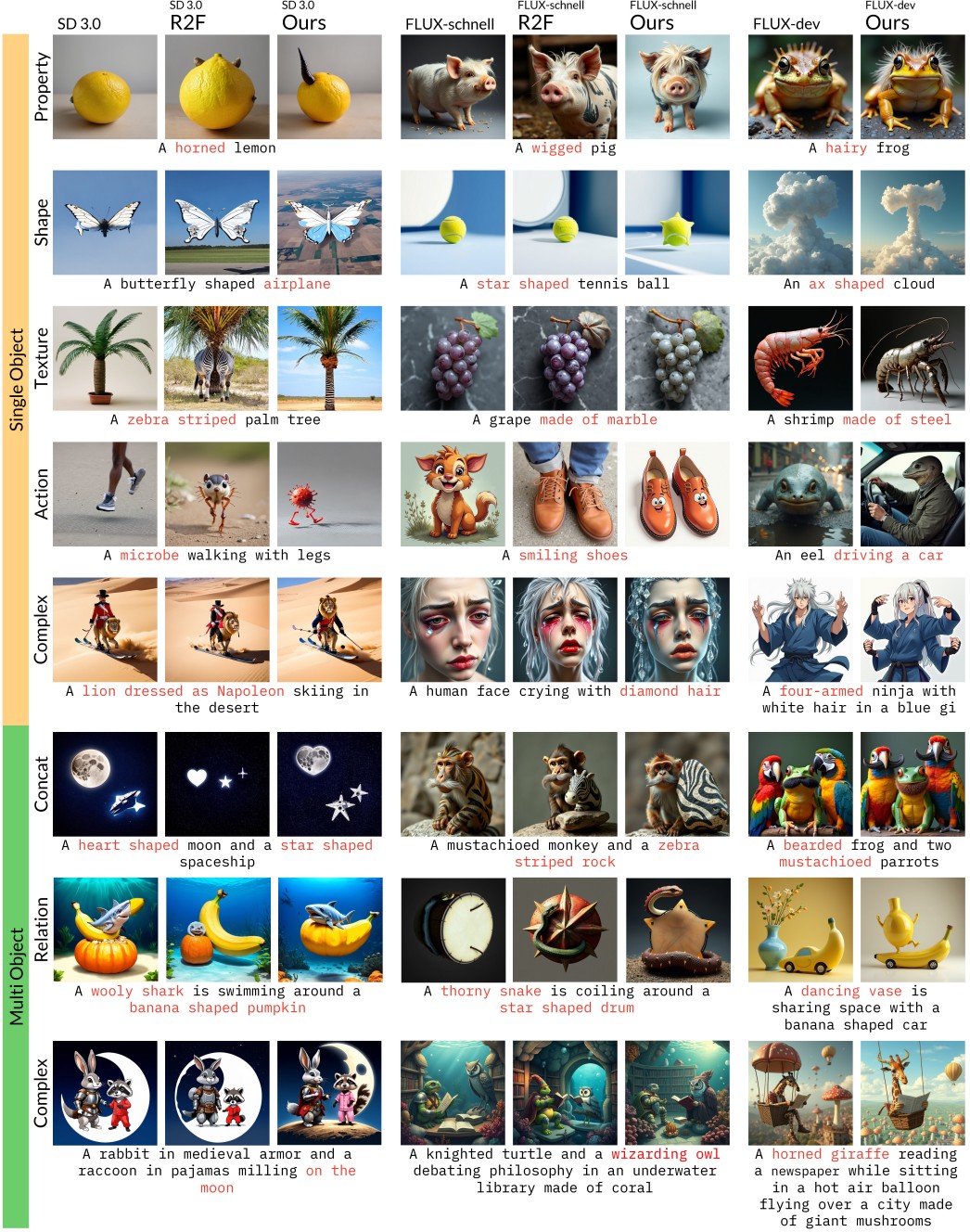

Figure 7: Qualitative comparison between SD3, FLUX-schnell, FLUX-dev, R2F [5], and our method on RareBench [5]. Results are organized by category; Red indicates baseline semantic failures.

subspace, with the projected semantic vector $s_{\text{proj}}$. Formally, the rotation matrix is defined as:

$$G_\theta v_{k+1} = \frac{s_{\text{proj}}}{||s_{\text{proj}}||}. \tag{3}$$

We note that $G_\theta$ is not an arbitrary matrix, but rather a deterministically constructed orthogonal matrix [39], designed to align the dominant residual component $v_{k+1}$ with the semantic direction $s$ by the angle $\theta$ between them. Finally, we rotate the residual subspace vectors accordingly as $\tilde{V}_{\text{res}} = G_\theta V_{\text{res}}$. The enhanced embedding is then reconstructed as: $\tilde{e} = U\tilde{\Sigma}[V_{pr}; \tilde{V}_{\text{res}}]^\top$. Details on Residual Alignment and Givens rotation, along with results addressing the side effects in Fig. 5, are provided in the §D.3.

Table 1: RareBench performance comparison across various models and categories in Image and Video generation. Darker cells indicate the best-performing model in each column, while lighter cells indicate the second-best scores. GPT-4o was evaluated using the same seed setting as in R2F [5].

| | | Single Object | | | | | | | | | | Multi Object | | | | | |
| | Models | Property | | Shape | | Color | | Shape | | Texture | | Concat | | Relation | | Complex | |
| | | GPT4o | Human | GPT4o | Human | GPT4o | Human | GPT4o | Human | GPT4o | Human | GPT4o | Human | GPT4o | Human | GPT4o | Human |
|---|---|---|---|---|---|---|---|---|---|---|---|---|---|---|---|---|---|
| **Text-to-Image** | *MM-DiT Baselines* | | | | | | | | | | | | | | | | |
| | SD3.0 | 49.4 | 33.3 | 76.3 | 69.9 | 53.1 | 51.3 | 71.9 | 62.0 | 65.0 | 65.5 | 55.0 | 40.6 | 51.2 | 45.2 | 70.0 | 60.0 |
| | FLUX-dev | 58.1 | 65.2 | 71.9 | 60.9 | 47.5 | 47.0 | 52.5 | 63.8 | 60.0 | 50.7 | 55.0 | 58.0 | 48.1 | 51.6 | 70.3 | 63.8 |
| | FLUX-schnell | 72.3 | 67.5 | 68.8 | 68.4 | 51.9 | 48.7 | 60.0 | 58.3 | 70.0 | 67.8 | 68.1 | 65.8 | 62.5 | 58.6 | 78.1 | 73.1 |
| | R2F$_{SD3.0}$ | 89.4 | 78.6 | 79.4 | 75.1 | 81.9 | 76.8 | 80.0 | 78.0 | 72.5 | 70.7 | 70.0 | 64.6 | 58.8 | 50.1 | 73.8 | 65.2 |
| | R2F$_{FLUX-schnell}$ | 78.7 | 74.2 | 75.0 | 69.6 | 56.8 | 53.6 | 67.5 | 65.5 | 68.7 | 71.6 | 61.2 | 60.3 | 54.5 | 56.2 | 66.8 | 65.2 |
| | *MM-DiTs w/ ToRA* | | | | | | | | | | | | | | | | |
| | Ours$_{SD3.0}$ | 89.8 | 85.2 | 81.9 | 80.6 | 85.8 | 82.3 | 84.5 | 83.2 | 75.5 | 80.6 | 75.0 | 76.9 | 60.5 | 61.7 | 74.4 | 73.3 |
| | Ours$_{FLUX-dev}$ | 79.2 | 82.6 | 76.6 | 77.7 | 84.6 | 83.8 | 85.9 | 82.0 | 73.1 | 77.1 | 69.4 | 65.5 | 68.1 | 70.4 | 78.8 | 78.0 |
| | Ours$_{FLUX-schnell}$ | 77.5 | 82.6 | 75.5 | 82.9 | 85.9 | 84.6 | 89.5 | 90.1 | 73.8 | 78.6 | 70.6 | 74.2 | 67.3 | 69.9 | 72.3 | 72.8 |
| | R2F$_{SD3.0}$ + Ours | 90.1 | 85.5 | 83.3 | 79.4 | 86.7 | 89.0 | 84.8 | 80.9 | 77.0 | 77.7 | 76.3 | 76.8 | 62.0 | 61.7 | 75.1 | 75.1 |
| | R2F$_{FLUX-schnell}$ + Ours | 79.9 | 81.5 | 77.4 | 78.0 | 72.5 | 73.6 | 67.6 | 69.0 | 73.1 | 74.8 | 74.3 | 75.1 | 57.8 | 58.8 | 70.5 | 69.3 |
| **Text-to-Video** | *MM-DiT Baselines* | | | | | | | | | | | | | | | | |
| | CogVideoX-2B | 35.0 | 38.8 | 52.8 | 48.7 | 54.2 | 54.8 | 45.4 | 44.1 | 51.9 | 54.5 | 62.7 | 60.3 | 38.4 | 41.2 | 60.7 | 60.0 |
| | CogVideoX-5B | 44.7 | 38.3 | 55.5 | 52.8 | 57.8 | 60.6 | 50.2 | 49.6 | 54.8 | 58.3 | 64.5 | 60.9 | 38.6 | 39.4 | 68.8 | 64.8 |
| | *MM-DiTs w/ ToRA* | | | | | | | | | | | | | | | | |
| | Ours$_{CogVideoX-2B}$ | 54.6 | 55.9 | 60.2 | 63.7 | 58.4 | 57.4 | 60.9 | 61.7 | 61.3 | 59.4 | 69.2 | 69.9 | 58.3 | 62.0 | 61.1 | 62.0 |
| | Ours$_{CogVideoX-5B}$ | 60.8 | 61.7 | 62.3 | 65.2 | 57.5 | 59.4 | 68.4 | 67.0 | 70.5 | 68.4 | 71.7 | 70.7 | 69.6 | 68.7 | 71.7 | 72.2 |

# 5 Experiments

**Benchmarks.** We evaluate our method primarily on *Text-to-Image* generation, while demonstrating its versatility in *Text-to-Video* generation and *Text-driven Image Editing*. For image generation, we use RareBench [5] for rare concept evaluation, and T2I-CompBench [40] and GenEval [41] for compositional tasks with common concepts. We further validate generalizability through video generation and image editing evaluations using RareBench.

**Implementation details.** We apply our method to joint-attention-based MM-DiT models, evaluating text-to-image tasks on Stable Diffusion 3.0 [4], FLUX-dev, and FLUX-schnell [3], and extending to CogVideoX (2B/5B)[27] for text-to-video and Stable Flow[28] for image editing. All hyperparameters are maintained at their default settings except for a single scaling factor ($\sigma = 1.3$). Further implementation details are provided in the §D.1.

## 5.1 Main Results

**RareBench in text-to-image generation.** Table 1 shows that our method, ToRA, improves the performance of baseline models [3–5] across all categories in both GPT-4o [42] and human evaluations on RareBench [5]. Additionally, Fig. 7 provides quantitative visual comparisons of generated samples. Our model notably enables the successful emergence and alignment of rare semantics across numerous imaginative prompts. Furthermore, the improved results of R2F + Ours in Table 1 emphasize our model's simplicity and compatibility with existing methods. Furthermore, additional qualitative comparisons with GPT-4o [42] are provided in §H, with additional visualizations across various seeds and prompts in §F.

**Compositional alignments in text-to-image generation.** To assess the model's broader applicability, we extend our evaluation to T2I-CompBench [40] and GenEval [41], which focus on common prompts rather than rare concept cases. As Table 3 shows, the model enhanced with our method outperforms the baseline across all categories on these benchmarks, mirroring its gains on RareBench. This confirms that our approach not only improves performance on rare text inputs but also preserves the intrinsic meaning of common prompts. Qualitative results are provided in the §F.

## 5.2 Generalizability Across Various Text-to-Vision Tasks

**RareBench in text-to-video generation.** To verify the effectiveness of our method across diverse tasks, we extended our approach to text-to-video generation using the MM-DiT-based CogVideoX [27] and evaluated rare-prompt performance on RareBench [5] with GPT4o-MT scores [43] and a human study (Table. 1). In most evaluation categories, our method improves performance by over $20\%p$, reflecting its effectiveness in enhancing semantic alignment across diverse visual As illustrated in Fig. 8, our method enables the baseline model to effectively capture rare concepts from the prompts (*e.g.,* "...fox playing piano..."), allowing these concepts to emerge clearly across temporal motion and action at each frame of the generated videos. Additional results and ablation studies for video generation are provided in the §F.

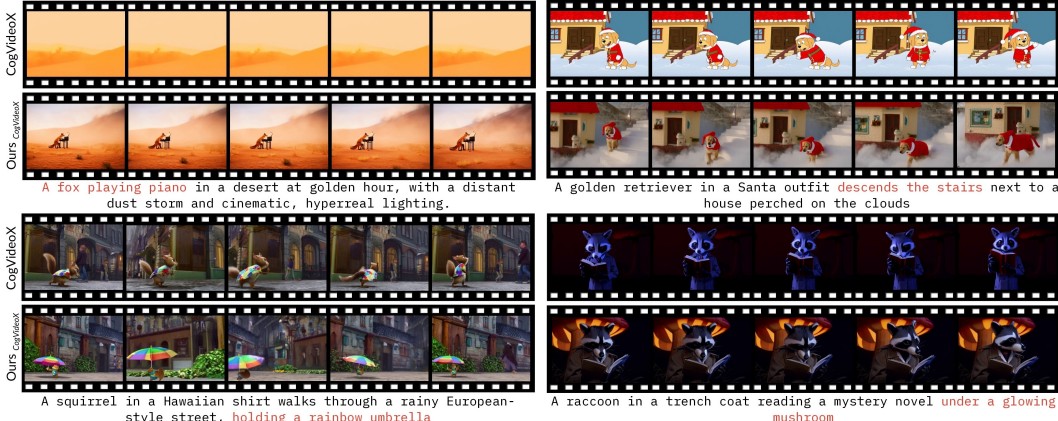

Figure 8: Qualitative comparison of Text-to-Video generation results between the standard CogVideoX [27] and CogVideoX with our method applied, evaluated on custom rare prompts.

Table 2: Performance comparison between baseline and ours in Image Editing.

| Metric | Stable Flow | + Ours |
|---|---|---|
| $CLIP_{img}$ | 0.87 | 0.80 |
| $CLIP_{text}$ | 0.23 | 0.28 |
| $CLIP_{dir}$ | 0.08 | 0.20 |
| Human | 50.8 | 87.7 |
| GPT4o | 62.1 | 82.8 |

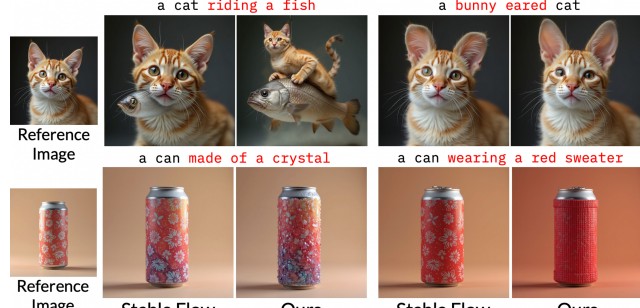

Figure 9: Qualitative comparison of Text-driven Image Editing results between the Stable Flow [28] and our method, evaluated on custom rare prompts.

**RareBench in text-driven image editing.** We investigated the applicability of our method to text-driven image editing with Stable Flow [28]. Stable Flow edits images using reference images generated from simple prompts like "a photo of [OBJ]," where [OBJ] denotes objects from RareBench, allowing us to edit with rare concept prompts. As shown in Fig. 9, our approach helps the model generate rare semantics. Edited images were evaluated using three CLIP-based metrics, GPT-4o, and a human study, following Stable Flow's evaluation framework. In Table 2, our method improved upon the baselines across all evaluation metrics. Notably, the $CLIP_{img}$, which compares reference and generated images, yielded relatively lower scores due to significant visual changes natural to RareBench prompts. See the §F for detailed evaluation metrics, extended results, and ablations on image editing.

## 5.3 Ablation Studies

**Effects of $\sigma$.** To assess the impact of the token spacing parameter $\sigma$, we varied $\sigma$ from 1.0 to 1.5 and measured RareBench scores for each setting. As shown in Fig. 10(a), $\sigma = 1.3$ yields the highest performance across all categories, whereas both lower and higher values produce lower scores. To understand this behavior, we visualized the attention maps in Fig. 10(c) and found that, at $\sigma = 1.3$, the model consistently concentrates on the correct regions corresponding to attributes like "bearded" and "spotted," demonstrating that this range best facilitates fine-grained semantic alignment. See the §F, for additional results of various $\sigma$ values.

**Effects of residual alignment.** In Fig. 10(b), we evaluated Residual Alignment, showing improved performance across all categories when combined with Token Spacing compared to Token Spacing alone. Additionally, Fig. 10(c) indicates that 'w/o Residual Alignment' results in attention maps with appropriately positioned yet weak attention, insufficient for the effective emergence of desired semantic elements in the images. Therefore, Residual Alignment is identified as a crucial technique for guiding embeddings toward semantic directions as discussed in §3.4.

Table 3: Comparison results on broader applicability in Text-to-Image generation using GenEval [41] and T2I-CompBench [40]. Darker cells indicate best scores; Lighter cells are second-best.

| Models | GenEval | | | | | | T2I-Compbench | | | | | | | | | | | |
|---|---|---|---|---|---|---|---|---|---|---|---|---|---|---|---|---|---|---|
| | Single Object | Two Object | Counting | Colors | Position | Attribute Bindings | Color | | Shape | | Texture | | Spatial | | Non-Spatial | | Complex | |
| | | | | | | | GPT4 | BLIP | GPT4 | BLIP | GPT4 | BLIP | GPT4 | UniDet | GPT4 | CLIP | GPT4 | 3-in-1 |
| *MM-DiT Baselines* | | | | | | | | | | | | | | | | | | |
| SD3.0 | 0.98 | 0.74 | 0.63 | 0.67 | 0.34 | 0.36 | 90.3 | 84.0 | 76.2 | 63.3 | 91.3 | 80.1 | 72.0 | 34.0 | 88.5 | 31.4 | 85.2 | 47.7 |
| FLUX-schnell | 0.98 | 0.77 | 0.70 | 0.63 | 0.18 | 0.29 | 88.7 | 82.5 | 71.2 | 59.4 | 90.0 | 78.1 | 73.0 | 35.0 | 88.7 | 31.7 | 83.4 | 45.5 |
| FLUX-dev | 0.98 | 0.81 | 0.74 | 0.79 | 0.22 | 0.45 | 91.7 | 84.3 | 77.1 | 65.0 | 92.9 | 80.0 | 74.4 | 36.5 | 90.1 | 32.3 | 86.1 | 48.8 |
| *MM-DiT w/ LLM* | | | | | | | | | | | | | | | | | | |
| R2F$_{SD3.0}$ | 0.98 | 0.76 | 0.65 | 0.68 | 0.36 | 0.40 | 90.5 | 84.3 | 77.6 | 63.9 | 91.9 | 81.7 | 75.6 | 45.6 | 89.2 | 32.5 | 85.3 | 47.9 |
| R2F$_{FLUX-schnell}$ | 0.99 | 0.77 | 0.59 | 0.61 | 0.20 | 0.45 | 87.4 | 82.5 | 70.9 | 58.5 | 90.4 | 78.7 | 74.7 | 41.4 | 90.2 | 32.0 | 82.1 | 45.4 |
| *MM-DiT w/ ToRA* | | | | | | | | | | | | | | | | | | |
| Ours$_{SD3.0}$ | 0.98 | 0.77 | 0.66 | 0.70 | 0.50 | 0.61 | 90.5 | 84.5 | 77.5 | 64.2 | 92.4 | 82.0 | 77.3 | 45.5 | 88.9 | 33.8 | 86.0 | 48.1 |
| Ours$_{FLUX-schnell}$ | 0.99 | 0.84 | 0.77 | 0.85 | 0.47 | 0.55 | 92.3 | 84.6 | 78.9 | 65.7 | 93.0 | 81.2 | 76.6 | 46.0 | 90.3 | 35.8 | 86.9 | 47.7 |
| Ours$_{FLUX-dev}$ | 0.99 | 0.80 | 0.74 | 0.70 | 0.39 | 0.51 | 90.8 | 84.2 | 73.7 | 61.5 | 91.3 | 79.8 | 74.7 | 46.6 | 90.1 | 33.3 | 83.9 | 46.2 |

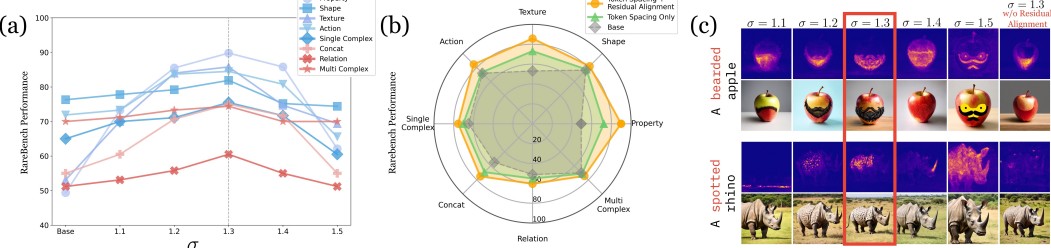

Figure 10: Ablation studies on RareBench [5]: (a) Token Spacing with varying $\sigma$; (b) impact of Residual Alignment; (c) qualitative results and corresponding attention maps.

## 6 Related Work

**Transformer in diffusion models.** Recent advancements in diffusion models have brought to the forefront transformer-based architectures like Diffusion Transformer (DiT) [29], especially Multi-modal Diffusion Transformers (MM-DiTs). MM-DiTs integrate text and visual information through joint-attention mechanisms, significantly enhancing semantic alignment between prompts and generated visuals. In particular, FLUX [3] and SD3.0 [4] now represent state-of-the-art performance, surpassing previous UNet-based approaches [1]. Despite these advances, accurately visualizing complex textual descriptions remains challenging, leading researchers to explore both finetuning methods [6–10] and training-free techniques [5, 10, 11] for improved semantic coherence. Our work shares similarities with these efforts, focusing on pretrained MM-DiT models through simple interventions that require neither structural modifications nor retraining.

**Improving text embeddings in diffusion models.** A fair amount of practical work has explored various strategies to improve text embeddings in UNet-based diffusion models, underpinning better semantic fidelity, compositional accuracy, and controllability. Ahmed and Mittal [44] and Zarei et al. [45] have demonstrated improved prompt fidelity and attribute binding through finetuning embeddings. Other recent studies [5, 46–48] have integrated LLMs, garnering deeper semantic insights and substantially improving complex prompt understanding. Moreover, other approaches have focused on inference-time interventions, such as regulating cross-attention dynamics [49–51], to better reflect prompt semantics. Beyond common prompts, our work investigates how text embedding variance scale-up influences semantic alignment for imaginative and rare prompts, focusing on the relatively underexplored MM-DiT architecture.

## 7 Discussion and Conclusion

In this paper, we have presented ToRA, a simple yet effective intervention for improving MM-DiT models for vision generation tasks of rare text semantics. In particular, we investigated whether text embeddings can naturally emerge their specific meanings in MM-DiT and whether models can effectively apply these semantics to vision generation. We showed ToRA can effectively elicit this *semantic emergence*, leading to improved generation performance. Through extensive evaluation, we confirmed our method's generalizability across *text-to-image, text-to-video, and text-driven image editing tasks*. Lastly, we hope our findings help shift the focus from imposing external semantic constraints to naturally revealing the embedded semantics already there within MM-DiT; We provide more discussion in the §G.

## Acknowledgements

This work was supported in part by the IITP RS-2024-00457882 (AI Research Hub Project), IITP 2020-II201361, NRF RS-2024-00345806, NRF RS-2023-00219019, and NRF RS-2023-002620.

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

# Technical Appendices

# A    Additional Material: Media Gallery

As displaying video content frame by frame within the paper offers only limited insight into temporal coherence and visual quality, we provide a Media Gallery Page featuring the full video outputs from both the main and additional experiments. This page allows for a more faithful assessment of motion consistency and prompt alignment. The results can be viewed at: `https://neurips2025-1573.github.io/`

# B    The Join-Attention in Multi-Modal Diffusion Transformers

Multi-modal Diffusion Transformer (MM-DiT) extends Diffusion Transformer (DiT) to the multi-modal text-to-vision setting, jointly processing textual and visual representations. Conditioning text embeddings come from CLIP (L/14, G/14) [30] and T5-XXL [31] in Stable Diffusion 3 (SD3) [4], and from T5-XXL alone in FLUX.1 [3]. The initial text embedding $e^{\text{init}} \in \mathbb{R}^{V \times d}$, where $V$ denotes the number of text tokens, and the latent-noise embedding $x^{\text{init}} \in \mathbb{R}^{N \times d}$, where $N$ represents the number of image tokens, are processed through $B$ *joint-attention blocks*.

For every block $b \in \{1, 2, \ldots, B\}$, the text condition $e^{b-1}$ and latent noise $x^{b-1}$ are updated. These dual-stream joint-attention blocks (also referred to as multi-modal attention layers or MMATTNs in some literature [52]) are designed to keep the image and text modalities in separate residual streams while allowing for interaction. The process within each block can be detailed as follows:

1. **Adaptive Layer Normalization (AdaLN):** The input image patch representations $x^{b-1} \in \mathbb{R}^{N \times d}$ and prompt token embeddings $e^{b-1} \in \mathbb{R}^{V \times d}$ are first processed by adaptive layer norm (AdaLN) operations. As described by Peebles and Xie [29] for DiTs, these AdaLN layers are conditioned on the diffusion time-step $t$ and a global CLIP vector embedding. The AdaLN operation effectively applies a LayerNorm and then modulates the normalized embeddings using learned affine transformation parameters (scales $\gamma$ and shifts $\beta$, though sometimes only scales are used) derived from $t$ and the CLIP vector. Let $h_{\text{txt}}^{b-1} = \text{AdaLN}(e^{b-1})$ and $h_{\text{img}}^{b-1} = \text{AdaLN}(x^{b-1})$ be the modulated embeddings. These $\gamma, \beta$ parameters (or just $\gamma$) are also used to scale the residual connections later.

2. **Query, Key, and Value Projections:** Separate learned projection matrices are used for text and image modalities to generate queries ($Q$), keys ($K$), and values ($V$). These are applied to the modulated embeddings from the AdaLN step:

$$Q_\mu^{b-1} = h_\mu^{b-1} W_{q,\mu}^b$$
$$K_\mu^{b-1} = h_\mu^{b-1} W_{k,\mu}^b$$
$$V_\mu^{b-1} = h_\mu^{b-1} W_{v,\mu}^b$$

   for each modality $\mu \in \{\text{txt}, \text{img}\}$. Here, $W_{q,\mu}^b, W_{k,\mu}^b, W_{v,\mu}^b \in \mathbb{R}^{d \times d}$ are the modality-specific projection weights at block $b$.

3. **Joint Attention Mechanism:** The core of the block is the attention operation. The attention outputs for text ($o_e$) and image ($o_x$) are computed as:

$$o_e = \left[ \text{softmax}\left( \frac{Q_{\text{txt}}^{b-1}(K_{\text{img}}^{b-1})^\top}{\sqrt{d}} \right) ; \text{softmax}\left( \frac{Q_{\text{txt}}^{b-1}(K_{\text{txt}}^{b-1})^\top}{\sqrt{d}} \right) \right] V_{\text{txt}}^{b-1} \qquad (4)$$

$$o_x = \left[ \text{softmax}\left( \frac{Q_{\text{img}}^{b-1}(K_{\text{txt}}^{b-1})^\top}{\sqrt{d}} \right) ; \text{softmax}\left( \frac{Q_{\text{img}}^{b-1}(K_{\text{img}}^{b-1})^\top}{\sqrt{d}} \right) \right] V_{\text{img}}^{b-1} \qquad (5)$$

   where the $[\cdot\,;\cdot]$ operation indicates that the attention scores (maps) from cross-modal attention (*e.g.*, text queries attending to image keys) and self-modal attention (*e.g.*, text queries attending to text keys) are combined (*e.g.*, concatenated or summed) before being applied to the values of the query's own modality.

4. **Output Linear Layer and Residual Connection:** The attention outputs $o_e$ and $o_x$ are then passed through another linear projection layer. These projected outputs are then added back to the original input embeddings of the block (before AdaLN, *i.e.*, $e^{b-1}$ and $x^{b-1}$),

scaled by a factor (*e.g.,* $\gamma'_{\text{txt}}$, $\gamma'_{\text{img}}$) derived from the AdaLN modulation parameters, thus completing the residual path for this attention stage:

$$\boldsymbol{e}^b = \boldsymbol{e}^{b-1} + \gamma'_{\text{txt}} \cdot \text{Linear}(\boldsymbol{o_e})$$
$$\boldsymbol{x}^b = \boldsymbol{x}^{b-1} + \gamma'_{\text{img}} \cdot \text{Linear}(\boldsymbol{o_x})$$

It's important to note that a full joint-attention block, similar to standard Transformer blocks, would typically also include a position-wise Feed-Forward Network (FFN) or MLP, itself conditioned via AdaLN and with its own residual connection, for both the text and image streams after the attention mechanism described above. The description provided focuses on the multi-modal attention aspects.

This dynamic embedding evolution distinctly differentiates MM-DiT from traditional UNet-based diffusion models. These joint-attention blocks operate sequentially across $B$ layers at each timestep for latent noise prediction.

## C  Analyses Details

### C.1  Setup Details

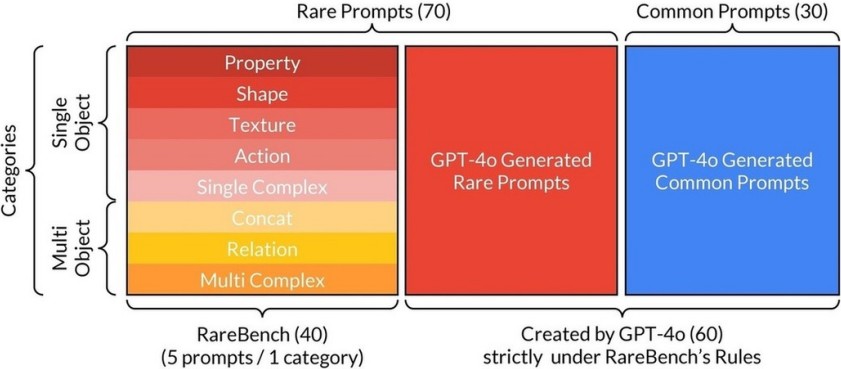

Figure 11: Distribution diagram of prompt samples used in the analysis.

As stated in the main paper, we retain the default settings provided by the baseline models [3, 4, 27, 28] and only adjust the $\sigma$ values derived from our proposed method through ablation studies. In our analysis, we utilize both prompts from the original RareBench [5] and additional prompts generated using GPT-4o [42].

In total, our analysis considers 100 text prompts. 40 prompts come directly from RareBench's eight benchmark categories (five per category) [5], and the remaining 60 prompts are generated with GPT-4o [42], comprising 30 *rare* prompts, created strictly under RareBench [5]'s rarity guidelines, and 30 *common* prompts added as a bias check to ensure that our analysis is not limited to rare cases. A concise overview of the entire prompt set appears in Fig. 11. While the main manuscript reports results with random seed 42 for reproducibility, following the evaluation protocol of R2F [5], the present analysis probes robustness by sampling additional seeds other than 42, thereby capturing stochastic variability that could influence generation quality.

Due to the intrinsic characteristics of the MM-DiT architecture, the initial text embeddings are consistently fed into the first block of each timestep. Additionally, we observed that embedding patterns across the 24 blocks within each timestep exhibit strong similarities from one timestep to the next. Based on this insight, our analyses primarily focus on the behavior observed at the block level.

### C.2  Further Discussion for Local Isotropy

To rigorously verify the isotropic properties of the text semantic space discussed in Section 2.3, we experimentally analyze how variance scale-up impacts isotropy using the IsoScore metric proposed by Rudman and Eickhoff [23], noting that *this metric does not locally measure isotropy*. Additionally, beyond the examples presented in Fig. 3, we provide supplementary examples consisting of generated

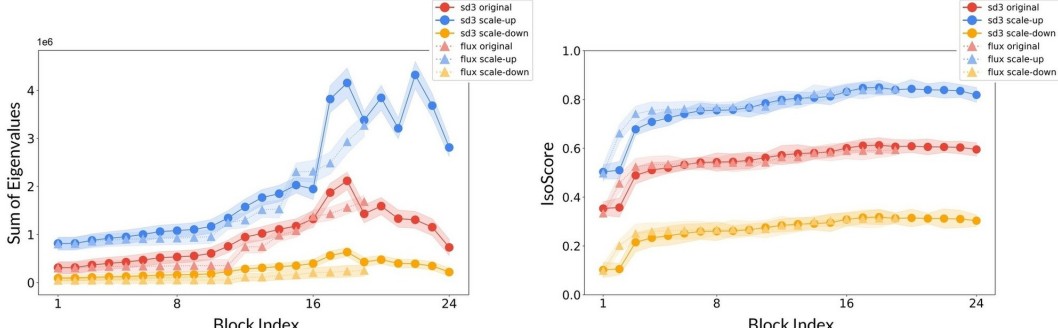

Figure 12: Results of measuring isotropy based on the IsoScore [23] metric with variance scaling. *Left*: A comparative plot of the sum of eigenvalues for the original and variance-scaled data. *Right*: A comparison of IsoScore values before and after variance scaling.

images and their corresponding text self-attention maps to further assess the generality of our approach across diverse prompts.

**Another metric to measure isotropy: IsoScore [23].** While our analysis in Section 2.3 verified the isotropic properties of textual semantics by calculating local isotropy through embedding clustering, Rudman and Eickhoff [23] provides a complementary global perspective by introducing *IsoScore*, an isotropy metric computed without relying on clustering or cosine similarity-based methods. Specifically, given a $V$ text token embeddings represented by the matrix $e \in \mathbb{R}^{V \times d}$ with sample mean $\bar{e}$, IsoScore is calculated as follows:

First, embeddings are projected onto their first $k$ principal components to obtain $e^{\mathrm{PCA}} \in \mathbb{R}^{V \times k}$, and the covariance diagonal $\mathbf{\Sigma}_D$ is computed:

$$\mathbf{\Sigma}_D = \sqrt{k} \cdot \frac{\mathrm{diag}(\mathrm{Cov}(e^{\mathrm{PCA}}))}{\|\mathrm{diag}(\mathrm{Cov}(e^{\mathrm{PCA}}))\|}, \tag{6}$$

where $\| \cdot \|$ is the Euclidean norm. We determine the number of principal components $k$ for each block using the Maximum Distance to Chord (MDC) method [38], as described in Section 3.1. Then, isotropy defect $\delta(e)$ and dimension occupancy $\varphi(e)$ are defined as:

$$\delta(e) = \frac{\|\mathbf{\Sigma}_D - \mathbf{1}\|}{\sqrt{2(k - \sqrt{k})}}, \quad \text{where } \mathbf{1} = (1, \ldots, 1)^\top \in \mathbb{R}^k \tag{7}$$

$$\varphi(e) = \frac{(k - \delta(e)^2(k - \sqrt{k}))^2}{k^2}. \tag{8}$$

Finally, the metric is rescaled to range between 0 and 1, defining IsoScore$^\star$ as:

$$\xi_{\mathrm{IsoScore}}(e) = \frac{k \cdot \varphi(e) - 1}{k - 1}. \tag{9}$$

Here, $\xi_{\mathrm{IsoScore}} \in [0, 1]$, with $\xi_{\mathrm{IsoScore}} = 1$ indicating perfect isotropy (uniform distribution across all principal directions) and values near 0 indicating high anisotropy.

Fig. 12 shows the block-wise IsoScore results computed from MM-DiTs using the procedure described above. The results with variance scale-up exhibit higher isotropy across blocks compared to the original and scale-down. This aligns with our findings using other isotropy metrics, confirming that variance scale-up improves the isotropic properties of the text semantic space.

**More analysis results for local isotropy.** In Fig. 3 of Section 2.2, we observed that variance scale-up in MM-DiTs facilitates rare semantic emergence for rare prompts— a phenomenon not occurring under the original configuration. Additionally, variance scale-up intensifies activation among text

tokens, as reflected in the text self-attention maps, thereby enhancing inter-token relationships. Fig. 18 provides supplementary results supporting this observation, presenting additional examples and corresponding text self-attention maps from SD 3.0 [4] and FLUX [3], beyond those included in the main manuscript.

### C.3  Further Discussion for Global Anisotropy

In the main manuscript we demonstrated, from a global standpoint, that the text-semantic space of MM-DiT exhibits pronounced block-wise anisotropy. Empirically, this anisotropy does not impair the fidelity of the generated images; rather, an overly aggressive attempt to eliminate it produces almost entirely noisy outputs (see Section 2.4 in the main paper). In this section, we present a mathematical derivation of the global anisotropy-reduction procedure adopted in our study, namely the post-processing technique proposed by Mu et al. [14]. We further examine related studies[20–23] to explain, from a global perspective, why text semantic space's anisotropic representations arise naturally in transformer architectures and how moderate regularization, instead of wholesale suppression, preserves the useful structure of the embedding space.

**Reducing global anisotropy via principal component removal.** Mu et al. [14] identify pronounced *global anisotropy* in standard word–embedding spaces, showing that a shared mean vector and a few high-variance principal directions dominate the geometry. They propose a lightweight post-processing step, mean subtraction followed by removal of the top principal components, that markedly restores isotropy and enhances downstream performance. As outlined earlier in our main manuscript, we apply the same strategy to MM-DiT; the detailed mathematical derivation used in our implementation is presented below.

Let the original embedding matrix be $e = [e_1^\top \ldots e_{|V|}^\top] \in \mathbb{R}^{|V| \times d}$.

$$\bar{e} = \frac{1}{|V|} \sum_{i=1}^{|V|} \mathbf{v}_i, \qquad \dot{e}_i = e_i - \bar{e}. \tag{10}$$

$$\mathbf{C} = \frac{1}{|V|} \sum_{i=1}^{|V|} \dot{e}_i \dot{e}_i^\top = \mathbf{U}\mathbf{\Lambda}\mathbf{U}^\top, \tag{11}$$

where $\mathbf{U} = [\mathbf{u}_1, \ldots, \mathbf{u}_d]$ holds eigenvectors in descending order of eigenvalue. Choosing $D = \lfloor d/100 \rfloor$, we form $\mathbf{U}_D = [\mathbf{u}_1, \ldots, \mathbf{u}_D]$ and project out the dominant subspace:

$$\mathbf{e}'_i = \dot{e}_i - \mathbf{U}_D \mathbf{U}_D^\top \dot{e}_i. \tag{12}$$

Following Mu et al. [14], we use $D=15$ when $d=1536$ in Stable Diffusion 3.0 [4]. The resulting block-wise anisotropy scores and corresponding generations are shown in Fig. 4 of our main manuscript, highlighted in yellow.

**Global anisotropy revisited: Extended discussion and related work.** In this section, we further explore the nature of anisotropy within the text semantic space of MM-DiT, building upon long-standing discussions in the Natural Language Processing (NLP) community.

Intuition might suggest that anisotropy, where text embeddings are predominantly distributed across a few key dimensions, could negatively impact downstream tasks. However, our experiments, detailed in Section 2.4 of the main manuscript, reveal its harmlessness concerning the *semantic emergence* effect we proposed. We aim to provide a more comprehensive explanation, drawing upon related works [20–23], of why this characteristic, particularly in transformer models, can be a *natural global phenomenon* that doesn't necessarily correlate with downstream model performance.

*The Implication of Global Anisotropy in Neural Networks:* Contrary to certain traditional NLP literature [13, 14], global anisotropy in text semantic spaces is not inherently detrimental to downstream task performance. Instead, it is increasingly recognized as an intrinsic property of transformer architectures and a direct consequence of stochastic gradient descent (SGD) optimization [20]. This perspective posits that anisotropy, arising from SGD, facilitates the model's convergence to flat minima in the loss landscape, which is known to promote superior generalization compared to sharp minima [20, 23]. Furthermore, extensive research indicates that neural networks inherently compress data into lower-dimensional manifolds [22, 23]. These compressed representations often lead to

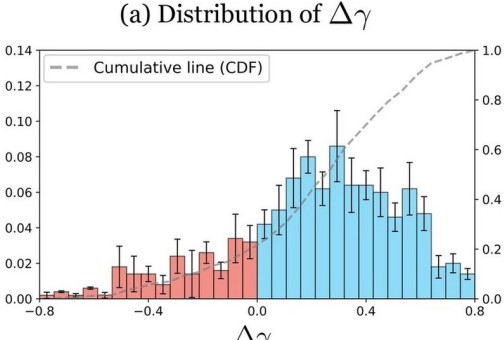
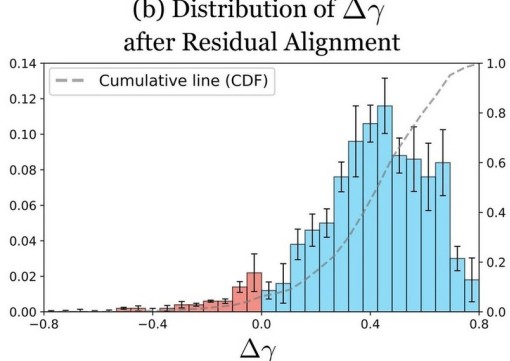

Figure 13: Pitfalls of variance scale-up as seen through $\Delta\gamma$ in additional examples (random sampled 50 prompts from Fig. 11). (a) Analysis of extended samples: It presents results from 50 randomly selected samples used in our analysis. The distribution illustrates the average frequency value for each sample, with error bars indicating the standard deviation. (b) Impact of *residual alignment* on side effects: We examine the change in $\Delta\gamma$ values across samples after applying Residual Alignment, demonstrating its effect on mitigating side effects.

enhanced performance on downstream tasks. Notably, a lower intrinsic dimensionality (ID) in the final layers has been identified as a robust predictor of classification accuracy on test data [21]. This perspective directly contrasts with earlier claims suggesting that *increasing global isotropy (decreasing global anisotropy)* improves model representations, potentially at the cost of performance degradation [23]. Conversely, a growing body of evidence suggests that heightened anisotropy, through this very compression into lower-dimensional manifolds, can lead to performance gains [20].

*Local Isotropy and Generalization:* Several studies [20, 23] corroborate these observations: SGD intrinsically introduces anisotropic noise, which aids in escaping sharp minima and converging to flatter, more generalizable minima. Deep neural networks progressively compress data representations into manifolds of progressively lower intrinsic dimensionality in later layers. This dimensional compression in later layers strongly correlates with improved generalization performance [20]. The low-dimensional representations learned by neural networks are thus believed to prevent overfitting and facilitate generalization by effectively discarding task-irrelevant dimensions. These discrepancies collectively motivated our investigation into *local isotropy* to accurately characterize the text semantic space.

*Our Conclusion:* In conclusion, the evidence strongly suggests two key points. First, global anisotropy naturally emerges from neural network training. Second, the compression of representations into lower intrinsic dimensions is intrinsically linked to enhanced generalization performance. These findings hold true across general neural network learning and classification tasks, as supported by existing literature [20–23]. Lastly, we find that these insights extend naturally to flow- and diffusion-based generative models for text-to-vision tasks that utilize natural language input.

### C.4    Pitfalls of Variance Scale-Up via Semantic Vector Analysis

In Section 2.5 of our main manuscript, we argued that, although variance scale-up demonstrably benefits semantic emergence, it does not invariably producce positive results for every text prompt or random seed. Here, we provide a mathematical derivation that clarifies this limitation and present additional experimental examples beyond Fig. 5 in the main paper.

**Mathematical derivations.** We restate the key definitions from the main paper before delving into new analyses. We first introduce the semantic vector: $s = e_{\mathrm{cond}} - e_{\varnothing}$, the difference between the conditional text embedding $e_{\mathrm{cond}}$ and its unconditional (null) counterpart $e_{\varnothing}$ used in classifier-free guidance (CFG). Let $e$ be the original text embedding and $\hat{e}$ its variance scale-up version. The cosine difference quantifies the effect of variance scale-up on semantic alignment: $\Delta\gamma = \cos(s, \hat{e}) - \cos(s, e)$. A positive value of $\Delta\gamma$ indicates that variance scale-up enhances alignment

with the semantic direction $\boldsymbol{s}$, whereas a negative value signals a degradation. $\Delta\gamma$ can be rewritten as

$$\Delta\gamma = \cos(\boldsymbol{s},\hat{\boldsymbol{e}}) - \cos(\boldsymbol{s},\boldsymbol{e}) = \frac{\boldsymbol{s}\cdot\hat{\boldsymbol{e}}}{\|\boldsymbol{s}\|\,\|\hat{\boldsymbol{e}}\|} - \frac{\boldsymbol{s}\cdot\boldsymbol{e}}{\|\boldsymbol{s}\|\,\|\boldsymbol{e}\|} = \frac{\boldsymbol{s}\cdot\hat{\boldsymbol{e}}\cdot\|\boldsymbol{e}\| - \boldsymbol{s}\cdot\boldsymbol{e}\cdot\|\hat{\boldsymbol{e}}\|}{\|\boldsymbol{s}\|\,\|\boldsymbol{e}\|\,\|\hat{\boldsymbol{e}}\|}. \quad (13)$$

Because the denominator is strictly positive, the sign of $\Delta\gamma$ is governed solely by the numerator $\boldsymbol{s}\cdot\hat{\boldsymbol{e}}\cdot\|\boldsymbol{e}\| - \boldsymbol{s}\cdot\boldsymbol{e}\cdot\|\hat{\boldsymbol{e}}\|$.

Now let the two comparison vectors share a common component $\bar{\boldsymbol{e}}$, the mean of $\boldsymbol{e}$, and differ in a residual direction $\boldsymbol{u} = \frac{\boldsymbol{e}-\bar{\boldsymbol{e}}}{\sqrt{\mathrm{Var}(\boldsymbol{e})}}$, writing $\hat{\boldsymbol{e}} = \bar{\boldsymbol{e}} + \sigma\boldsymbol{u}$ and $\boldsymbol{e} = \bar{\boldsymbol{e}} + \boldsymbol{u}$ with scaling factor $\sigma > 0$. Substituting these forms gives the inner products $\boldsymbol{s}\cdot\hat{\boldsymbol{e}} = \boldsymbol{s}\cdot\bar{\boldsymbol{e}} + \sigma(\boldsymbol{s}\cdot\boldsymbol{u})$ and $\boldsymbol{s}\cdot\boldsymbol{e} = \boldsymbol{s}\cdot\bar{\boldsymbol{e}} + \boldsymbol{s}\cdot\boldsymbol{u}$. Keeping the norms symbolic, $\|\hat{\boldsymbol{e}}\|$ and $\|\boldsymbol{e}\|$ depend on $\bar{\boldsymbol{e}}$, $\boldsymbol{u}$, and $\sigma$ but remain positive, we obtain

$$\begin{aligned}
\boldsymbol{s}\cdot\hat{\boldsymbol{e}}\cdot\|\boldsymbol{e}\| &- \boldsymbol{s}\cdot\boldsymbol{e}\|\hat{\boldsymbol{e}}\| \\
&= (\boldsymbol{s}\cdot\bar{\boldsymbol{e}} + \sigma\boldsymbol{s}\cdot\boldsymbol{u})\|\boldsymbol{e}\| - (\boldsymbol{s}\cdot\bar{\boldsymbol{e}} + \boldsymbol{s}\cdot\boldsymbol{u})\|\hat{\boldsymbol{e}}\| \\
&= (\boldsymbol{s}\cdot\bar{\boldsymbol{e}})(\|\boldsymbol{e}\| - \|\hat{\boldsymbol{e}}\|) + (\boldsymbol{s}\cdot\boldsymbol{u})(\sigma\|\boldsymbol{e}\| - \|\hat{\boldsymbol{e}}\|).
\end{aligned} \quad (14)$$

Hence,

$$\mathrm{sign}\big(\Delta\gamma\big) = \mathrm{sign}((\boldsymbol{s}\cdot\bar{\boldsymbol{e}})(\|\boldsymbol{e}\| - \|\hat{\boldsymbol{e}}\|) + (\boldsymbol{s}\cdot\boldsymbol{u})(\sigma\|\boldsymbol{e}\| - \|\hat{\boldsymbol{e}}\|)) \quad (15)$$

In words, whether $\cos(\boldsymbol{s},\hat{\boldsymbol{e}})$ exceeds $\cos(\boldsymbol{s},\boldsymbol{e})$ is determined by two weighted gaps: the shared-component inner product $\boldsymbol{s}\cdot\bar{\boldsymbol{e}}$ multiplied by the difference of norms, and the residual inner product $\boldsymbol{s}\cdot\boldsymbol{u}$ multiplied by the scaled norm gap. Plugging explicit formulas for the norms, if desired, turns this boxed condition into concrete inequalities for $\Delta\gamma > 0$ or $\Delta\gamma < 0$. This sign rule shows that scaling the residual ($\sigma$) improves alignment ($\Delta\gamma > 0$) when $\boldsymbol{s}$ is more aligned with the residual direction $\boldsymbol{u}$, and worsens it ($\Delta\gamma < 0$) when $\boldsymbol{s}$ is closer to the mean component $\bar{\boldsymbol{e}}$. Consequently, as we mentioned in the main manuscript, even though variance scale-up often helps in a broad sense, it does not automatically improve alignment along the semantic direction—mathematically, $\Delta\gamma$ can still turn negative, so a semantic gain is not guaranteed.

**More analysis results.** Fig. 13(a) extends the analysis of the $\Delta\gamma$ distribution for text embeddings, originally presented in Fig. 5 of the main manuscript, to a larger sample set (50 prompts and 100 random seed per prompt). When $\Delta\gamma < 0$, the average CLIP score registered at $\mathrm{CLIP_{text}} = 0.16 \pm 0.1$, whereas for $\Delta\gamma > 0$, it consistently reached $\mathrm{CLIP_{text}} = 0.45 \pm 0.1$.

# D  Methods Details

In this section, we provide an analysis of our proposed method, *Token Spacing* and *Residual Alignment*. Specifically, we investigate how isotropy, previously analyzed in the main paper in Section 2, is affected by our proposed approach. We further analyze how Residual Alignment mitigates potential side effects arising from Token Spacing, quantifying its effectiveness in enhancing semantic alignment within the embedding space.

## D.1  Implementation Details and GPU Time/Memory Analysis

Our experiments and results derivation were based on Stable Diffusion 3.0 [4] and FLUX.1 [3], both publicly available on HuggingFace. We reproduced the results of R2F [5] using their publicly accessible code from their official GitHub repository. It is worth noting that R2F exclusively supports SD 3.0 and FLUX-schnell within the MM-DiT model family, and does not provide support for FLUX-dev. Our computational resources included a single NVIDIA 48GB A6000 GPU for general experiments and results, with a single NVIDIA 80GB A100 GPU dedicated to Text-to-Video generation.

Table 4: Evaluation of GPU performance across diverse methods for text-to-image generation with Stable Diffusion 3.0.

| Models | SD 3.0 | R2F [5] | Ours |
|---|---|---|---|
| Peak Memory (GB) | 22.02 | 38.49 | 22.17 |
| GPU Time (sec) | $21.30 \pm 1.1$ | $78.13 \pm 2.36$ | $44.74 \pm 1.9$ |

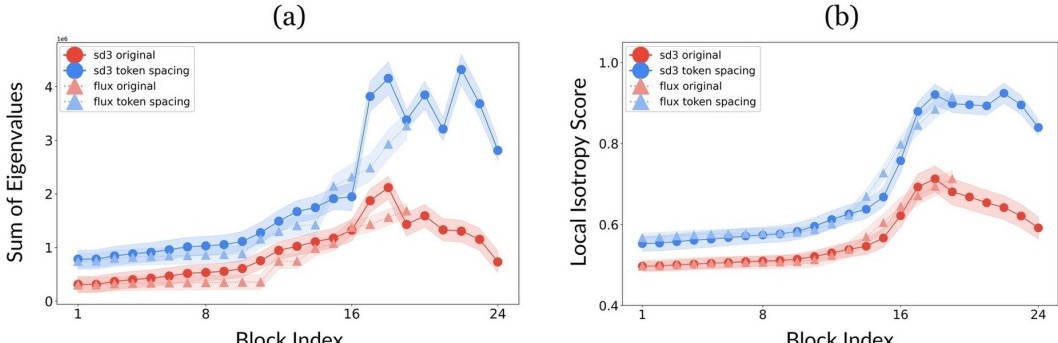

Figure 14: We analyzed the geometric properties of the text embedding space after applying Token Spacing. (a) The left plot shows the sum of eigenvalues for the original text embeddings compared to those with Token Spacing applied. (b) The right plot illustrates the local isotropy score.

Beyond this, for a practical demonstration of our algorithms' performance, we analyzed their GPU time and memory usage on an NVIDIA 48GB A6000 GPU, conducting all experiments ourselves. We measured the resources needed to generate the complex prompt "a horned lion and a wigged elephant", which features two rare concepts, consistent with the experimental setup in Park et al. [5]. For each measurement, we performed 100 trials using different random seeds for the same prompt and report the average values and their standard deviations in Table. 4. These results are limited to the diffusion sampling steps.

### D.2   Effects of Token Spacing for Local Isotropy

We analyze our method, Token Spacing, focusing particularly on its influence on local isotropy and inter-token relationships. We demonstrate that increasing the distances between embeddings enhances the local isotropy, effectively improving semantic emergence. Here, we investigate whether Token Spacing yields comparable benefits. To briefly summarize Token Spacing, we first perform Principal Component Analysis (PCA) on the embedding space, decomposing it into principal and residual spaces. Token Spacing then expands the distances between token embeddings within the principal space, effectively amplifying their variance along principal directions. While this approach conceptually parallels a simple variance scaling, our goal is to specifically examine whether Token Spacing also improves local isotropy, an important factor associated with clearer semantic differentiation among tokens.

As shown in Fig. 14, we observe results that closely align with our intended outcomes from the variance scale-up analysis in Fig. 3 of the main manuscript. Token Spacing elevates local isotropy to a level comparable with direct variance scale-up, validating that Token Spacing effectively boosts the isotropic structure within local cluster neighborhoods.

### D.3   Effects of Residual Alignment for Mitigating the Side Effects from Token Spacing

To mitigate potential side effects (see Section 2.5 and C.4) introduced by variance scaling, we propose an additional technique named Residual Alignment. Residual Alignment involves rotating the residual space, previously obtained through PCA, toward a meaningful semantic direction, enhancing semantic coherence while preserving variance-increased embeddings.

Specifically, as depicted in Fig. 13(a) and (b), we compare the values of $\Delta\gamma$ before and after applying Residual Alignment. Here, $\Delta\gamma$ is defined as $\cos(s, \tilde{e}) - \cos(s, e)$, where $\tilde{e}$ denotes the adjusted embedding. More explicitly, before Residual Alignment, $\tilde{e}$ represents embeddings modified solely by Token Spacing. After applying Residual Alignment, however, $\tilde{e}$ reflects embeddings adjusted by both Token Spacing and Residual Alignment. This distinction allows us to directly evaluate the incremental impact of Residual Alignment on semantic coherence. The results clearly illustrate that Residual Alignment effectively reduces discrepancies, refining semantic alignment and demonstrating its complementary role alongside Token Spacing.

# E  Evaluation Details

We follow the same evaluation protocol as RareBench [5] for both Human Study and GPT-4o evaluation [42]. In the following sections, we describe the scoring criteria for Human Study and GPT-4o, as well as provide a brief explanation of the prompts used for each task. Basically, our evaluation methodology followed Park et al. [5]: all evaluations were initially scored on a $[1, 2, 3, 4, 5]$ point scale by GPT-4o and Human, which were then normalized to $[0, 25, 50, 75, 100]$ for reporting the final benchmark performance.

## E.1  Human Study

For the human study, we recruited 23 distinct participants. Participants evaluated the alignment between the given prompt and the generated images using scores ranging from 1 to 5, where a score of 5 indicates a perfect alignment between the image and text, while a score of 1 means that the image fails to capture any aspect of the prompt. We utilized the same prompt categories defined by RareBench, and participants assessed outputs generated by our model and baseline models under identical conditions within each prompt category. To ensure unbiased evaluation, model identities were anonymized, and the presentation order of generated outputs was randomly shuffled within each category and prompt. The detailed scoring guidelines are identical to those used in the GPT-4o evaluation; please refer to the GPT-4o evaluation example provided below.

## E.2  GPT-4o Evaluation

We conducted an identical evaluation procedure using GPT-4o, employing the same scoring guidelines and assessment protocol as in the human evaluation described above. To ensure consistency and reproducibility, we set the random seed to $42$, following the exact methodology outlined in RareBench [5].

## GPT-4o Instruction: RareBench for Text-to-Image

You are my assistant evaluating the correspondence of an image to a given text prompt.
Focus specifically on:

- Objects in the image and their attributes (e.g., color, shape, texture)
- Spatial layout and positioning
- Action relationships among objects

Evaluate how well the provided image aligns with the following prompt:

**[PROMPT]**

Assign a score from 1 to 5 based on the criteria below:

- **5** : Image perfectly matches the content of the text prompt with no discrepancies.
- **4** : Image portrays most of the content with minor discrepancies.
- **3** : Image depicts some elements, but omits key parts or details.
- **2** : Image depicts few elements, omitting many key parts or details.
- **1** : Image fails to convey the main scope of the text prompt.

Provide your evaluation clearly within 20 words using the format below:

```
### SCORE: [your score]
### EXPLANATION: [brief justification]
```

## GPT-4o Instruction: RareBench for Text-to-Video

You are my assistant evaluating the correspondence of a time-lapse video to a given text prompt.
You will receive eight key frames extracted from the video, each filename indicating its position in a sequence.
Focus specifically on:

- Objects in each frame and their attributes (e.g., color, shape, texture)
- Spatial layout and positioning
- Action relationships among objects
- Consistency and appearance/disappearance of elements across frames

Evaluate how well the provided video aligns with the following prompt:

**[PROMPT]**

Assign a score from 1 to 5 based on the criteria below:

- **5** : All frames perfectly match the text prompt with no discrepancies.
- **4** : Most content matches, but minor discrepancies exist in a few frames.
- **3** : Some key elements match, but several important details are missing or incorrect.
- **2** : Only a few prompt elements appear; many key details are absent or wrong.
- **1** : The video largely fails to convey the prompt's content.

Provide your evaluation clearly within 20 words using the format below:

```
### SCORE: [your score]
### EXPLANATION: [brief justification]
```

**[PROMPT]**

Focus specifically on:

- Changes in objects and their attributes (e.g., color, shape, texture)
- Adjustments in spatial layout and positioning
- Modifications in action relationships among objects

Evaluate how effectively the edited image reflects the intended transformation described by the prompt compared to the reference image.
Assign a score from 1 to 5 based on the criteria below:

**5** : Edited image perfectly matches the intended transformation described by the prompt.

**4** : Edited image effectively conveys the transformation with minor discrepancies.

**3** : Edited image captures some intended transformations but misses key details.

**2** : Edited image reflects few intended changes, omitting many key transformations.

**1** : Edited image fails to convey the intended transformation from the reference image.

### E.3 Evaluation Prompts for Each Benchmarks

**RareBench.** We primarily utilize RareBench [5] to evaluate rare prompts. RareBench comprises prompts featuring single objects across five categories: property, shape, texture, action, and complex. Multi-object prompts are categorized into concat, relation, and complex. Each category contains 40 diverse prompts, totaling a comprehensive set for evaluation.

**T2I-Compbench.** To assess performance on more common prompts, we also evaluate using T2I-CompBench [40]. T2I-CompBench offers a holistic evaluation framework, encompassing six categories: color, shape, texture, spatial relationships, non-spatial relationships, and complex compositions. Each category provides tailored prompts, such as "a blue bench and a green cake" for the color category. For evaluating attributes like color, shape, and texture, we employ BLIP. Spatial relationships are assessed using UniDet for object detection, while non-spatial relationships are evaluated with CLIP. Complex compositions are analyzed using the 3-in-1 evaluation method proposed by T2I-CompBench.

**GenEval.** We also evaluated with GenEval [41], which is an object-focused framework designed to evaluate compositional image properties, including object co-occurrence, position, count, and color. It leverages object detection models to verify the presence and attributes of objects in generated images, facilitating fine-grained, instance-level analysis. GenEval's evaluation pipeline includes tasks such as single object recognition, two-object co-occurrence, counting, color classification, position assessment, and attribute binding. This comprehensive approach allows for detailed evaluation of text-to-image models' capabilities in generating semantically accurate and compositionally coherent images.

## F   Additional Experiement Results

### F.1   Additional Ablation Studies

**Experiments for extended $\sigma$ range on Text-to-Vision.** Fig. 15 presents an extended ablation study, building upon Fig. 10 of the main manuscript, to thoroughly investigate the impact of the hyperparameter $\sigma$ on both Text-to-Image and Text-to-Video tasks. In our method, TORA, $\sigma$ is critically

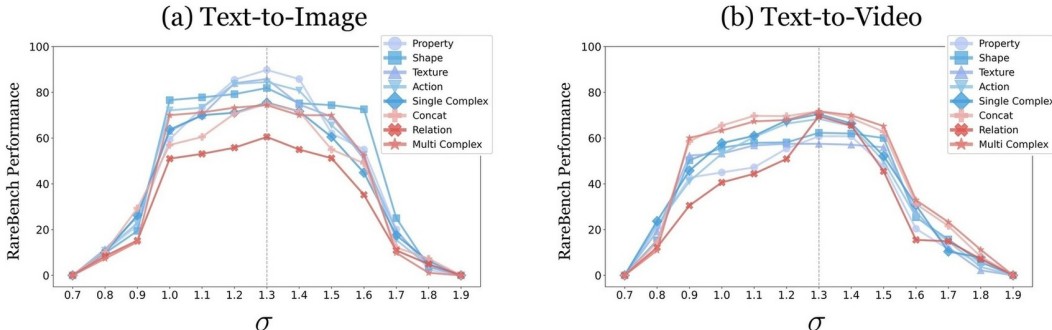

Figure 15: Ablation study investigating the impact of varying the hyperparameter $\sigma$ on our method's performance across the RareBench [5] benchmark. (a) Results plot depicting Text-to-Image performance for each category in RareBench, generated using the SD 3.0 [4]. (b) Corresponding results plot for Text-to-Video performance across RareBench categories, utilizing the CogVideoX-5B [27]. In both tasks, $\sigma = 1.3$ was consistently identified as the optimal hyperparameter setting.

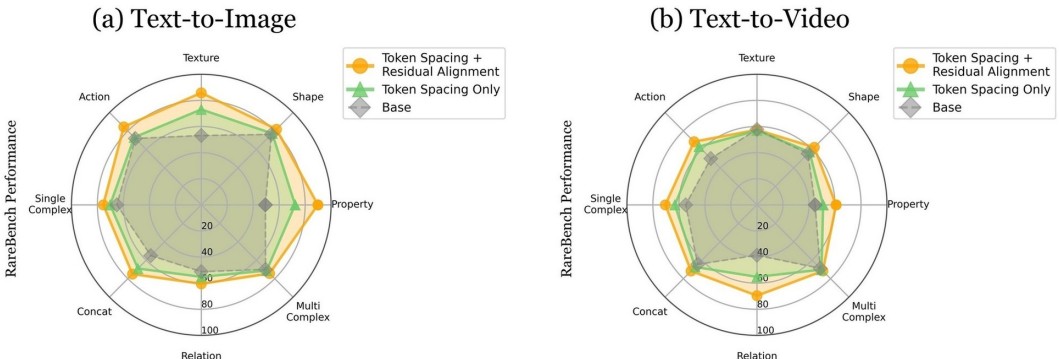

Figure 16: Ablation studies investigating the impact of Residual Alignment within our proposed method, TORA, evaluated on RareBench [5]. (a) For the Text-to-Image task, incorporating Residual Alignment consistently delivered superior performance across all categories, as measured on the SD 3.0 [4]. (b) Similarly, the Text-to-Video task also demonstrated the best quantitative results when Residual Alignment was utilized, with measurements taken on the CogVideoX-5B [27].

utilized for Token Spacing. Our experiments consistently demonstrate that optimal performance for both tasks is achieved at $\sigma = 1.3$. A notable observation is the significant performance degradation when $\sigma < 1.0$. Specifically, for $\sigma \leq 0.7$ or $\sigma \geq 1.9$, the generated outputs consistently exhibit characteristics close to pure noise or bear no relevance to the input prompt whatsoever, resulting in the lowest possible score of 0.0. While our method exhibits some sensitivity to the value of $\sigma$, it is crucial to highlight that within the range of $1.0 < \sigma \leq 1.5$, our model consistently outperforms the baseline across all evaluated categories.

**Effects of *Residual Alignment* on Text-to-Vision.** Fig. 16 presents additional quantitative results demonstrating the impact of residual alignment on our method, building upon the insights from Fig. 10. As evidenced in Fig. 16(b), the integration of residual alignment significantly enhances performance in Text-to-Video generation when combined with token spacing within our framework. This outcome demonstrates that employing only token spacing in our method, TORA, does not achieve optimal performance due to the side effects we previously identified in Section 2.5 of the main paper and Appendix C.4. Consequently, this quantitative ablation study further validates that our proposed residual alignment effectively mitigates these undesirable symptoms.

**Ablations on Text-Driven Image Editing.** Table 5 presents an extended ablation study, building upon Table 2 from the main manuscript, to comprehensively examine the influence of the hyperparameter $\sigma$ on text-driven image editing performance. Consistent with the findings reported in the main manuscript, the $CLIP_{img}$ scores generally show lower values compared to the baseline across various settings. However, our method achieves the highest directional alignment, measured by $CLIP_{dir}$, at

Table 5: CLIP similarity scores for image, text, and directional alignment across different $\sigma$ values of Stable Flow [28] + Our approach on Text-Driven Image Editing. The analysis also demonstrates the impact of residual alignment in our methods. The values highlighted in yellow are taken from our main paper, and those marked in blue indicate the highest-performing results per metric.

| Experiments | $\text{CLIP}_{\text{img}}\uparrow$ | $\text{CLIP}_{\text{text}}\uparrow$ | $\text{CLIP}_{\text{dir}}\uparrow$ | GPT-4o$\uparrow$ |
|---|---|---|---|---|
| *Ablation Studies for Various $\sigma$ (w/ residual alignment)* | | | | |
| 0.7 | 0.77 | 0.23 | 0.13 | 55.7 |
| 0.8 | 0.82 | 0.25 | 0.14 | 68.4 |
| 0.9 | 0.82 | 0.25 | 0.13 | 72.9 |
| 1.0 | 0.83 | 0.25 | 0.14 | 73.4 |
| 1.1 | 0.82 | 0.28 | 0.17 | 77.2 |
| 1.2 | 0.81 | 0.28 | 0.18 | 79.3 |
| 1.3 | 0.80 | 0.28 | 0.20 | 82.8 |
| 1.4 | 0.81 | 0.28 | 0.20 | 84.5 |
| 1.5 | 0.82 | 0.29 | 0.18 | 87.2 |
| 1.6 | 0.81 | 0.29 | 0.18 | 86.1 |
| 1.7 | 0.82 | 0.29 | 0.16 | 81.4 |
| 1.8 | 0.81 | 0.29 | 0.17 | 70.6 |
| 1.9 | 0.80 | 0.29 | 0.17 | 68.4 |
| *Ablation Studies for Residual Alignment ($\sigma = 1.3$)* | | | | |
| w/o residual alignment | 0.83 | 0.25 | 0.14 | 68.4 |

$\sigma = 1.3$ and $\sigma = 1.4$. Additionally, we observe a clear trend where increasing $\sigma$ consistently improves $\text{CLIP}_{\text{text}}$ and $\text{CLIP}_{\text{img}}$ performance, whereas significantly lower values, such as $\sigma = 0.7$, result in substantial performance degradation. For GPT-4o score, performance rises when the scale factor $\sigma$ lies between $1.3$ and $1.6$; pushing $\sigma$ below $1.3$ or above $1.6$ consistently degrades performance, a trend that matches what we observe on other diverse tasks.

Looking at these results, what we find particularly noteworthy is that, given that Stable Flow [28] only controls a subset of layers rather than all layers simultaneously, the variations in performance relative to baseline methods appear limited. Nevertheless, applying either Token Spacing or Residual Alignment methods improves the baseline performance. As previously discussed, choosing an appropriate $\sigma$ value remains crucial for optimal model performance, and it's important to note that while $\sigma = 1.3$ demonstrates strong results overall, it does not universally produce superior performance across all metrics.

**Evaluating robustness across random seeds.** To check if our method consistently performs well no matter the random seeds, we tested its strength on 20 prompts from RareBench [5]. We used 50 different random seeds for each prompt and then took the average score. As shown in Fig. 17, our approach consistently bolsters the baseline's reliability. Moreover, when integrated with other methods [5], we observe a general uplift in benchmark performance.

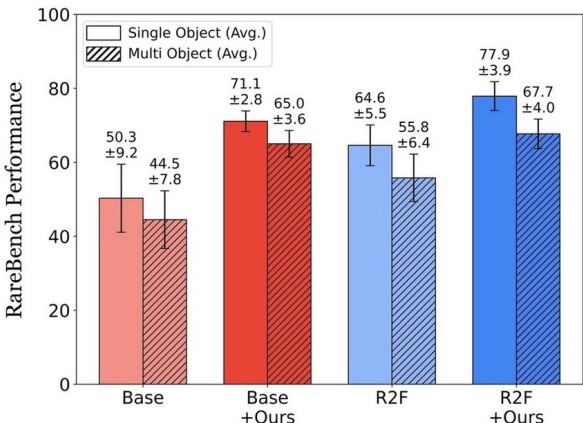

Figure 17: Quantitative results evaluating robustness across random seeds on RareBench.

### F.2 More qualitative results

In this section, we present further qualitative results. These include insights into compositional alignment, alongside Text-to-Image generation, Text-to-Video generation, and Text-Driven Image Editing with rare prompts.

*Note.* Full-page figures are placed at the bottom of the document.

#### F.2.1 T2ICompbench

Qualitative results in Fig. 19-20 demonstrate how our method, TORA, influences the model's compositional alignments in Text-to-Image generation.

#### F.2.2 Text-to-Image Generation

Supplementary Text-to-Image generation results for rare prompts are shown in Fig 21-29.

#### F.2.3 Text-to-Video Generation

We present further Text-to-Video generation outcomes for rare prompts in Fig. 30-34.

#### F.2.4 Text-driven Image Editing

Fig. 35 and 36 illustrate additional Text-Driven Image Editing results for rare prompts.

## G Limitations and Further Discussions

### G.1 Limitations and Future Works

*Architectural Scope*: Our technique is presently tailored to generative models with joint text–image self-attention. Extending it to U-Net backbones [1, 2] and other emerging diffusion variants is a natural next step as the architectural landscape evolves. Moreover, as Diffusion Transformers are an actively researched area in generative modeling, similar to REPA [53], it is essential to further explore our method's applicability to these emerging architectures.

*Prompt Length*: We have not yet stress-tested extremely long prompts [54]. A systematic evaluation of this regime will clarify the method's conditioning limits. Investigating such cases would provide further insights into prompt robustness and conditioning boundaries.

*Broader Applicability for Training Phase with* TORA: Our method was developed as a training-free approach. It would be interesting to investigate how its underlying principles might transfer to the training process itself. Extending our approach to diverse learning settings remains an important direction for future work.

*Deeper Theoretical Insights for* TORA: Finally, while our empirical findings reveal the role of isotropy and anisotropy in semantic representations, exploring deeper theoretical insights into why variance scaling is effective could more concretely explain these observations. Such insights might also bring clarity to the phenomenon of *semantic emergence*, referring to how meaningful semantic properties arise through the interplay of these representational characteristics, offering an exciting avenue for future exploration.

### G.2 Further Discussions

Our investigation centered on the *semantic emergence* within text embeddings in vision generative models, particularly MM-DiT. We found that this phenomenon, where intrinsic meaning naturally surfaces within the model, can be effectively induced through a relatively simple yet potent technique: *variance scale-up*. This effect, we suggest, can be explained by the established properties of isotropy and anisotropy as discussed in natural language processing research.

The significance of this *semantic emergence* finding lies in its potential to facilitate successful textual semantic and often elusive compositional alignment internally within the model's embedding space, achieved through a simple yet effective intervention. This is accomplished without the need for

external modules, such as large language models (LLMs), to forcibly inject semantic alignment. This inherent capability appears to offer broad generalizability, allowing for seamless integration with other methods. Moreover, its applicability is not confined to a single output data type or task, extending to a wide array of text-to-vision tasks that leverage natural language input. We believe this work thus presents an exciting avenue for elevating the intrinsic capabilities and potentially pushing the upper bounds of performance for MM-DiT architectures, a vibrant area of research in modern generative modeling.

## H  Qualitative Comparison with GPT-4o

As GPT-4o [42] currently represents the state-of-the-art in generative models, we evaluate our method in direct comparison to it. Although GPT-4o has recently demonstrated remarkable performance across a variety of generative tasks, its closed-source nature limits direct reproducibility and integration. Our method is built entirely on open-source components, yet achieves results that are qualitatively comparable to GPT-4o across a wide range of prompts, as shown in Fig. 37-40. This highlights the potential of open models to narrow the performance gap while maintaining accessibility and transparency. We also find that in certain instances where prompts require precise numerical understanding, such as "A four armed ninja" in Fig. 40, GPT-4o tends to misrepresent the intended structure. In contrast, our method accurately depicts the number of arms, highlighting improved semantic fidelity in these challenging cases.

> **Note.** Despite being developed entirely with open-source components, our method achieves performance that is *comparable to GPT-4o*, the kind of state-of-the-art in text-to-image generation. Unlike GPT-4o, which often produces visually similar outputs for a given prompt, our method exhibits greater generative diversity across different random seeds, showing its robustness and flexibility. Furthermore, its plug-and-play design allows seamless integration into various generative pipelines, enabling easy experimentation and broader applicability. We believe this accessibility positions our method as a practical and versatile contribution toward the advancement of future generative modeling research.

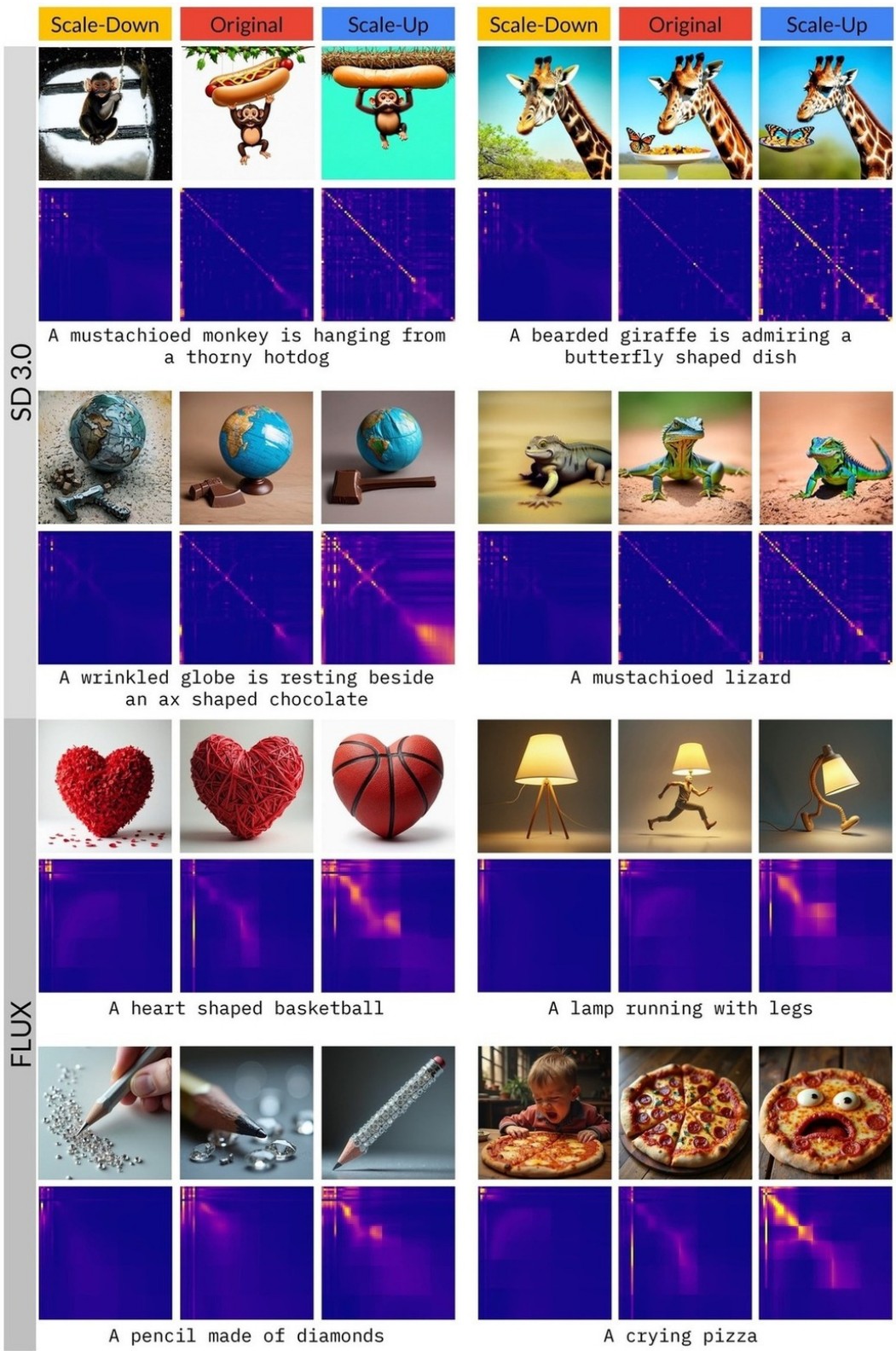

Figure 18: Further results illustrating the influence of variance scaling (Scale-Down and Scale-Up) on text embeddings within the original configuration, examining its effects on both generated images and text-to-text self-attention maps.

Figure 19: Qualitative comparisons of text-to-image compositional alignments: baseline (SD 3.0) vs. baseline + our method.

Left: Original (FLUX-schnell)
Right: Original + Ours

A brown dog and a yellow horse

A brown cow and a pink sheep

A blue boat and a red car

Three helicopters

Six apples

Four tomatoes

A horse on side of a bicycle

A boy on the left of a balloon

A key on the left of a butterfly

A metallic earring and
a leather sofa

A metallic ring and
a wooden knife

A cubic block and a cylindrical
container of hand cream

A circular rug and
a triangular coffee table

A phone hidden by a frog

A fish hidden by a sofa

A gold backpack and a blue clock

The soft white feathers of the
owl contrasted with the sharp
black talons

The glossy apple sat next to the
fuzzy peach and the prickly pear

Figure 20: Qualitative comparisons of text-to-image compositional alignments: baseline (FLUX-schnell) vs. baseline + our method.

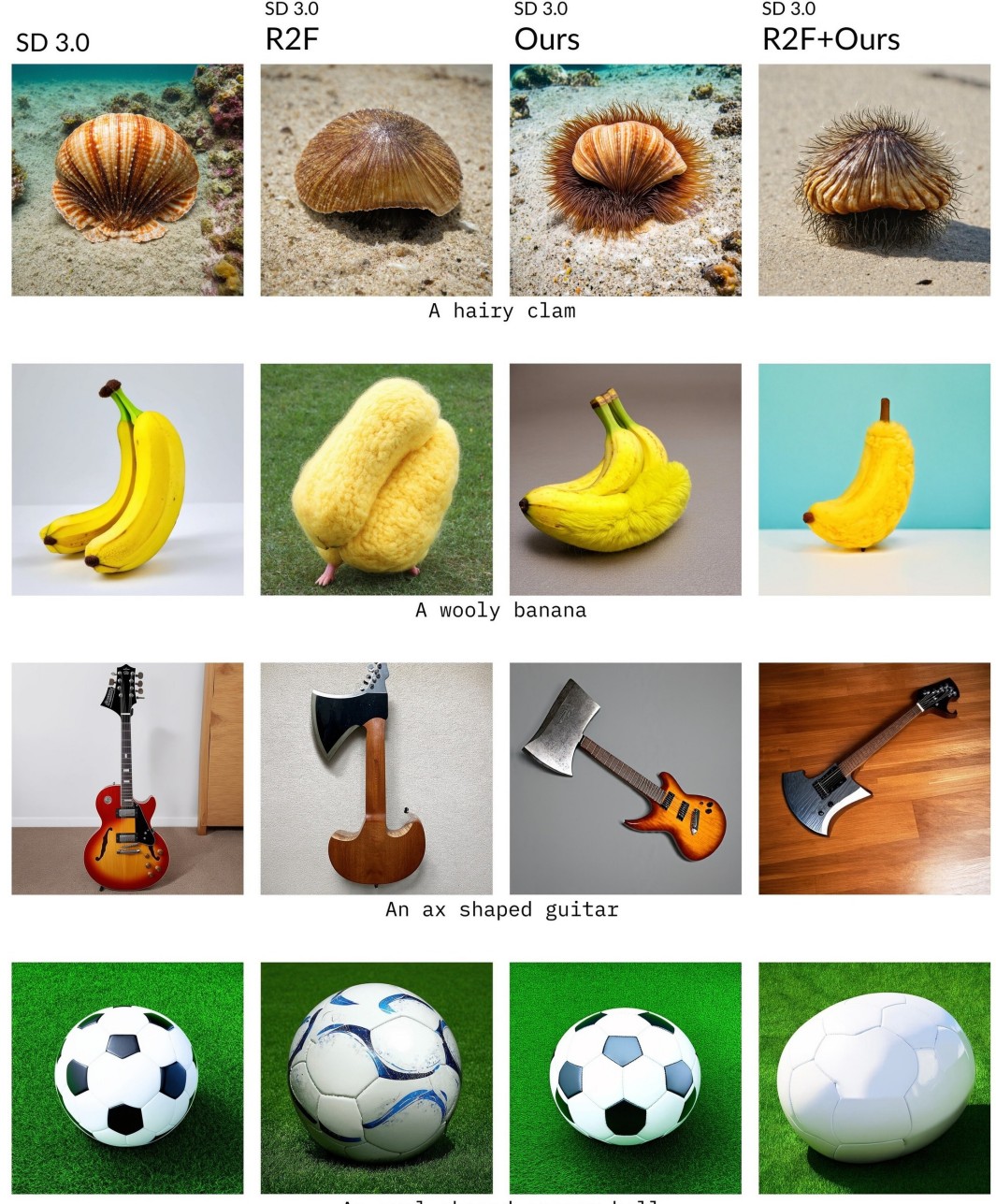

Figure 21: Further Text-to-Image generation comparisons for rare prompts.

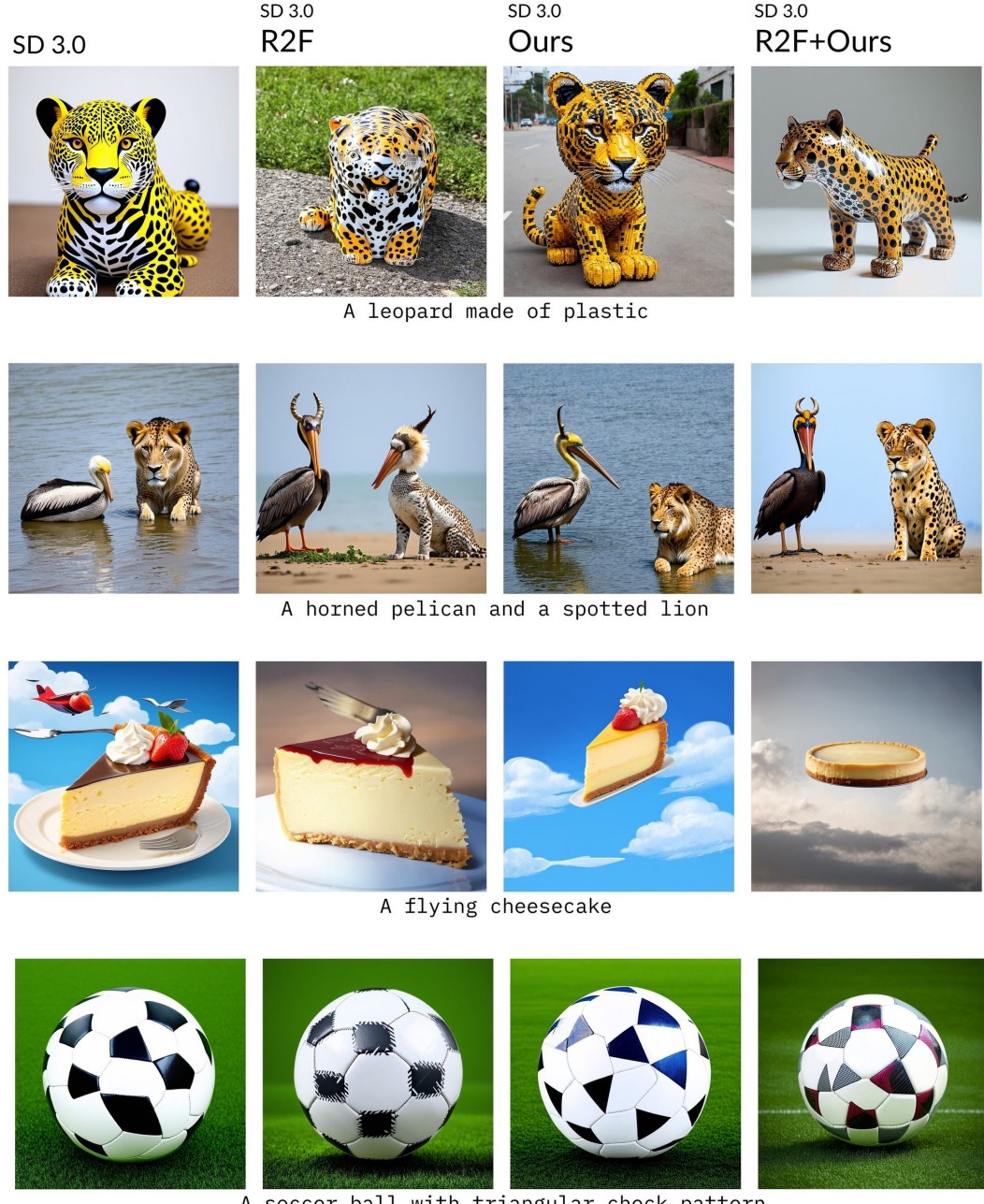

Figure 22: Further Text-to-Image generation comparisons for rare prompts.

SD 3.0              SD 3.0 R2F           SD 3.0 Ours          SD 3.0 R2F+Ours

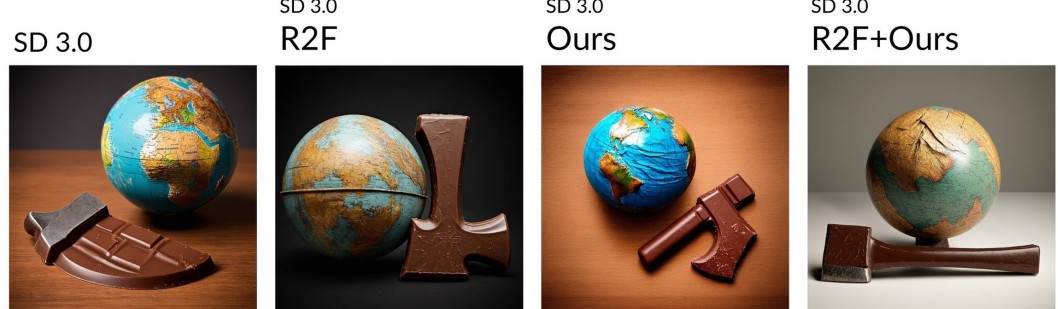

A wrinkled globe is resting beside an ax shaped chocolate bar

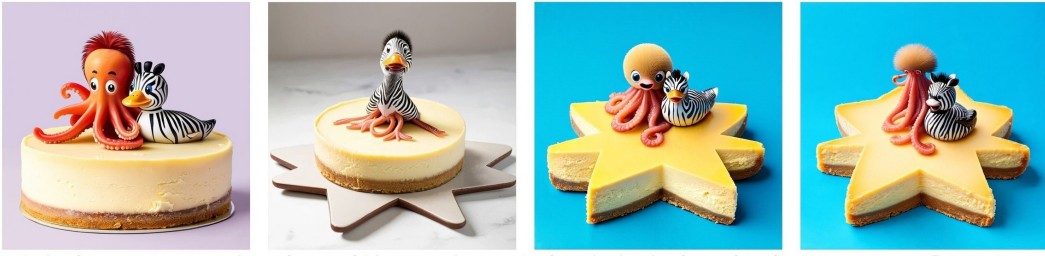

A hairy octopus dancing with a zebra striped duck is sitting on top of a star shaped cheesecake

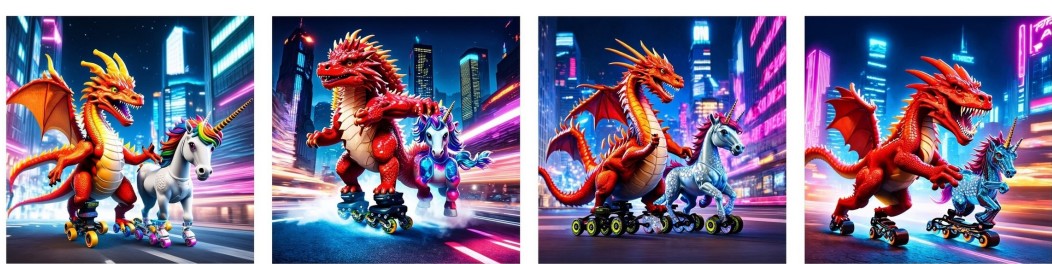

A red dragon with a mane on its back and a unicorn made of diamond rollerblading through a neon-lit alien cityscape at high speed

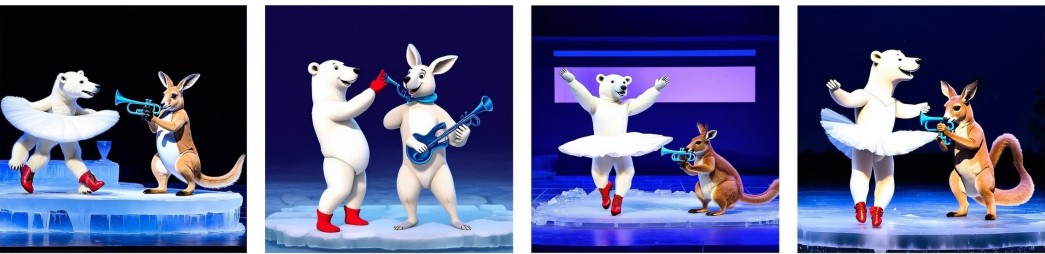

A ballet-dancing polar bear with red toe shoes and a jazz-playing kangaroo with a blue trumpet performing together on a stage made of ice

Figure 23: Further Text-to-Image generation comparisons for rare prompts.

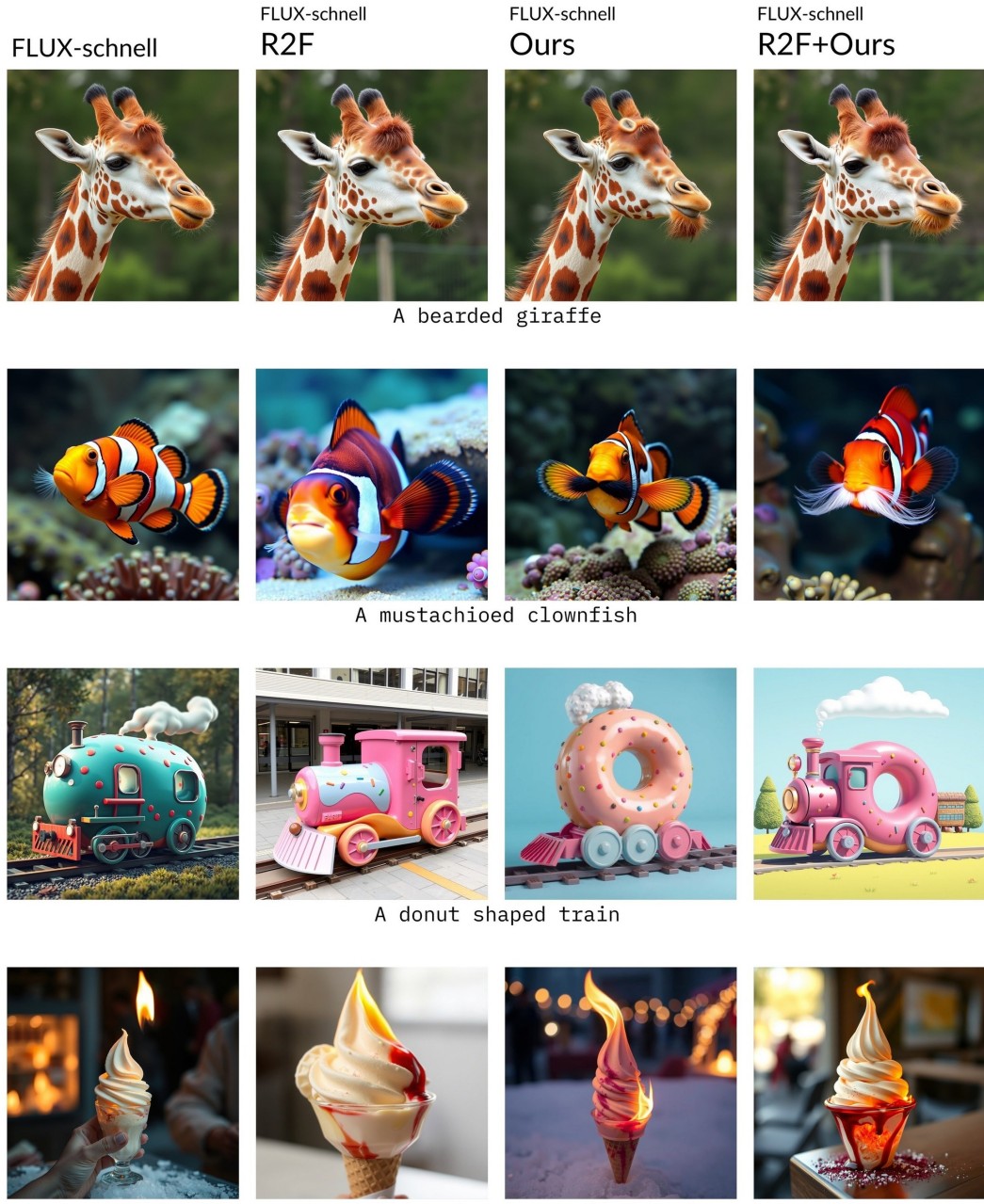

Figure 24: Further Text-to-Image generation comparisons for rare prompts.

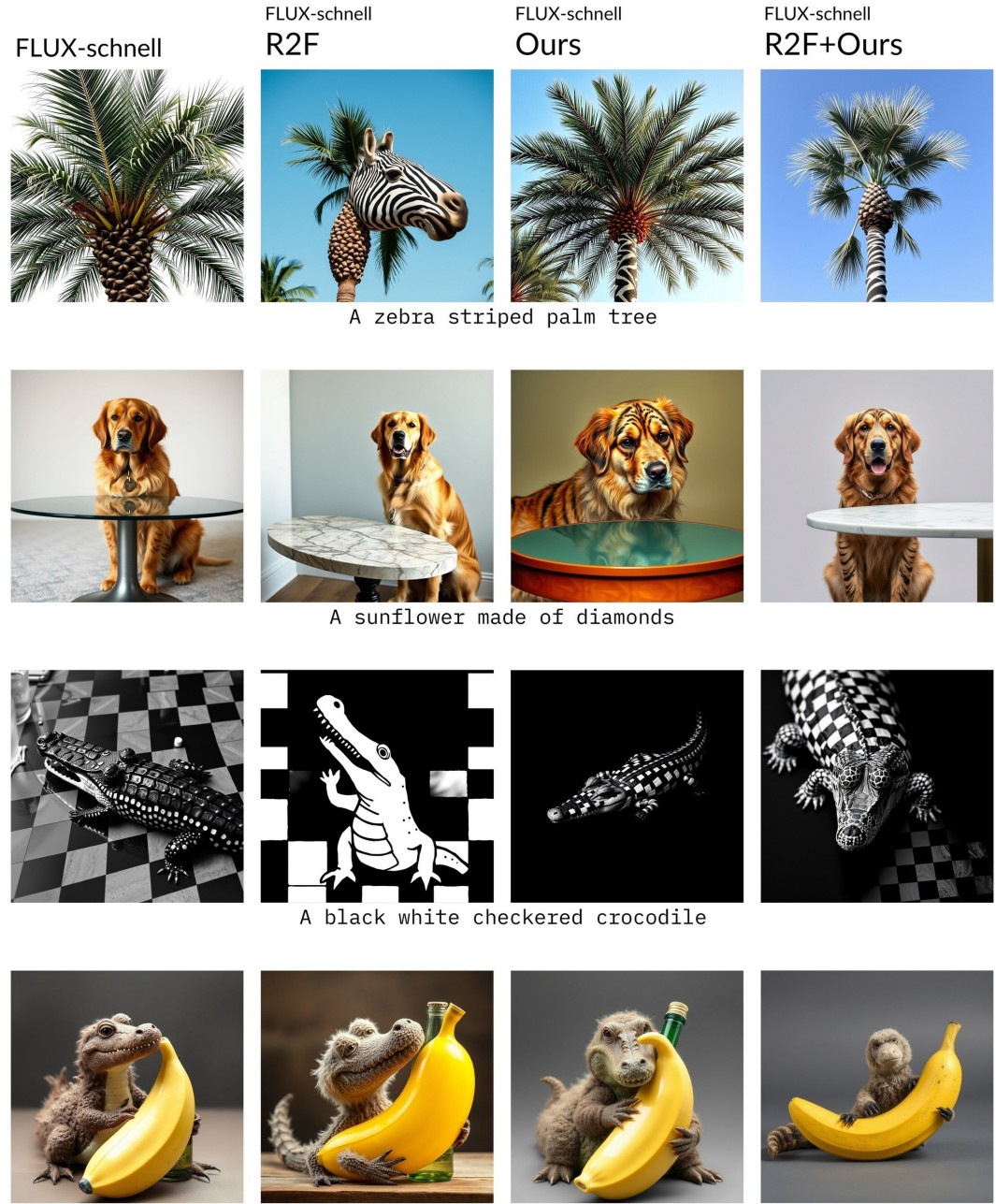

Figure 25: Further Text-to-Image generation comparisons for rare prompts.

FLUX-schnell       FLUX-schnell       FLUX-schnell       FLUX-schnell
                   R2F                Ours               R2F+Ours

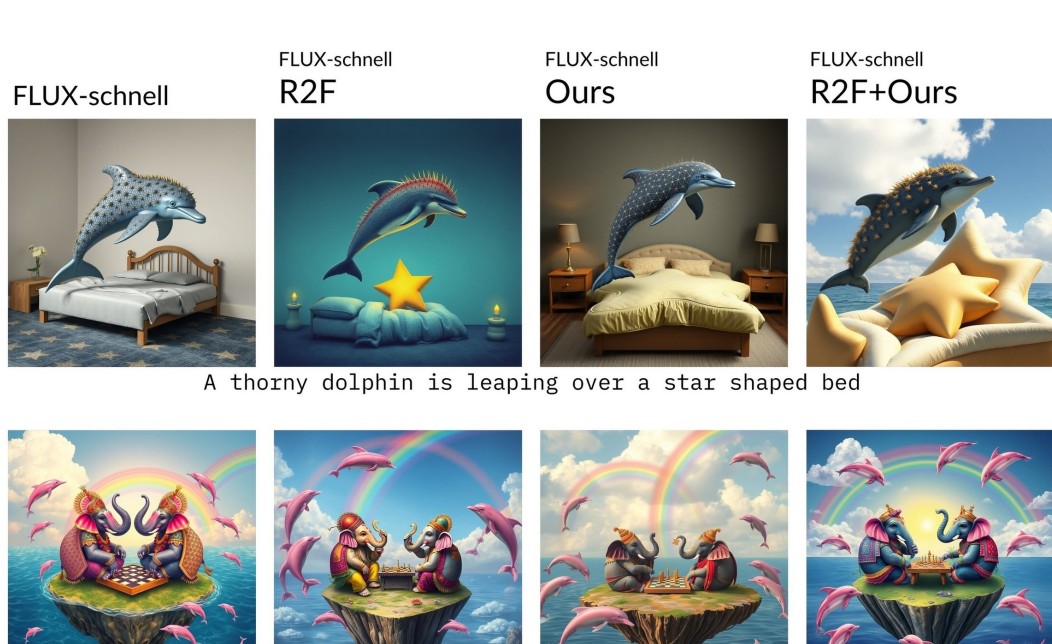

A thorny dolphin is leaping over a star shaped bed

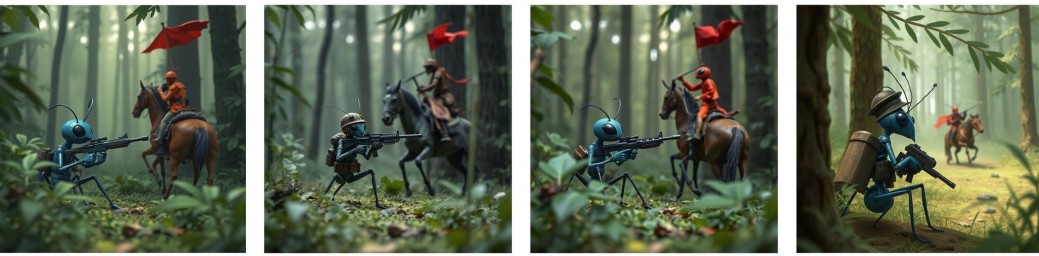

A two decorated indian elephants playing chess on a floating island surrounded
by pink dolphins painting rainbows in the sky

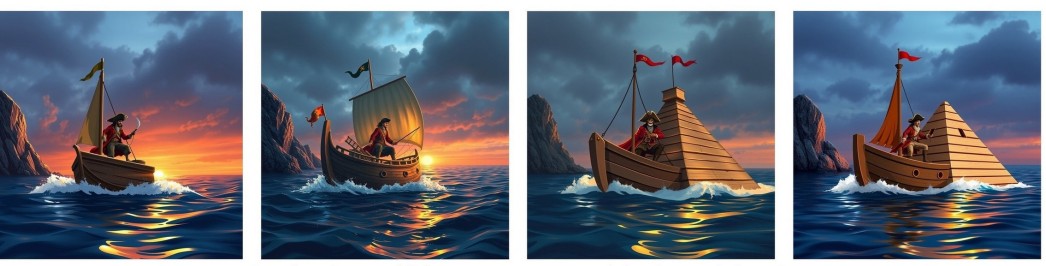

A blue ant hides in the forest in military uniform aiming a gun at a red enemy
ant on horseback

A pirate sailing on a pyramid-shaped boat

Figure 26: Further Text-to-Image generation comparisons for rare prompts.

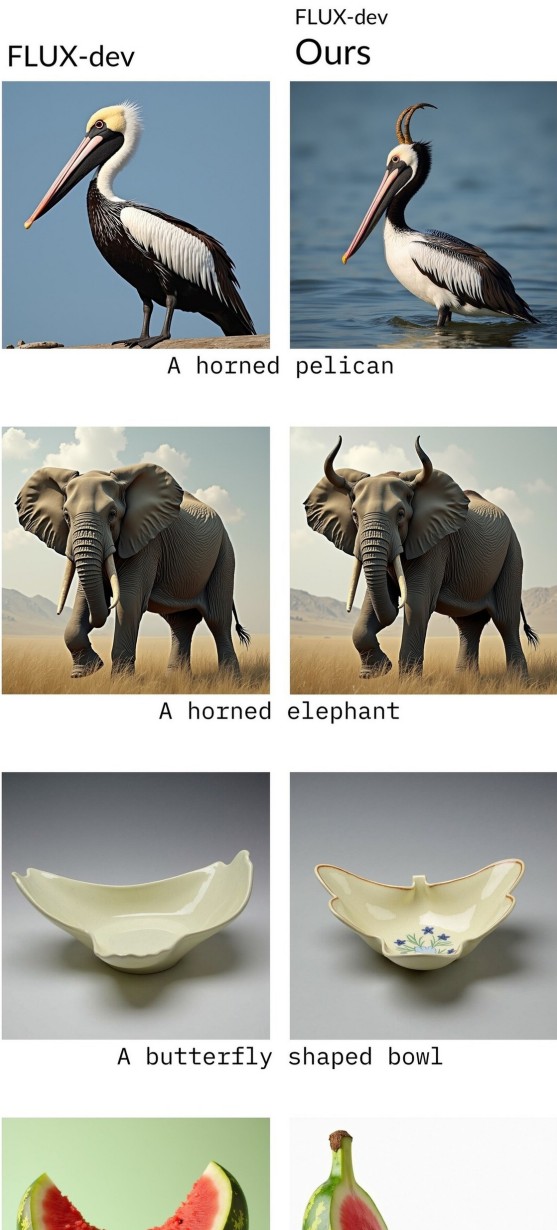

Figure 27: Further Text-to-Image generation comparisons for rare prompts.

FLUX-dev

FLUX-dev
Ours

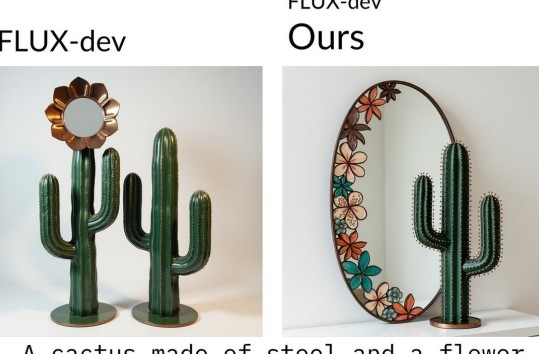

A cactus made of steel and a flower
patterned mirror

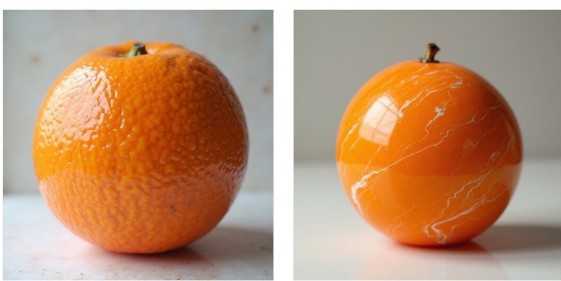

An orange made of marble

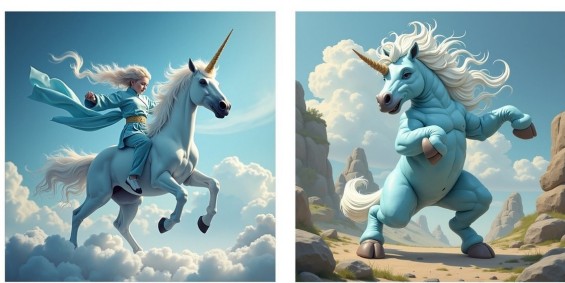

A skyblue unicorn doing kung fu

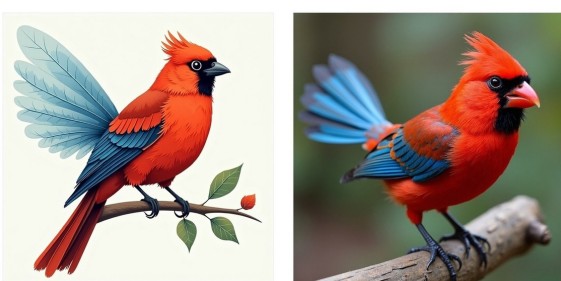

A red bird with blue fish tail

Figure 28: Further Text-to-Image generation comparisons for rare prompts.

FLUX-dev

FLUX-dev
Ours

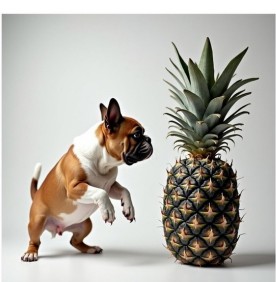 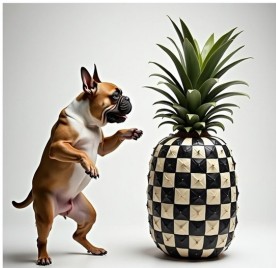

A shrimp made of steel and a spotted
dog

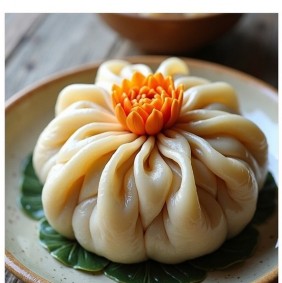 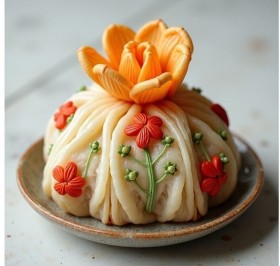

A dancing bulldog is staring at a
black white checkered pineapple

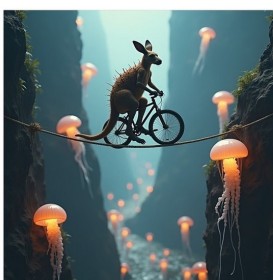 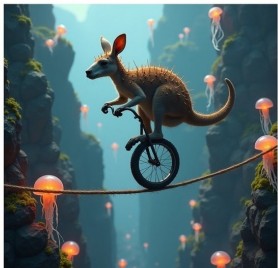

A flower patterned dumpling

A thorny kangaroo riding a unicycle on
a tightrope stretched across a canyon
filled with floating jellyfish
emitting soft glowing lights

Figure 29: Further Text-to-Image generation comparisons for rare prompts

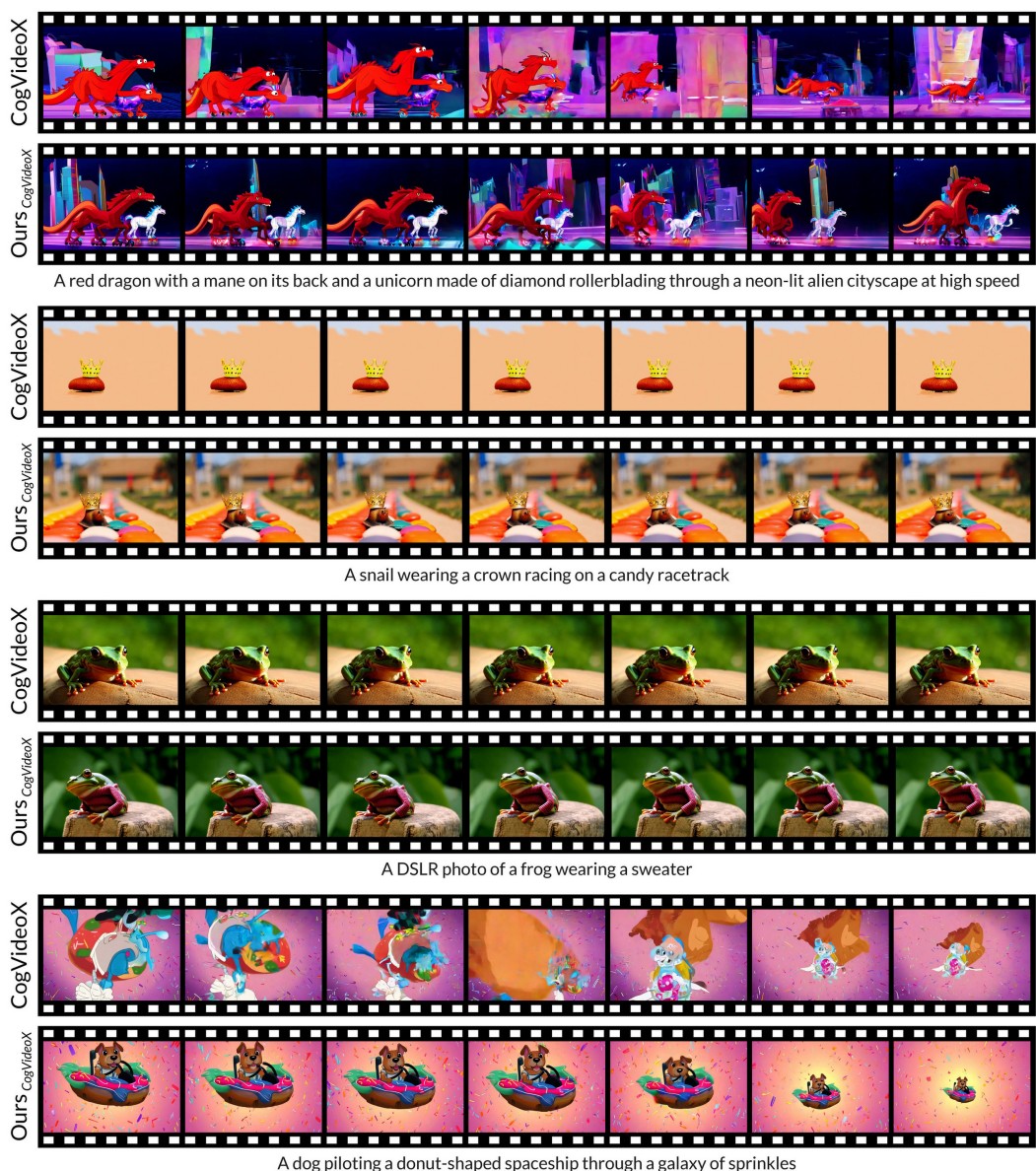

Figure 30: Further Text-to-Video generation comparisons for rare prompts.

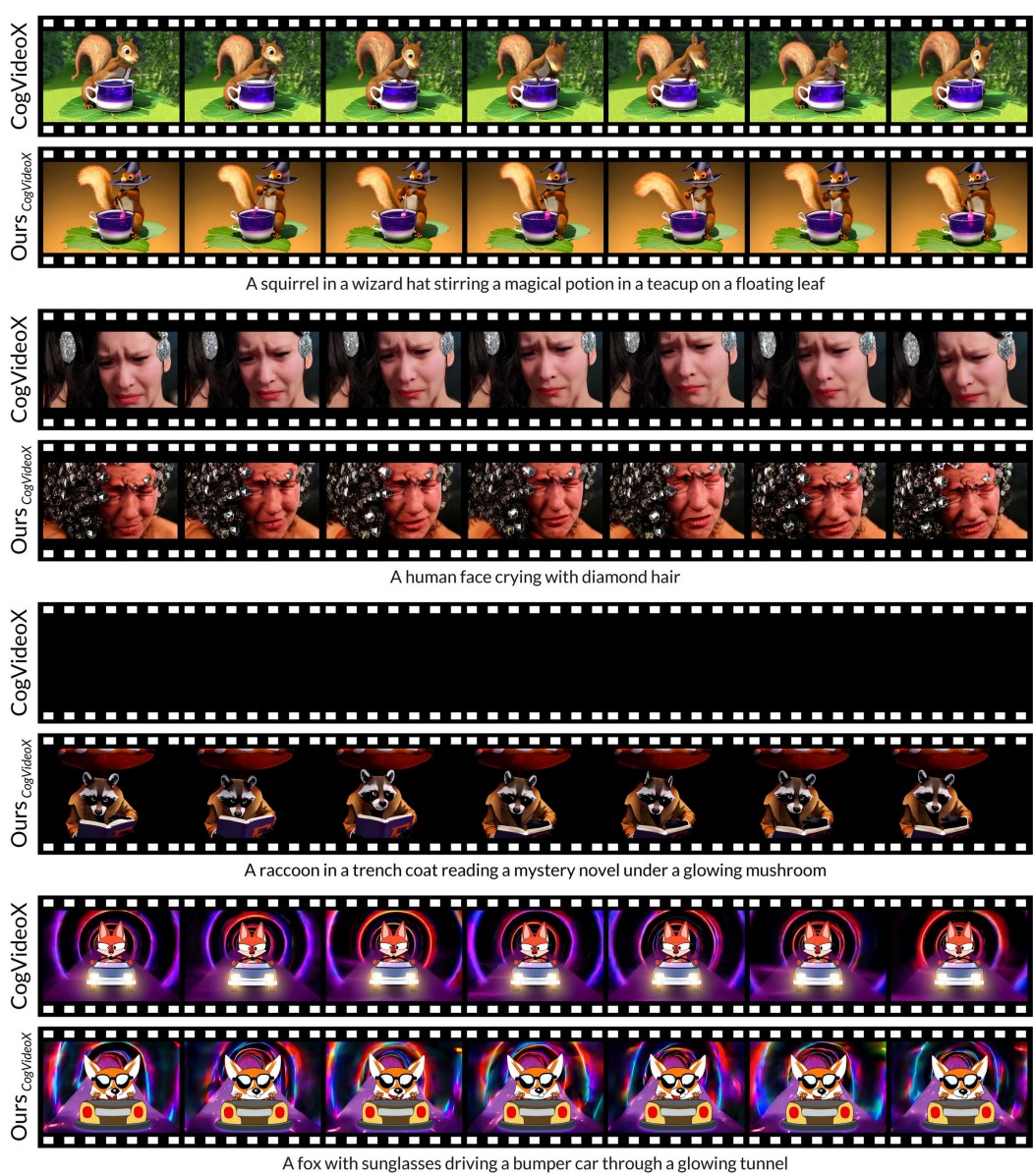

Figure 31: Further Text-to-Video generation comparisons for rare prompts.

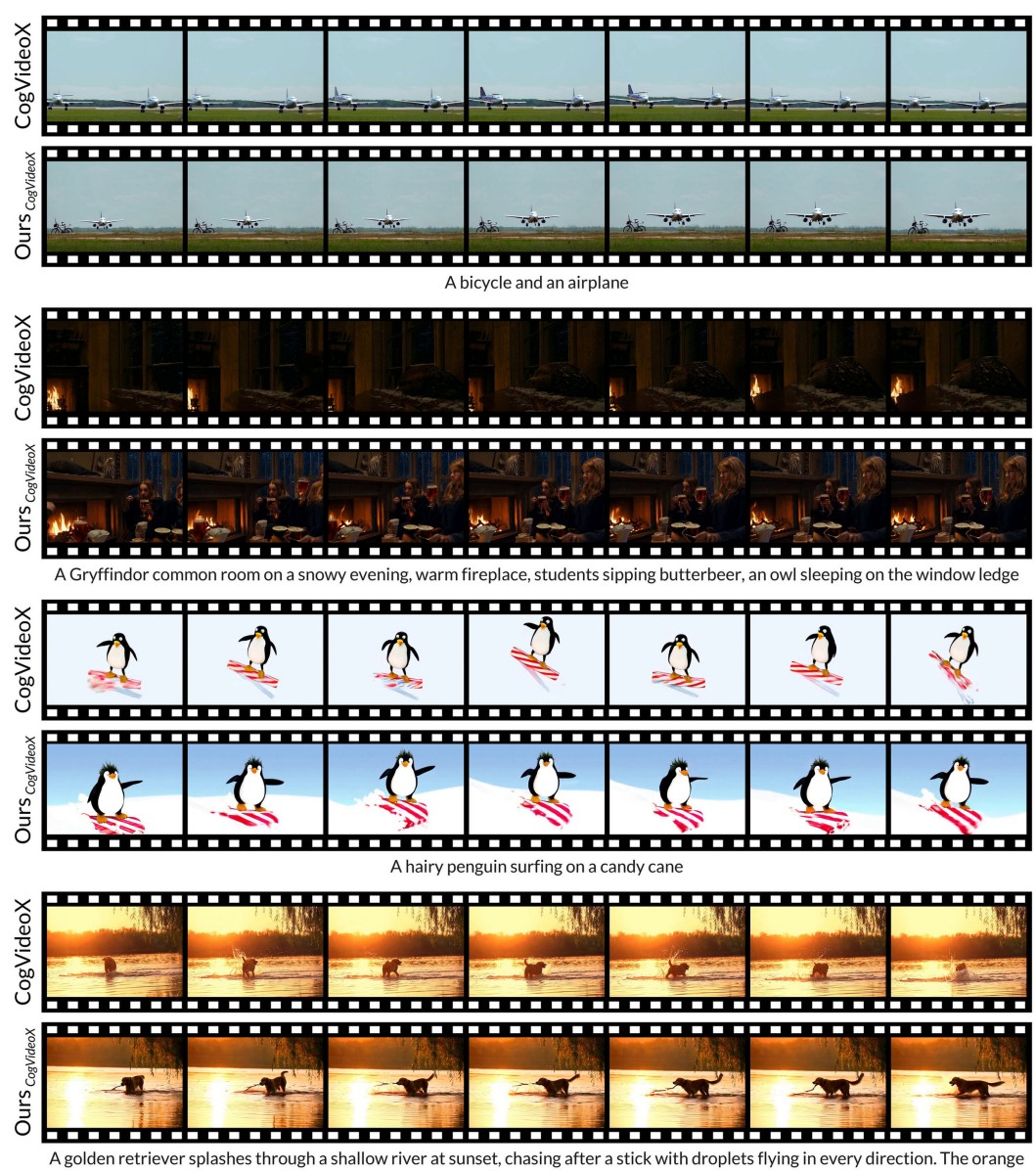

Figure 32: Further Text-to-Video generation comparisons for rare prompts.

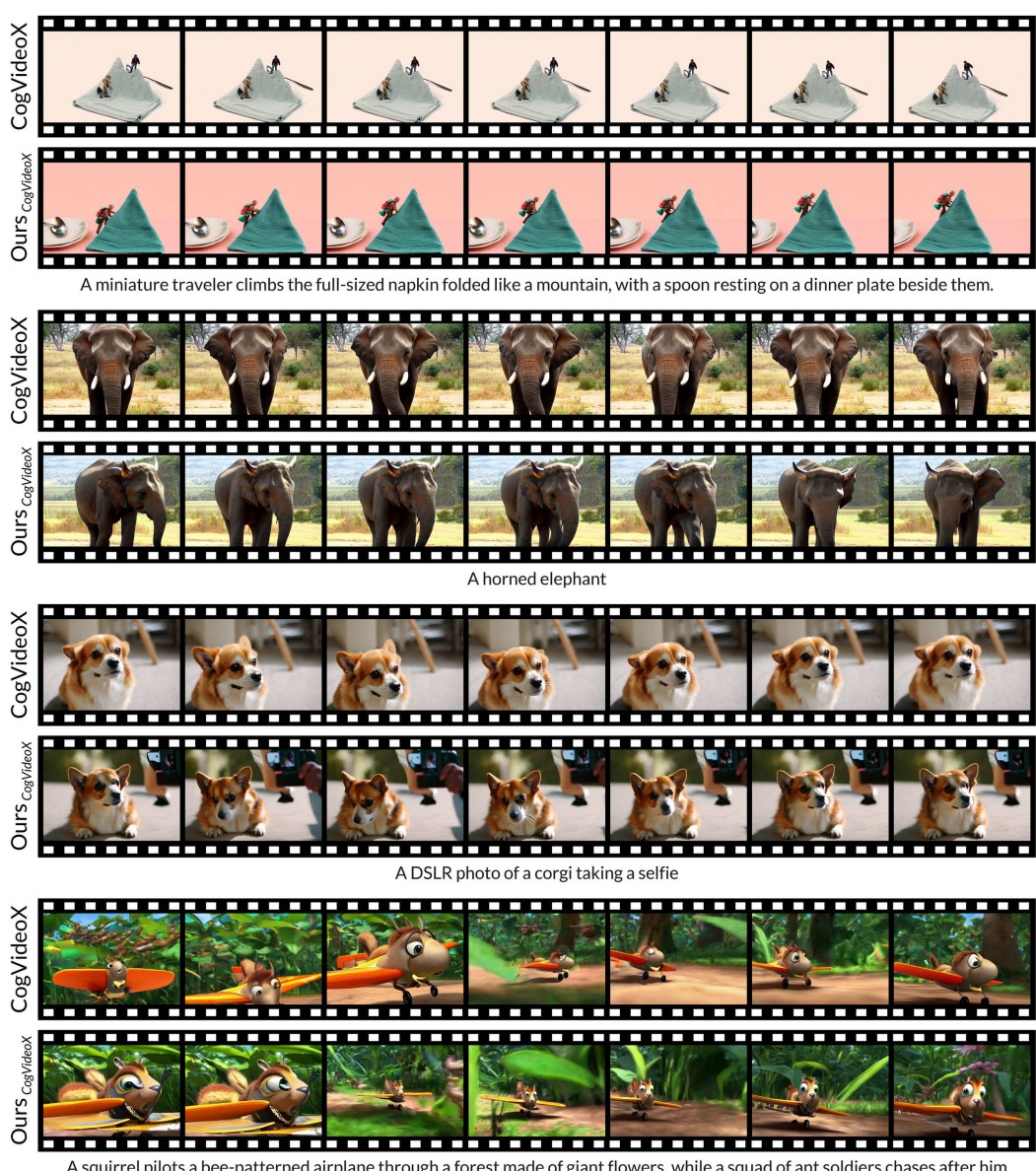

Figure 33: Further Text-to-Video generation comparisons for rare prompts.

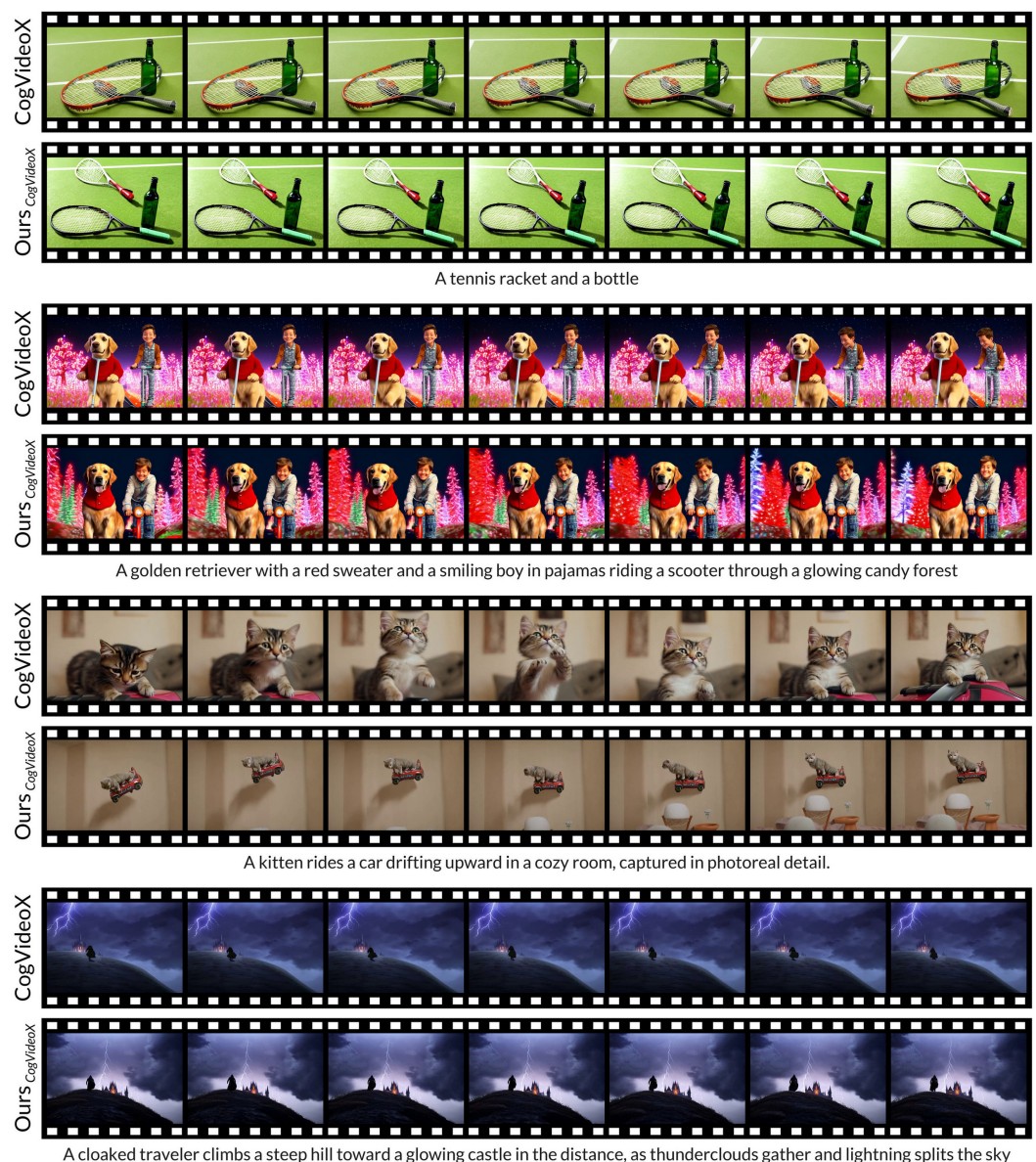

Figure 34: Further Text-to-Video generation comparisons for rare prompts.

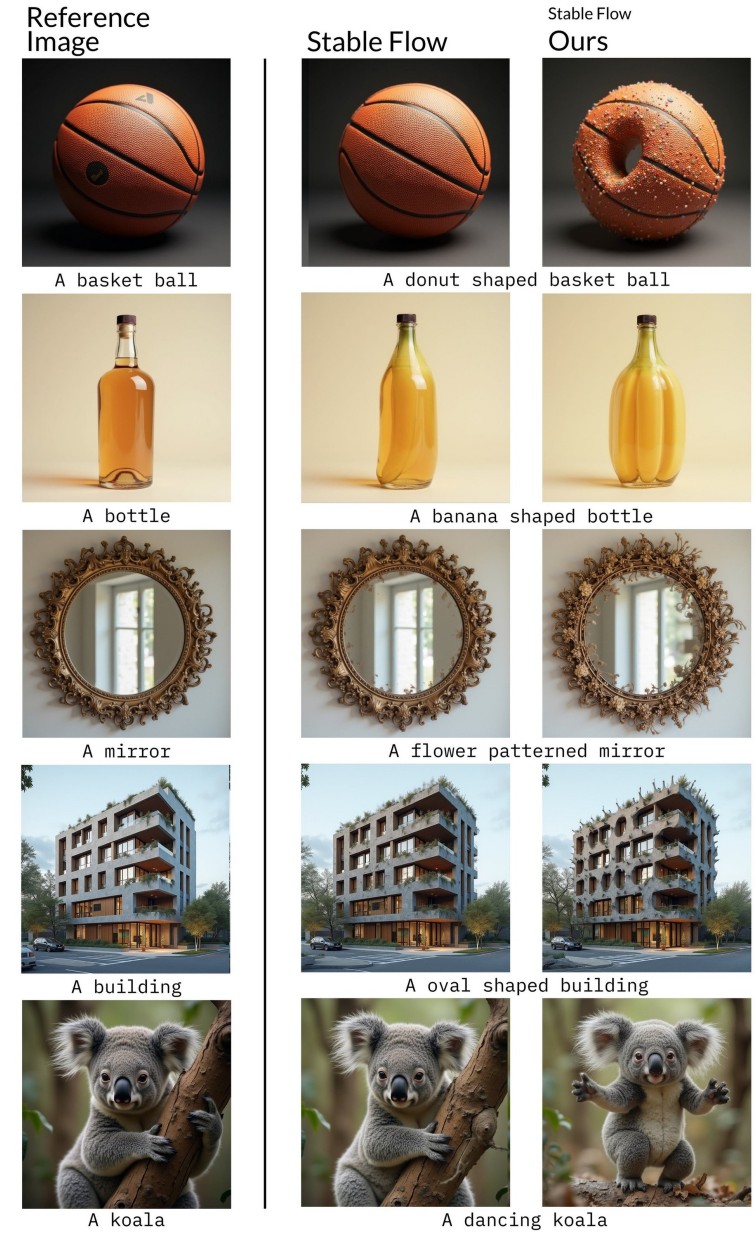

Figure 35: Further Text-Driven Image Editing comparisons for rare prompts.

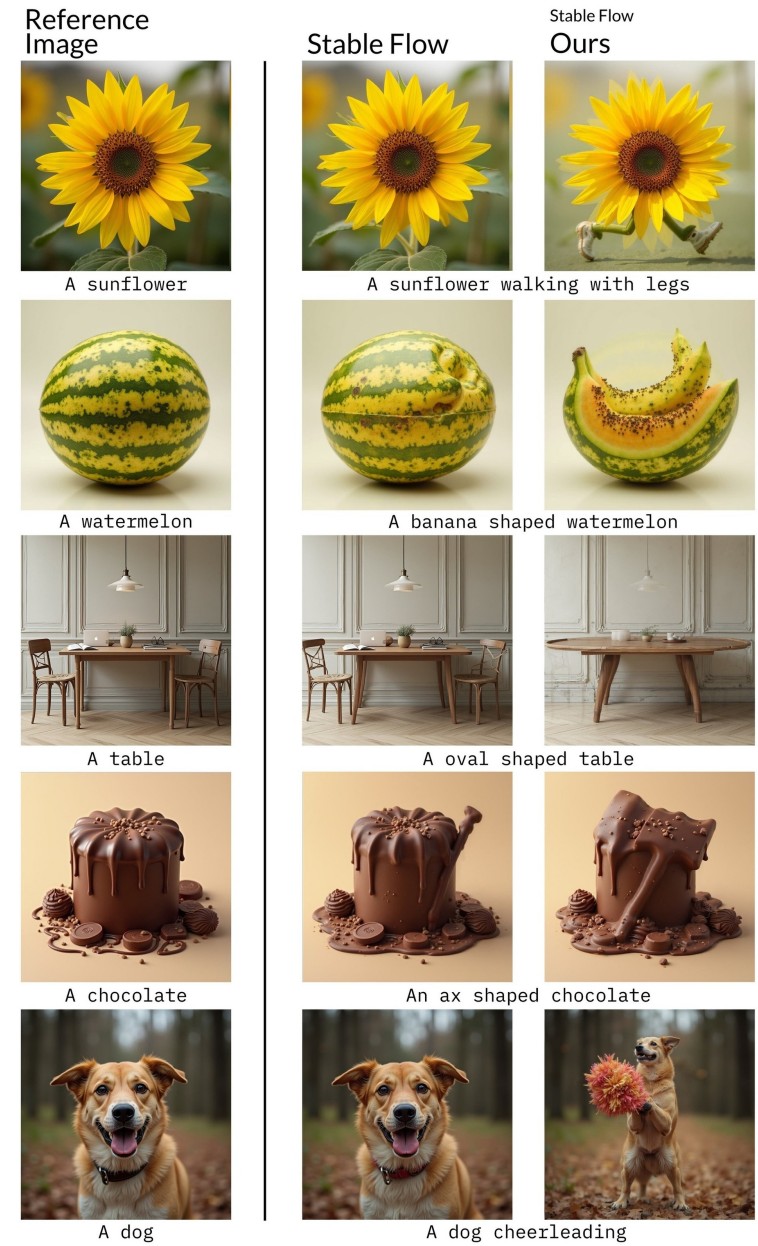

Figure 36: Further Text-Driven Image Editing comparisons for rare prompts.

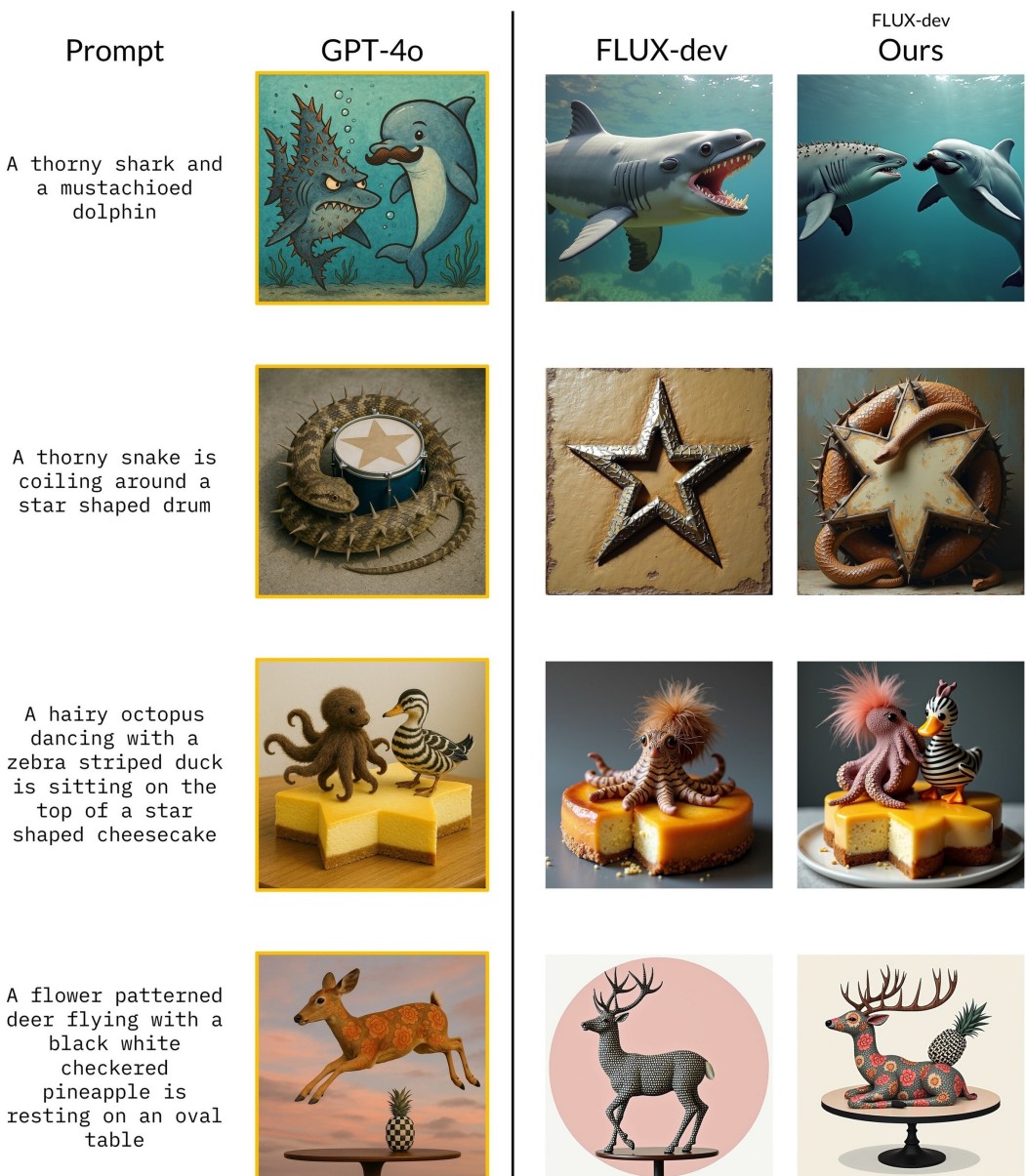

Figure 37: Qualitative comparisons with GPT-4o-generated images.

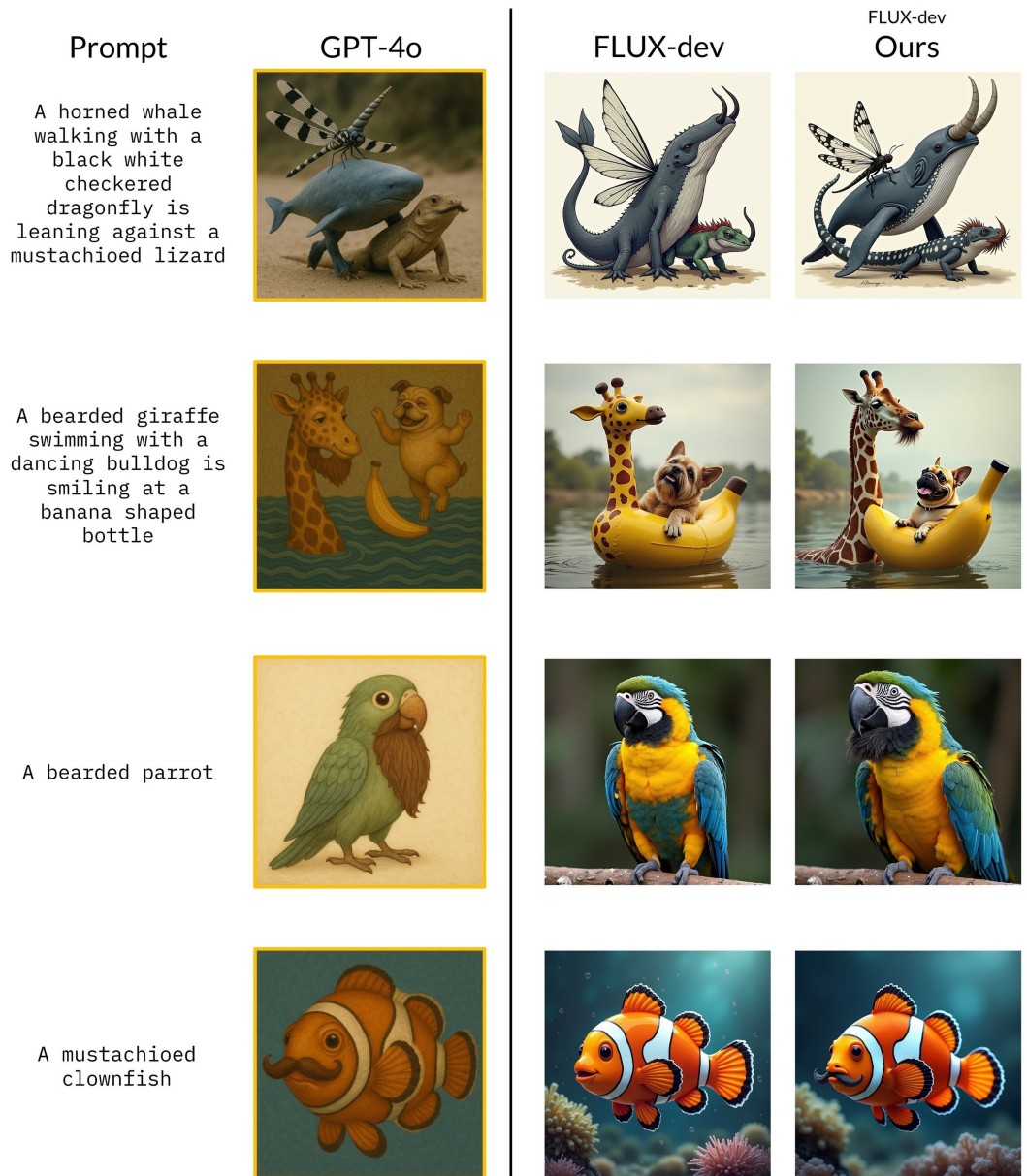

Figure 38: Qualitative comparisons with GPT-4o-generated images.

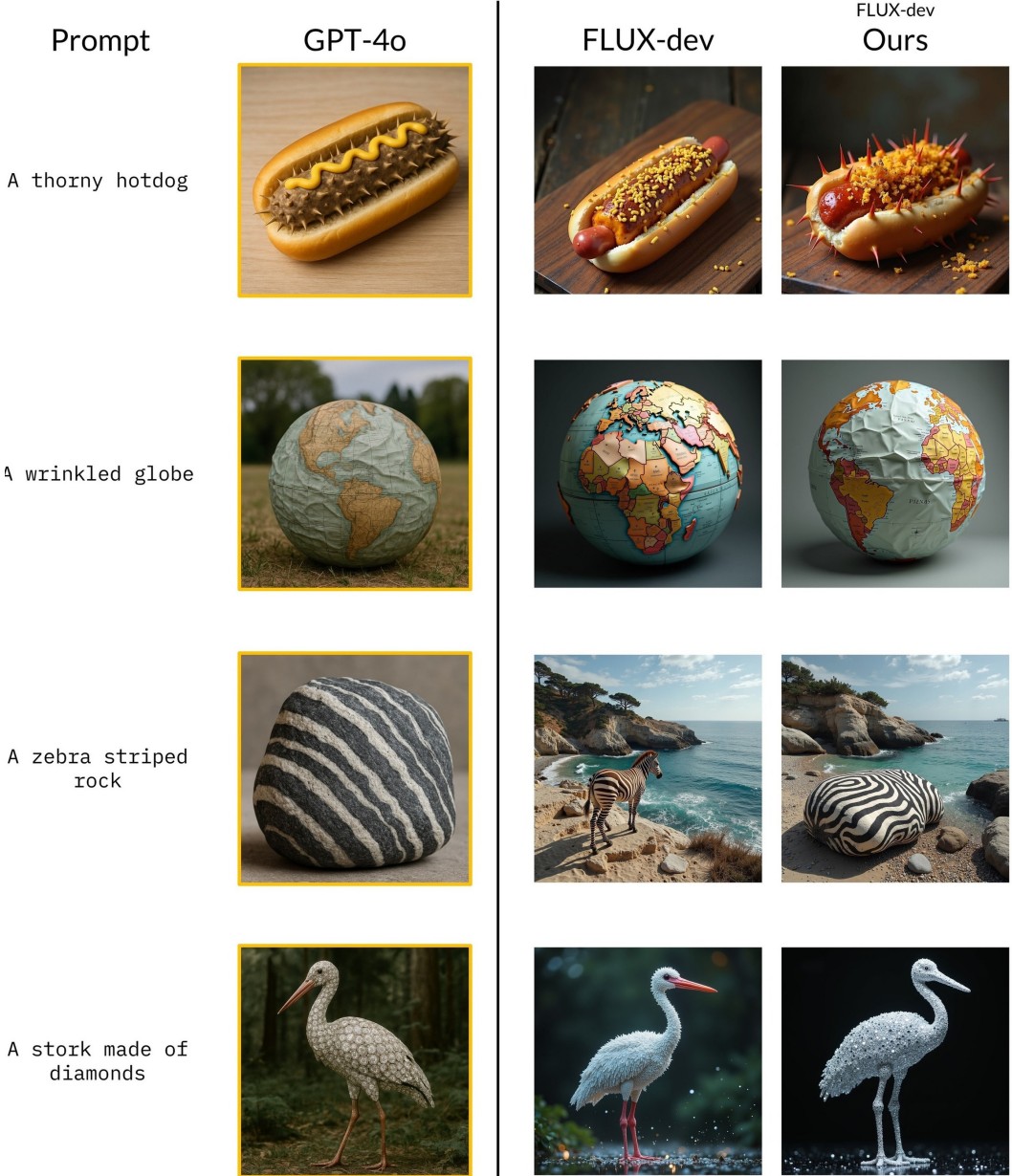

Figure 39: Qualitative comparisons with GPT-4o-generated images.

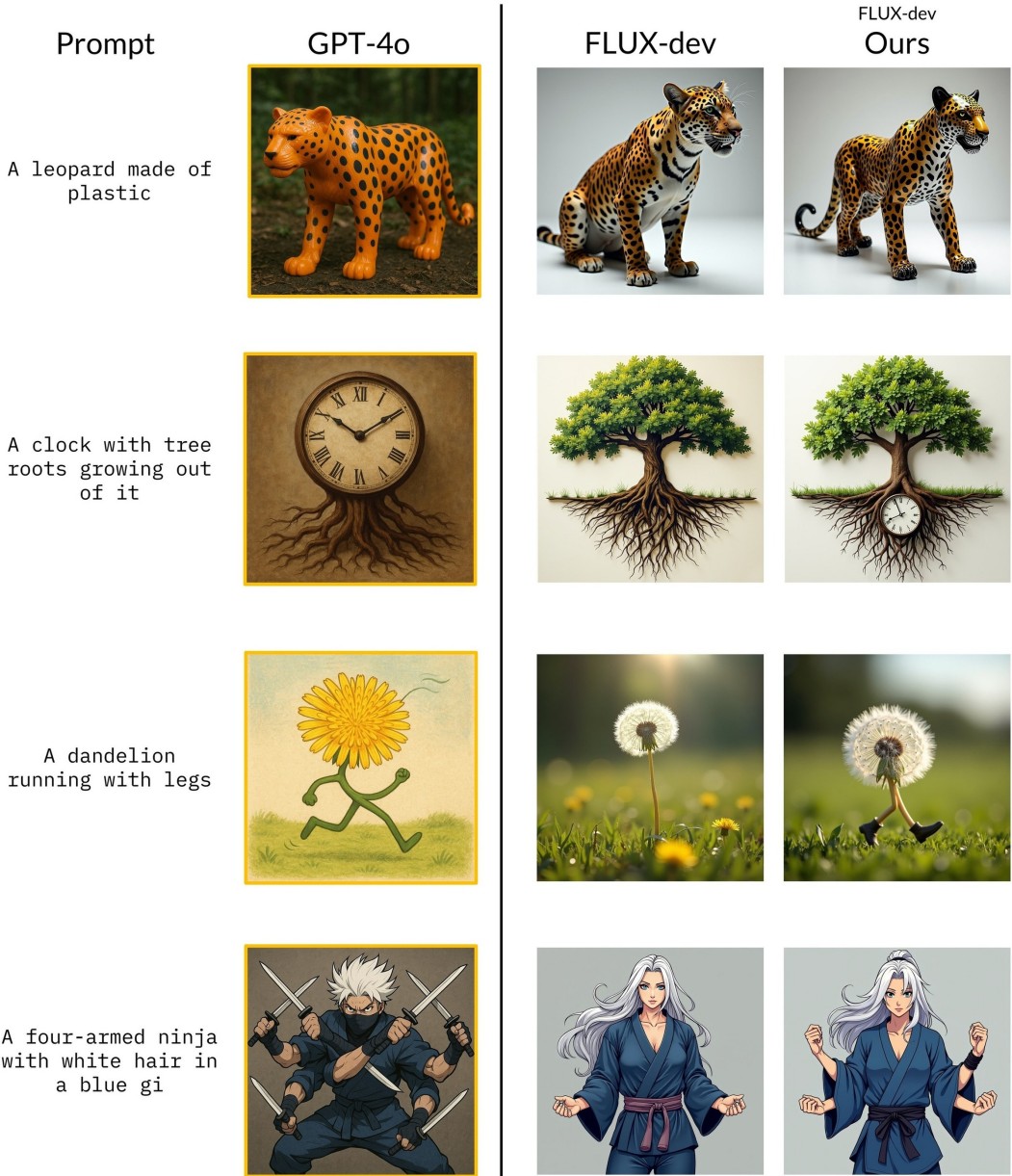

Figure 40: Qualitative comparisons with GPT-4o-generated images.

