# OpenReview forum: "Rare Text Semantics Were Always There in Your Diffusion Transformer"
_NeurIPS.cc/2025/Conference — NeurIPS 2025 poster_

### Official Review · Reviewer_VqR3 · 2025-06-18

**Clarity:** 1
**Significance:** 3
**Originality:** 3
**Rating:** 4
**Confidence:** 3

**Summary:**

This paper proposes ToRA (Token Spacing and Residual Alignment), a training-free method for improving the generation of rare semantic concepts in Multi-modal Diffusion Transformers (MM-DiTs). The core insight is that rare semantics already exist within the text embeddings but are not effectively accessible during generation. The method operates by expanding representational basins around text token embeddings via variance scale-up before joint-attention blocks. TORA consists of two components: (1) Token Spacing, which scales up variance in the principal semantic space using PCA and singular value decomposition, and (2) Residual Alignment, which rotates the residual space toward semantic directions to mitigate potential side effects. The method is evaluated on text-to-image generation (using RareBench, T2I-CompBench, GenEval), text-to-video generation (CogVideoX), and text-driven image editing (Stable Flow).

**Questions:**

**PCA**: When is the PCA computed in practice? Is it pre-computed using a fixed dataset, computed per-batch, or computed dynamically for each input?

**Computational overhead**: What is the actual computational cost of applying TORA during inference? The paper claims "low-overhead" but provides no quantitative analysis.

**Failure mode analysis**: Under what conditions does the method fail or produce worse results? The authors do not expand on the failure modes.

**Ethical Concerns:**

["NO or VERY MINOR ethics concerns only"]

**Final Justification:**

I am grateful to the authors for taking the time to answer my questions, and for performing additional experiments. Their answers have cleared my concerns, besides the structure of the paper, which cannot be updated during the rebuttal because of the NeurIPS policy.

I will raise my score from 3 to 4, but I believe this submission would strongly benefit from a second round of review for its writing. Two of the reviewers agree that writing should be improved, while the other two appreciate the structure. For this reason, I do not intend to raise my score to 5.

**Limitations:**

The discussion of limitations is deferred to the appendix. Naturally, the clarity of the writing is not discussed as a limitation, although it is the strongest limitation in my opinion.

**Paper Formatting Concerns:**

No particular concerns regarding the official formatting instructions, rather some concerns regarding the general writing style and organization chosen by the authors.

**Quality:**

2

**Strengths And Weaknesses:**

## Strengths

**Strong empirical performance across multiple tasks**: The method shows consistent improvements across text-to-image generation (Table 1), text-to-video generation (Figure 8), and text-driven image editing (Table 2). On RareBench, TORA achieves substantial gains over baselines across all categories when evaluated with both GPT-4o and human evaluation.

**Training-free**: Unlike existing methods that require finetuning (Figure 2a), optimization during inference (Figure 2b), or external LLM modules (Figure 2c), TORA requires no additional training.


**Comprehensive evaluation framework**: The evaluation spans multiple benchmarks (RareBench for rare concepts, T2I-CompBench and GenEval for common prompts) and includes both automated metrics and human evaluation, providing robust evidence for the method's effectiveness.

**Good compatibility with existing methods**: The paper demonstrates that TORA can be combined with existing approaches like R2F, showing improved performance in "R2F + Ours" configurations (Table 1), indicating practical compatibility.

## Weaknesses
**Clarity**: the manuscript is sometimes vague and misses details. For example:
* In Section 3.1, the authors are not specific enough regarding when PCA is computed, whether it is precomputed using many samples and fixed afterwards, or whether it is computed dynamically for each new batch. Additionally, it is not clear whether the PCA decomposition is shared between transformers blocks, although I assume a different PCA is computed for each layer.
* The writing style is non-standard and difficult to follow on a first read. The paper would benefit from a clearer separation between background, methodology, and experiments. For example, Section 2 mixes background information with experimental results, making it hard to distinguish prior work from novel contributions. This lack of structure is especially challenging for readers unfamiliar with interventions on the neural network activations, such as those proposed in this manuscript.
* Another area lacking clarity is the discussion of anisotropy and isotropy in the introduction (third paragraph). The explanation of these concepts is vague, particularly regarding why, after clustering, the cosine similarity within clusters is low while the global average similarity is high. This is counterintuitive, as clustering based on cosine similarity should group similar tokens together. The authors should provide a clearer explanation for this phenomenon in the background.

**Baseline comparisons**: The paper primarily compares against base models and R2F. It would be relevant to compare ToRA with methods that fine-tune the diffusion transformer or perform inference optimizations. Even if the method does not beat those baselines, it would be important for practitionners to understand how large the gap would be.

**Figure quality issues**: Figure 2 contains elements that are not clearly explained. Specifically, it is not clear what the "rotating arrow" in Figure 2b represents.



Overall, the empirical results seem strong, however the background section deserves a major rewrite in my opinion.

---

> ### Author Rebuttal · Authors · 2025-07-31
>
> We sincerely thank you for your thoughtful and constructive feedback. Below, we address each of your comments in turn. If you need to check the technical appendix, it can be found in the submitted supplementary materials.
>
> ---
> ### **[W1-1 & Q1] Further detail is required for the implementation details of PCA**
>
> The reviewer questioned implementation details of PCA. PCA is recomputed at each diffusion timestep and each block, we take the current token‑embedding matrix $e_b$ of joint‑attention block $b$, perform SVD, and apply Token Spacing / Residual Alignment before attention is executed. Because nothing is pre‑computed over an external corpus, the decomposition only reflects the block‑ and timestep‑specific embeddings.  Consequently, each transformer block receives its own PCA decomposition, and no basis is shared between blocks.
>
> We will revise our paper as *"We perform PCA **separately** for every block at every timestep; the resulting bases are **not shared** across blocks or timesteps."* Together, we will insert the following sentence after `line 174`.
> > *"At each timestep and each block, we perform PCA on the $e^b$ of each joint‑attention block $b$ before applying Token Spacing; this intervention is uniformly applied across all blocks and timesteps, and we denote $e^b$ simply as e for brevity."*
>
> ---
> ### **[W1-2] Concerns about the paper's readability for the first reader.**
>
> The reviewer raises a concern about the paper's readability on a first pass. Unfortunately, due to the NeurIPS 2025 rebuttal policy, re-uploading a revised version is not permitted. Therefore, we will illustrate the structure of the updated manuscripts with examples.
>
> In the current manuscript, `Sec. 2` presents our analysis, while `Sec. 3` introduces the method derived from this analysis. In the revised version, we will adopt the structure you suggested by:
>
> - introducing a Sec. 2: Background,
> - moving our empirical findings to Sec. 3: Analyses, and
> - presenting the technical approach in Sec. 4: Methodology.
>
> We will also adjust the section titles to the more intuitive *Background*, *Analyses*, and *Method* so that readers can immediately distinguish prior work from our novel contributions, thereby improving readability for those unfamiliar with techniques similar to ours.
>
> ---
> ###  **[W1-3] Lack of clarified explanation for the concepts of anisotropy and isotropy.**
>
> We fully understand the source of confusion. As we thought, the reviewer's counter‑intuitive impression arises because two seemingly opposing phenomena coexist in transformer text‑embedding spaces and the peculiar property of contextual embedding spaces [1,2,3,4]:
>
> - **Global Anisotropy.** Contextual embeddings tend to collapse into a narrow hyper‑cone, a property reported by Cai et al. (2021) [1]. This geometry yields high cosine similarity between almost any pair of vectors, even when they are semantically unrelated.
> - **Local Isotropy.** To measure local isotropy, clustering methods are generally based on calculating a specific distance metric (typically Euclidean distance for K-means or Mahalanobis distance for Gaussian Mixture Models), rather than cosine similarity.
> Also, in each cluster, the cluster‑wise mean is subtracted. This centering step removes the dominant global direction. Within each cluster, the directions diversify, so pair‑wise cosines drop toward a lower value. This procedure is a conceptual overview of `Equation 1` in our main manuscript.
>
> It is precisely because our clustering relies on distance metric which capture absolute proximity, **not angular alignment**. In short, the embedding space resembles a single hyper‑cone at the global level, yet becomes isotropic once examined from the “center of each cluster”. It is also mentioned in `line: 123` of our manuscript.
>
> For your better understanding, we use a celestial analogy to explain this concept. Global anisotropy is analogous to the cosmic web, where these galaxies align along the same filaments share a common axis. This alignment causes even semantically unrelated vectors (stars) in the embedding space (galaxies) to have uniformly high cosine similarity.
>
> Within any single galaxy, however, the situation changes. If we treat the solar system as one cluster and set the Earth at its centroid, the planets around us appear in every direction. This is our local isotropy.
>
> Lastly, of course, we will revise the manuscript to make this distinction clear.
>
> *[1] Cai, Xingyu, et al. "Isotropy in the contextual embedding space: Clusters and manifolds." ICLR 2021*
>
> *[2] William Rudman and Carsten Eickhoff. Stable anisotropic regularization. ICLR 2024*
>
> *[3] Kawin Ethayarajh. How contextual are contextualized word representations? comparing the geometry of bert, elmo, and gpt-2 embeddings. EMNLP 2019*
>
> *[4] Sara Rajaee and Mohammad Taher Pilehvar. A cluster-based approach for improving isotropy in contextual embedding space. ACL 2021*
>
> ---
>
> ### **[W2] Requiring baseline comparison with fine-tuned and optimization methods**
>
> Because RareBench is designed around prompts the model has barely seen during pre-training in the open-sourced MM-DiT models, each prompt is, by definition, *rare*. Finetuning on these prompts would defeat the justification of the benchmark, and the absence of *ground‑truth images* makes supervised fine‑tuning infeasible. If there is a fair evaluation protocol, we will gladly adopt it.
>
> In contrast, benchmarks such as GenEval and T2I‑CompBench contain more conventional prompts, allowing direct comparisons with finetuned or inference‑optimized models. The table below reports ToRA’s performance alongside these state‑of‑the‑art baselines.
>
> Finally, we emphasize that **ToRA can be applied to any MM-DiT‑based model, including all finetuned or inference‑optimized variants**, highlighting a key contribution of our work.
>
> - *GenEval*
>
> |Models|Single Obj.|Two Obj.|Count|Colors|Position|Attr. Bindings|
> |--|--|--|--|--|--|--|
> |**MM-DiT w/ Optimization**|
> |$\text{CaPO}_\text{SD3.0}$ [1]|0.99|0.87|0.63|0.86|0.31|0.59|
> |$\text{RankDPO}_\text{SD3.0}$ [2]|1.00|0.90|0.72|0.87|0.31|0.66|
> |$\text{GRAT}_\text{FLUX-dev}$ [3]|0.98|0.84|0.48|0.75|0.26|0.37|
> |**Ours (ToRA)**|
> |$\text{Ours}_\text{SD3.0}$|0.98|0.77|0.66|0.70|0.50|0.61|
> |$\text{Ours}_\text{FLUX-schnell}$|0.99|0.84|0.77|0.85|0.47|0.55|
>
> - *T2ICompbench*
>
> |Models|Color|Shape|Texture|Spatial|Non-Spaital|Complex|
> |--|--|--|--|--|--|--|
> |**MM-DiT w/ Finetuning**|
> |$\text{CoT-Diff}_\text{FLUX-schnell}$ [4]|78.1|61.1|69.1|55.8|31.3|50.0|
> |**MM-DiT w/ Optimization**|
> |$\text{CaPO}_\text{SD3.0}$ [1]|78.8|57.2|73.1|23.0|31.3|50.9|
> |$\text{RankDPO}_\text{SD3.0}$ [2]|83.3|63.5|78.7|36.5|31.3|48.7|
> |$\text{RENO}_\text{FLUX-schnell}$ [5]|80.0|60.0|75.0|-|-|-|
> |**Ours (ToRA)**|
> |$\text{Ours}_\text{SD3.0}$|84.5|64.2|82.0|45.5|33.8|48.1|
> |$\text{Ours}_\text{FLUX-schnell}$|84.6|65.7|81.2|46.0|35.8|47.7|
>
> *[1] Lee, Kyungmin, et al. "Calibrated multi-preference optimization for aligning diffusion models.", CVPR 2025.*
>
> *[2] Karthik, Shyamgopal, et al. "Scalable ranked preference optimization for text-to-image generation." arXiv 2024.*
>
> *[3] Ren, Sucheng, et al. "Grouping First, Attending Smartly: Training-Free Acceleration for Diffusion Transformers." arXiv 2025.*
>
> *[4] Liu, Zheyuan, et al. "CoT-lized Diffusion: Let's Reinforce T2I Generation Step-by-step." arXiv:2507.04451 (2025).*
>
> *[5] Eyring, Luca, et al. "Reno: Enhancing one-step text-to-image models through reward-based noise optimization." NeurIPS 2024*
>
> ---
> ### **[W3] Unclear figure quality (e.g., `Fig. 2b`)**
>
> In `Fig. 2b`, the *rotating arrow* was meant only as a simple way to show the latent‑vector update loop at each timestep. As the icon might be unclear to readers, we’ll replace it with the label *''Iterative update over timesteps $t = 1 \dots T$''* and also add this to the caption for clarity. We are happy to incorporate any further revisions you may suggest.
>
> ---
> ### **[Q2] Questions on claimed low overhead.**
>
> The reviewer pointed out the absence of a detailed computational cost analysis. As shown in `Appendix D.1` and `Table 4`, we provide a comprehensive comparison. For convenience, herein, we imported the key results comparing our method against the baseline (SD 3.0) and the R2F method below:
>
> |Models|Peak Memory (GB)|GPU Time (sec)|
> |--|--|--|
> |SD3.0|$22.02$|$21.30 \pm 1.1$|
> |R2F|$38.49$|$78.13 \pm 2.36$|
> |Ours|$22.17$|$44.74 \pm 1.9$|
>
> As this results illustrates:
> - Memory: TORA's peak vRAM usage is only marginally higher than the baseline, representing an increase of less than 0.1% ($\sim×1.007$).
> - Inference Time: While our method does increase GPU time relative to the baseline, it is substantially more efficient than R2F ($44.74\ sec$ vs. $78.13\ sec$).
>
> To improve visibility of these results, we will also summarize the relevant analysis in the main text. We believe our validation is thorough, but we welcome any suggestions for further constructive experiments.
>
> ---
> ### **[Q3] Necessity of failure mode analysis**
>
> We appreciate the reviewer’s comment for the detailed failure mode analysis. Due to policy restrictions on external links, we offer a descriptive account of observed failure cases instead:
>
> > **Case 1:** As noted in our limitations in `Sec. G`, our method can struggle with long prompts where an entity is mentioned multiple times (e.g., with a pronoun). This issue can be exacerbated as our method's increased variance in the representation space may weaken the semantic link between these words.
>
> > **Case 2:** As discussed in `Sec. 2.5`, scaling up variance to better distinguish between tokens can occasionally misalign embeddings from their intended semantic direction. Our Residual Alignment strategy significantly mitigates this (`Fig. 13`), but cannot entirely eliminate it.
>
> We will add a visualization of these cases to the appendix in the revised version to further clarify these points.

---

> > ### Comment · Reviewer_VqR3 · 2025-08-04
> >
> > I am grateful to the authors for taking the time to answer my questions, and for performing additional experiments. Their answers have cleared my concerns, besides the structure of the paper, which cannot be updated during the rebuttal because of the NeurIPS policy.
> >
> > I will raise my score from 3 to 4, but I believe this submission would strongly benefit from a second round of review for its writing. Two of the reviewers agree that writing should be improved, while the other two appreciate the structure. For this reason, I do not intend to raise my score to 5.

---

> ### Author Response · Authors · 2025-08-08
>
> As the discussion period nears its end, we thank you for your thoughtful engagement in what may be our final response at this stage.
>
> We are pleased that our answers and additional experiments addressed your main concerns regarding the PCA procedure, computational overhead, and potential failure modes.
> \
> (1) For the computational overhead point, we revisited content already present in the manuscript and presented it more prominently, so that the relevant comparisons are easier to locate and interpret.
> \
> (2) On anisotropy and isotropy, we expanded on our original explanation by adding an analogy to help convey the distinction in a more intuitive way.
> \
> (3) We additionally prepared more detailed description of how PCA is applied and outlined section-level adjustments so that readers of a future version can follow the method more easily and intuitively.
>
> Lastly, we appreciate your acknowledgment of our contributions and the score increase. We also value your suggestion on restructuring sections for smoother navigation by readers unfamiliar with similar techniques. Thank you again for the time and thoughtful feedback you dedicated to reviewing our work.

---

### Official Review · Reviewer_41Hj · 2025-07-01

**Clarity:** 3
**Significance:** 3
**Originality:** 3
**Rating:** 5
**Confidence:** 3

**Summary:**

This paper investigates the generation of rare prompts in multi-modal diffusion transformers. Without requiring additional training or external modules, this paper finds that rare semantics can merge from MM-DiT by proposing a variance scale-up method. Experimental results show that the proposed method achieves strong performance in various tasks, from text-to-image generation to image editing.

**Questions:**

Please refer to the strength and weaknesses.

**Ethical Concerns:**

["NO or VERY MINOR ethics concerns only"]

**Final Justification:**

I have carefully reviewed the paper and read the comments from other reviewers.  As also pointed by Reviewer bpoM and Reviewer VqR3,  some important clarifications and implementation details are missing in the original submission. I appreciate the authors’ efforts during the rebuttal to provide additional explanations, which have solved my concerns. I am glad to raise my rating to 5.

**Limitations:**

This paper discusses the limitations but does not address the broader impact.

**Quality:**

3

**Strengths And Weaknesses:**

**Strength:**
1. The paper presents an insightful analysis on the isotropic properties of rare semantics.
2. Without requiring additional training, the proposed ToRA method significantly improves text-image alignment for rare prompts.
3. This work conducts extensive experiments to support its claims and performance improvements.
3. The paper is clearly written and well presented.

**Weaknesses:**
1. The proposed ToRA method performs well on rare prompts. How does it perform on common prompts?
2. Do common and rare prompts require different parameters or settings? If so, in real applications, how does the system determine whether a given prompt is rare or common?
3. Equation 2 shows that ToRA scales the top-k singular values. Is k determined by the entire Rarebench dataset, or set differently for different prompts? How does k influence the performance using different groups of prompts, similar to Figure 10(a)?

---

> ### Author Rebuttal · Authors · 2025-07-31
>
> We sincerely thank you for your thoughtful and constructive feedback. Below, we address each of your comments in turn. If you need to check the technical appendix, it can be found in the submitted supplementary materials.
>
>
> ---
>
> ### **[W1&W2] Performance of ToRA on common prompts and prompt-type sensitivity**
>
> In `Table 3`, we report ToRA’s performance on two benchmarks dominated by common prompts; GenEval and T2I‑CompBench and `Fig. 19 & 20` provide the corresponding qualitative examples. Across both datasets, ToRA matches or exceeds every baseline, showing that its advantages are not confined to rare‑prompt scenarios.
>
> $\sigma$ is the only hyperparameter in our methods, and we keep it fixed at $1.3$ for all experiments, rare and common alike. Since the same setting works well across the board, a deployed system need not decide whether a prompt is *rare* or *common*; it simply applies ToRA with the default $\sigma$. Although the method is particularly effective on truly rare prompts, it still improves common benchmarks by sharpening token semantics and reducing errors such as missing objects and attribute mismatches, as seen in `Table 3`.
>
> To further strengthen our evaluation, we have benchmarked our method on an additional benchmark, named HEIM [1], with the results presented in the table below.
>
> | Model | Alignment | Quality | Aesthetics | Reasoning | Knowledge | Art Styles |
> |:---|:---:|:---:|:---:|:---:|:---:|:---:|
> | SD3.0 | 0.81 | 0.65 | 0.63 | 0.53 | 0.72 | 0.48 |
> | SD3.0 + Ours | 0.90 | 0.65 | 0.65 | 0.56 | 0.80 | 0.48 |
>
> These results corroborate a central claim of our paper: our method demonstrates pronounced strengths in categories such as *alignment*, *reasoning*, and *knowledge*. These are precisely the categories that demand a deep semantic understanding of the input prompt. A more detailed discussion of these findings will be incorporated into the appendix. We are grateful to the reviewer for this insightful suggestion, which we believe has materially improved the paper.
>
>
> *[1] Lee, Tony, et al. "Holistic evaluation of text-to-image models." Advances in Neural Information Processing Systems 36 (2023): 69981-70011.*
>
> ---
> ### **[W3] Is k determined by the dataset? How does k influence the performance using different groups of prompts?**
> To begin with, we do not set the $k$ manually, nor do we assign a benchmark‑specific constant. Instead, $k$ is computed automatically for every prompt, at every diffusion timestep, and in every transformer block.
> As described in `lines 178–181` of our main text, we determine $k$ adaptively for each individual prompt and layer, using the Maximum Distance to Chord (MDC) method [2]. This approach identifies an *elbow point* in the singular value spectrum, which varies significantly across different prompts due to differences in length, composition, and semantic diversity. Thus, $k$ is not fixed globally across the Rarebench dataset, but instead dynamically computed per sample to better capture the effective dimensionality of the text embeddings in that context.
>
> To address the reviewer’s potential questions, we conducted an extensive fixed‑$k$ ablation.
>
> - *Fixed $k$ ablation studies for text-to-image (Rarebench)*
>
> | $k$ | SD3.0 | FLUX-schnell | FLUX-dev |
> |---|---|---|---|
> | **ours** | **76.7** | **76.0** | **75.3** |
> | 10 | 73.2 | 73.3 | 70.0 |
> | 20 | 74.0 | 73.8 | 71.2 |
> | 30 | 74.8 | 74.6 | 73.0 |
> | 40 | 75.8 | 75.1 | 74.2 |
> | 50 | 74.1 | 72.5 | 70.8 |
> | 60 | 65.2 | 70.3 | 65.2 |
>
> - *Fixed $k$ ablation studies for text-to-video (Rarebench)*
>
> | $k$ | CogVideo-5B |
> |---|---|
> | **ours** | **66.6** |
> | 10 | 63.5 |
> | 20 | 65.3 |
> | 30 | 64.7 |
> | 40 | 60.0 |
> | 50 | 55.7 |
> | 60 | 51.8 |
>
> We tested fixed $k$ values ranging from 10 to 60 in increments of 10. By checking the values in the tables above, we can see that the automatically selected $k$ in ToRA outperforms any fixed $k$.
> Furthermore, we found that using ToRA yields higher scores than not using it in most cases, regardless of the value of
>  we chose. (e.g., the original RareBench average for SD 3.0 is 60.9). For additional insight into the typical scale of dimensionality, we computed the average MDC-derived $k$ for each task, resulting in approximately ~42.0 for Text-to-Image (T2I) and ~23.0 for Text-to-Video (T2V).
>
> To confirm the robustness of our approach, we varied the automatically selected $k$ by −15 \% to +15 \% and assessed performance (see the table below). The results demonstrate the effectiveness of our MDC‑based method.
>
> - *Adaptive $k$ ablation studies for text-to-image (Rarebench)*
>
> | Adaptive $k$ | SD3.0 | FLUX-schnell | FLUX-dev |
> |---|---|---|---|
> | -15% | 72.9 | 73.8 | 73.1 |
> | -10% | 73.4 | 74.1 | 73.7 |
> | -5% | 75.2 | 75.4 | 74.4 |
> | **ours** | **76.7** | **76.0** | **75.3** |
> | +5% | 72.5 | 73.3 | 69.3 |
> | +10% | 69.8 | 70.9 | 65.0 |
> | +15% | 66.1 | 69.5 | 62.2 |
>
> Finally, although we employ exactly the same procedure irrespective of dataset or benchmark, we investigated whether the choice of  $k$ affects performance across benchmarks by testing on T2I‑CompBench, which features a larger share of common‑language prompts.
>
> - *Adaptive $k$ ablation studies for text-to-image (T2ICompbench)*
>
> | $k$ | SD3.0 | FLUX-schnell | FLUX-dev |
> |---|---|---|---|
> | ours | 85.4 | 84.5 | 84.1 |
> | 10 | 84.0 | 83.1 | 82.9 |
> | 20 | 84.5 | 83.8 | 83.5 |
> | 30 | 85.1 | 84.0 | 84.0 |
> | 40 | 84.2 | 83.0 | 83.5 |
> | 50 | 83.9 | 82.8 | 83.2 |
> | 60 | 83.1 | 82.5 | 80.8 |
>
> As the table above shows, our method again achieves the best results: just like on RareBench, the ToRA variant outperforms nearly every fixed‑$k$ setting, independent of the $k$ values chosen (note, for instance, that SD 3.0’s baseline T2I‑CompBench average is 56.8). For context, the average values of MDC‑derived $k$ on T2I‑CompBench are 37.5 for T2I and 21.0 for T2V.
> We will add these findings to the Appendix.
>
> *[2] David H. Douglas and Thomas K. Peucker. Algorithms for the reduction of the number of points required to represent a digitized line or its caricature. Cartographica: The International Journal for Geographic Information and Geovisualization, 10:112–122, 1973.*
>
> ---
>
> ### **[L] This paper discusses the limitations but does not address the broader impact.**
>
> We appreciate the reviewer's point regarding broader impact. In the revised version, we'll include a discussion on this. We believe our method enhances semantic robustness and mitigates mode collapse in rare-concept generation, thereby improving the inclusivity and reliability of generative models for real-world applications.
>
> Specifically, by preserving the semantic distinctiveness of less common concepts, even unseen ones, our approach enables the model to generalize more faithfully to novel prompts. Its adaptive nature may also reduce overfitting to dominant concepts, contributing to fairness across diverse prompt types. However, like all generative models, ours carries the potential for misuse (e.g., generating disinformation from rare semantics) depending on user intent.

---

> > ### Comment · Reviewer_41Hj · 2025-08-02
> >
> > Thank you to the authors for their rebuttal, which addressed my concern. I have also reviewed the other reviewers’ comments. While I support the manuscript’s contributions, I agree with other reviewers that some details are not clear in paper, such as the PCA method and discussions on K.  Improved paper writing with these details and discussions is recommended.

---

> > > ### Author Response · Authors · 2025-08-02
> > >
> > > Thank you for recognizing our contributions and for noting that our rebuttal addressed your concern. We agree with the reviewers’ valuable suggestions and have updated the manuscript accordingly. In accordance with the NeurIPS 2025 rebuttal policy, we cannot upload the revised PDF at this stage; however, in this console, we will provide detailed changes in point to address each concern as clearly as possible, and we hope they are helpful.
> > >
> > > - **PCA method**:
> > >     - We inserted the sentence after `line 171` as, **"Note that, for a single prompt we perform PCA independently for each block at each timestep, with no sharing across blocks or timesteps."**
> > >     - We will concretize the `line 174` as follows: **"At each timestep and each joint-attention block $b$, we perform PCA before applying joint-attention on the $e^b$, which is a set of text token vectors. This procedure continues sequentially in every block and timestep."**
> > >
> > > - **Discussion on $k$**:
> > >     - Although we initially stated in `lines 178-181` that we set $k$ based on MDC, we later realized that further ablation studies were possible. As we demonstrated in the rebuttal, we have now added the results and a discussion of this to `Appendix Section F.1`, as follows: **"Our method requires specifying $k$ to partition the representation into a Principal Space and a Residual Space. Although we adopt an MDC-based method to select $k$ and reduce manual tuning, we assess the effectiveness of this choice through an ablation study over $k$. [...] Furthermore, we found that using MDC-based dynamic $k$ selection yields higher scores than not using it in most cases, regardless of the value of $k$ we chose."**
> > >
> > > We sincerely thank the reviewers for the constructive feedback, which has enabled us to further improve the overall quality of the manuscript. We will reflect these updates clearly in the revision. Furthermore, the source code provided in our supplementary materials (to be made public in the future) will help readers more clearly reimplement, reproduce, and follow the methodological steps from our manuscript.

---

> ### Comment · Reviewer_41Hj · 2025-08-05
>
> Thank you to the authors for the response.  I understand that the paper cannot be updated due to the NeurIPS policy. I am glad to raise my score to 5.

---

> ### Author Response · Authors · 2025-08-08
>
> As the discussion period nears its end, we thank you for your thoughtful engagement in what may be our final response at this stage.
>
> We are glad that we could address your main concern regarding ToRA’s performance on common prompts, as well as the role of the adaptive selection of $k$.
> \
> (1) While $k$ in our method is never set manually, your request for an ablation study motivated us to conduct extensive fixed-$k$ and variation experiments, which we believe provide a more complete view of our approach’s behavior.
> \
> (2) On the PCA procedure, we built on our original description by adding step-by-step detail so that readers can follow the process more directly. In the rebuttal, we also illustrated concrete examples of how these additions would appear in a future version of the manuscript, so the intended improvements are transparent.
>
> We appreciate your acknowledgment of our contributions and your suggestion to expand these discussions. Thank you again for the time and thoughtful feedback you dedicated to reviewing our work.

---

### Official Review · Reviewer_bpoM · 2025-07-03

**Clarity:** 3
**Significance:** 3
**Originality:** 3
**Rating:** 4
**Confidence:** 3

**Summary:**

This paper proposes a simple method to help Multi-modal Diffusion Transformers better handle rare or unusual prompts in text-to-vision tasks. By adjusting the text embeddings before attention using the proposed principle of variance scale up, the model can bring out rare concepts more clearly, without needing extra data, training, or tools.

**Questions:**

see Strengths And Weaknesses.

**Ethical Concerns:**

["NO or VERY MINOR ethics concerns only"]

**Final Justification:**

The authors’ responses address the technical details. I would like to maintain my initial scores.

**Quality:**

3

**Strengths And Weaknesses:**

# Strengths

1. The paper is overall well written and clearly structured.
2. The proposed method operates without requiring external knowledge, enhancing its general applicability across tasks.
3. The method demonstrates superior performance compared to baseline approaches in the reported experiments.

# Weaknesses

These are not necessarily weaknesses, but addressing them may help strengthen the paper:

1. It appears that the parameter $k$ in Equation (2) plays a significant role in the proposed method. An ablation study on $k$ would help clarify its impact on performance.
2. In line 175, the authors mention "decomposing the space of original text embeddings" using PCA. PCA typically requires a set of vectors—could the authors clarify what this set consists of in this context?
3. The exposition in Section 3.2 could benefit from further revision. In particular, the definition of $G_{\theta}$ in Equation (3) is unclear. Do the authors intend for $G_{\theta}$ to denote any matrix satisfying Equation (3), or is it defined in a more specific way?
4. What would be the performance if $\sigma < 1$, for example $\sigma = 0.5$ or $\sigma = -1.3$? Some clarification or empirical discussion would help readers understand the the impact of the variance scaling up under such settings.

---

> ### Author Rebuttal · Authors · 2025-07-31
>
> We sincerely thank you for your thoughtful and constructive feedback. Below, we address each of your comments in turn. If you need to check the technical appendix, it can be found in the submitted supplementary materials.
>
> ---
> ### **[W1] An ablation study of the parameter $k$.**
>
> As described in `lines 178–181` of our manuscript, **we automatically determine $k$**, the number of top principal components, via the Maximum Distance to Chord (MDC) method [1], which identifies an optimal cutoff, also referred to as the *elbow point*. We chose this adaptive strategy instead of using a fixed $k$ because the number of meaningful dimensions in the principal and residual subspace can vary significantly across input prompts, especially given their variable user's text lengths.
>
> To empirically validate our choice of an adaptive strategy, we conducted an ablation comparing our MDC-based automatic selection of $k$ with fixed-$k$ alternatives. To this end, we tested fixed $k$ values ranging from 10 to 60 in increments of 10 (see Tables below).  For additional insight into the typical scale of dimensionality, we computed the average MDC-derived $k$ for each task, resulting in approximately ~42.0 for Text-to-Image (T2I) and ~23.0 for Text-to-Video (T2V).
>
> - *Fixed $k$ ablation studies for text-to-image (Rarebench)*
>
> | $k$ | SD3.0 | FLUX-schnell | FLUX-dev |
> |---|---|---|---|
> | **ours** | **76.7** | **76.0** | **75.3** |
> | 10 | 73.2 | 73.3 | 70.0 |
> | 20 | 74.0 | 73.8 | 71.2 |
> | 30 | 74.8 | 74.6 | 73.0 |
> | 40 | 75.8 | 75.1 | 74.2 |
> | 50 | 74.1 | 72.5 | 70.8 |
> | 60 | 65.2 | 70.3 | 65.2 |
>
> - *Fixed $k$ ablation studies for text-to-video (Rarebench)*
>
> | $k$ | CogVideo-5B |
> |---|---|
> | **ours** | **66.6** |
> | 10 | 63.5 |
> | 20 | 65.3 |
> | 30 | 64.7 |
> | 40 | 60.0 |
> | 50 | 55.7 |
> | 60 | 51.8 |
>
> As these tables show, we can see that the automatically selected $k$ in ToRA outperforms any fixed $k$. Furthermore, we found that using ToRA yields higher scores than not using it in most cases, regardless of the value of $k$ we chose. (*e.g.,* the original RareBench average for SD 3.0 is 60.9).
>
> Inspired by the reviewer’s insightful suggestion, we recognized that additional ablation studies could be conducted, incorporating adaptive shifts of ±15% per sample from the determined $k$ derived by MDC method for single prompt to evaluate performance.  This experiment truly strengthens our paper’s standing. The results of these ablation studies of the adaptive approach are presented below.
>
> - *Adaptive $k$ ablation studies for text-to-image (Rarebench)*
>
> | Adaptive $k$ | SD3.0 | FLUX-schnell | FLUX-dev |
> |---|---|---|---|
> | -15% | 72.9 | 73.8 | 73.1 |
> | -10% | 73.4 | 74.1 | 73.7 |
> | -5% | 75.2 | 75.4 | 74.4 |
> | **ours** | **76.7** | **76.0** | **75.3** |
> | +5% | 72.5 | 73.3 | 69.3 |
> | +10% | 69.8 | 70.9 | 65.0 |
> | +15% | 66.1 | 69.5 | 62.2 |
>
> Additional implementation details and expanded discussions will be provided in the Appendix of the revised manuscript.
>
> *[1] David H. Douglas and Thomas K. Peucker. Algorithms for the reduction of the number of points required to represent a digitized line or its caricature. Cartographica: The International Journal for Geographic Information and Geovisualization, 10:112–122, 1973.*
>
> ---
> ### **[W2] Need clarifying the target vector set for PCA decomposition**
>
> Yes, PCA requires a set of vectors. In our context, this set corresponds to the **sequence of text‑token embeddings from the user’s single input prompt**.
> To provide a concrete example using the SD 3.0 architecture, the text encoder outputs a tensor of token embeddings, denoted as $\mathbf{e} \in \mathbb{R}^{L \times D}$, where $L=333$ is fixed sequence length (composed by the user's prompt tokens and special tokens like [SOS] and [PAD]), and $D=4096$ is the embedding dimension.
>
> These embeddings are then propagated through a stack of a joint-attention blocks. At each block $b$, we extract the input embedding matrix $\mathbf{e}^b$, representing the text features before entering the block, which can be viewed as a set of $L=333$ vectors, each with $D=4096$ dimensions.
> We then apply PCA to this matrix $\mathbf{e}^b$. Technically, this PCA is implemented via Singular Value Decomposition (SVD), where the matrix $\mathbf{e}^b$ is decomposed by $U\Sigma V^T$. In this decomposition, the rows of the matrix $V^T \in \mathbb{R}^{D \times D}$ consist of the eigenvectors that represent the principal components of the embedding space. This meant that the target vectors for PCA is a set of text token embeddings from a single prompt. Following PCA, if we apply Token Spacing method to these vector set, it leads to an increased separation of text tokens along the principal components $V^T[:k, :]$.
>
> ---
>
> ### **[W3] Unclear interpreting the role and construction of $\mathbf{G}_\theta$ in `Equation (3)`**
>
> We clarify that $G_\theta$ is not an arbitrary matrix satisfying the `Equation (3)`, but rather a **specifically constructed orthogonal matrix** (a Givens rotation [2]), designed to align the dominant residual component with the semantic direction. Givens rotation is a type of orthogonal transformation represented by orthogonal matrix. We utilize this matrix because the eigenvectors must preserve their orthogonal property.
>
> More concretely, after decomposing the text embedding space via PCA, we obtain the residual subspace spanned by the singular vectors {$v_{k+1}, \dots, v_n$}, where $v_{k+1}$ is the most significant among the residual directions.
> We define the semantic vector $s = e_{\text{cond}} - e_\emptyset$,
> representing the difference between conditioned and unconditioned embeddings (as used in classifier-free guidance). We then project $s$ onto the residual subspace (denoted $s_\text{proj}$) to remove its principal components.
>
> The Givens rotation matrix $G_\theta$ is explicitly constructed to rotate the vector $v_{k+1}$ toward the normalized projected semantic vector $\frac{s_\text{proj}}{\Vert s_\text{proj} \Vert}$, while leaving all other residual basis vectors orthogonal and unchanged. Note that, $\theta$ is not an argument but instead refers to the resulting angle derived from the alignment process. This operation, as expressed in `Equation (3)`, ensures semantic alignment within the residual subspace.
> In short, the matrix $G_\theta$ is a *rotation operator* constructed to align the dominant residual direction with the projected *semantic vector*, and it can be computed directly without any learning or iterative optimization. We will revise `Section 3.2` to clearly reflect this clarification.
>
> *[2] Wallace Givens. Computation of plane unitary rotations transforming a general matrix to
> triangular form. Journal of the Society for Industrial and Applied Mathematics, 6:26–50, 1958. ISSN 03684245.*
>
> ---
>
> ### **[W4] Additional ablation studies on how a more diverse range of $\sigma$ affects performance.**
>
> The reviewer raised the question regarding a broader range of $\sigma$ values, including those with negative signs.
>
> - **Negative $\sigma$ values (*e.g.,* $\sigma = -1.3$):** In our method, $\sigma$ serves as a scaling factor that amplifies or suppresses singular values. Since both variance and singular values are mathematically **non-negative**, multiplying them by a negative $\sigma$ would reverse the direction of the corresponding semantic axes. This reversal could severely distort semantic relationships between tokens, leading to a disruption of the semantic space and ultimately causing mode collapse. For this reason, we do not consider negative $\sigma$ values.
>
> - **Performance results of more various $\sigma$ values in $0< \sigma < 1$ :** We provide an ablation study on $\sigma$ in `Appendix Figure 15`. While we did not test values below $0.7$ in the original submission, the results at $\sigma = 0.7$ already show complete failure, with outputs collapsing to fully noise. In response to the reviewer’s interest, we have conducted additional experiments with $\sigma = 0.5$, which also resulted in mode collapse of joint-attention in MM-DiT, producing null outputs. Qualitative evidence of this phenomenon can be found in `Fig. 4`. As highlighted in the yellow boxes of `Fig. 4`, setting $\sigma$ too low reduces variance excessively, leading to failed image and video generation. We attached the quantitative results associated with these ablation studies in the table below.
>
> | $\sigma$ | $0.1$ | $0.2$ | $0.3$ | $0.4$ | $0.5$ | $0.6$ | $0.7$ |
> |:---|:---:|:---:|:---:|:---:|:---:|:---:|:---:|
> | Rarebench avg. score (SD 3.0) | $0.0$ *(fully noise)* | $0.0$ (fully noise) | $0.0$ *(fully noise)* | $0.0$ *(fully noise)* | $0.0$ *(fully noise)* | $0.0$ *(fully noise)* | $0.0$ *(fully noise)* |

---

### Official Review · Reviewer_KirX · 2025-07-03

**Clarity:** 4
**Significance:** 3
**Originality:** 3
**Rating:** 5
**Confidence:** 4

**Summary:**

This work introduces TORA, a training-free method to enhance rare semantic emergence by expanding text token embeddings’ representational basins via variance scale-up before joint-attention blocks. By amplifying local isotropy and refining residual alignment, TORA improves semantic distinguishability without external modules or retraining. It achieves robust generalization across text-to-image, text-to-video, and image editing tasks, demonstrating superior performance on rare prompts while preserving common concept accuracy. The method’s low computational overhead and compatibility with existing pipelines make it a practical solution for boosting MM-DiT’s semantic fidelity. All code will be publicly released.

**Questions:**

1. Does ToRA work for other problems like missing objects or wrong numbers of objects?

**Ethical Concerns:**

["NO or VERY MINOR ethics concerns only"]

**Final Justification:**

This work introduces TORA, which is a highly effective, training-free approach to improve semantic alignment. By enhancing the local isotropy of text embeddings through variance scale-up and residual alignment, it successfully addresses a key challenge without needing extra data or modules. Its effectiveness is clearly demonstrated by strong generalization across text-to-image, text-to-video, and editing tasks, alongside state-of-the-art performance on rare prompts. These significant contributions justify my score of 5.

**Limitations:**

yes, the limitations have been addressed

**Quality:**

4

**Strengths And Weaknesses:**

Strengths:
1. TORA achieves rare semantic alignment without requiring additional training, data, or external modules.
2. TORA enhances local isotropy in text embeddings by expanding representational basins through variance scale-up and refining residual alignment.
3. TORA generalizes effectively across diverse text-to-vision tasks (text-to-image, text-to-video, and image editing) and achieves state-of-the-art performance on rare prompts.

Weaknesses:
Hard to find many weaknesses, it's a strong paper that is well written, with a clear argument, clear analysis, clear solution that seems to work after both quantitative and qualitative results.

---

> ### Author Rebuttal · Authors · 2025-07-31
>
> We sincerely thank you for your thoughtful and constructive feedback. Below, we address each of your comments in turn.
>
> ---
>
> ### **[Q1] Does ToRA work for other problems like missing objects or wrong numbers of objects?**
>
> Yes . ToRA performs well in mitigating missing objects and incorrect object counts. In `Figure 7` in our main manuscript, for example, for the prompt *"a thorny snake is coiling around a star‑shaped drum"*, baseline models fail to produce the *snake*, whereas ToRA successfully generates both the *snake* and the *drum*. Likewise, for *"a four‑armed ninja with white hair in a blue gi"*, competing models generate fewer than *four arms*, but our methods generates exactly *four*, faithfully matching the text. `Table 3` quantifies this advantage on the GenEval benchmark [1], where ToRA outperforms baselines on the *Single‑Object*, *Multi‑Object*, and *Counting* categories.
>
> Furthermore, to thoroughly validate the performance gains on RareBench [2], we carefully categorized prompts involving multiple objects' presence and numerical counts, reassessing benchmark performance against the baseline. Specifically, we refined the GPT-4o instructions in `Appendix Section E.2` to explicitly evaluate multiple objects presence and numerical counts accuracy by posing two precise yes/no questions: *“Are all objects mentioned in the text present in the image?”* and *“Is the specified number in the text correctly reflected?”* We conducted experiments with seed=42 and $\sigma=1.30$, consistent with our main manuscript protocols. The resulting scores, which clearly demonstrate the model’s enhanced capability, are presented in the table below.
>
> | | $\text{SD3.0}$ (base) | $\text{R2F}_\text{SD3.0}$ | $\text{Ours}_\text{SD3.0}$ | $\text{FLUX-schnell}$ (base) | $\text{R2F}_\text{FLUX-schnell}$ | $\text{Ours}_\text{FLUX-schnell}$ | $\text{FLUX-dev}$ (base) | $\text{Ours}_\text{FLUX-dev}$ |
> | :--- | :---: | :---: | :---: | :---: | :---: | :---: | :---: | :---: |
> | **GenEval (common)** |
> | Counting (max: 1.00) | 0.63 | 0.65 | **0.66** | 0.70 | 0.59 | **0.77** | 0.74 | **0.74** |
> | Two Objects (max: 1.00) | 0.74 | 0.76 | **0.77** | 0.77 | 0.77 | **0.84** | **0.81** | 0.80 |
> | **Rarebench** |
> | Numerical Counts (max: 100.0) | 72.1 | 72.5 | **76.4** | 74.4 | 72.6 | **75.1** | 77.3 | **79.0** |
> | Multi-Objects Presence (max: 100.0) | 81.0 | 80.8 | **82.1** | 83.7 | 83.7 | **84.9** | 83.9 | **85.1** |
>
> Together, these results confirm that ToRA is effective not only for compositionality but also for missing‑object and miscount problems. These results will be included in our revised version.
>
> ---
>
> *[1] Ghosh, Dhruba, Hannaneh Hajishirzi, and Ludwig Schmidt. "Geneval: An object-focused framework for evaluating text-to-image alignment." Advances in Neural Information Processing Systems 36 (2023): 52132-52152.*
> *[2] Park, Dongmin, et al. "Rare-to-frequent: Unlocking compositional generation power of diffusion models on rare concepts with LLM guidance." arXiv preprint arXiv:2410.22376 (2024).*

---

> > ### Comment · Reviewer_KirX · 2025-08-05
> >
> > I appreciate the authors’ thorough replies. The previously noted issues have been resolved, so I will retain my original scores.

---

### Note · Authors · 2025-08-13

Dear SACs, ACs, and Reviewers,

We sincerely thank the reviewers, area chairs, and organizers for their tireless efforts. Your thoughtful comments have given us invaluable opportunities to strengthen our work and clarify our presentation.

In this closing note, we summarize the key contributions of our study, and humbly ask that they be considered in the final discussion:

1. Our method, ToRA, achieves strong empirical performance across diverse text-to-vision tasks (text-to-image, text-to-video, and text-driven image editing) without additional training (all reviewers).
2. ToRA provides technical benefits without relying on external modules or knowledge, showing improvements over diverse baselines and enabling the integration of existing approaches (all reviewers).
3. The paper is well written, clearly structured, and effectively presented (KirX, bpoM, 41Hj).
4. Our analyses provide logical and insightful explanations for ToRA’s effectiveness (KirX, 41Hj).

Importantly, we appreciate that the main weaknesses mentioned by the reviewer were not about denying our contributions or the validity of our arguments and supporting evidence, but were instead largely questions or requests for additional clarification, such as:

1. $k$ ablation studies: While $k$ is automatically selected per sample in our method (not manually tuned), we acknowledged bpoM’s and 41Hj’s suggestions, and provided additional controlled experiments in the rebuttal for a more complete view.
2. Clearer explanation of the PCA process: Following the standpoints of reviewers 41Hj and VqR3, we specified the exact locations (e.g., lines, sections) where the revised manuscript will enhance reader comprehension.
3. Clarification of already-initialized experiments: We removed any possible ambiguity by adding further explanations and, where feasible, additional experiments during the rebuttal.

Once again, we greatly appreciate the reviewers’ active engagement, which allowed us to clarify claims, strengthen proofs, and elaborate on key technical details. Ultimately, our aim is to shift the use of MM-DiT in generative modeling from externally enforcing semantic alignment to naturally uncovering latent semantics, thereby advancing the multimodal generative research community.

Best regards,
Authors

---

### Decision · Program_Chairs · 2025-09-17

**Decision:**

Accept (poster)

**Comment:**

This work introduces TORA, a simple yet effective method to enhance rare semantics inside MM-DiT. Without additional training, denoising-time optimization, or reliance on external modules, TORA expands text token embeddings’ representational basins via variance scale-up before joint-attention blocks. Promising results are demonstrated across text-to-vision tasks, including text-to-image, text-to-video, and text-driven image editing.

Initially, the reviewers raised several concerns, which are briefly outlined below:

* Reviewer KirX: The generalization to other problems.

* Reviewer bpoM: More ablation studies, and PCA details.

* Reviewer 41Hj: Performance on common prompts, and ablation on k.

* Reviewer VqR3: Writing issue, method details, baselines, compute overhead, and failure mode discussion.

The rebuttal and subsequent author-reviewer discussions effectively addressed most of the reviewers' concerns. After carefully considering the reviews, rebuttal, and discussion, the AC concurs with the reviewers’ assessment and thus recommends acceptance of the paper. Finally, the authors are encouraged to incorporate the rebuttal experiments into the manuscript and address the reviewers’ feedback (e.g., the writing refinement) in the final revision.